# Hydrobiological Aspects of Fatty Acids: Unique, Rare, and Unusual Fatty Acids Incorporated into Linear and Cyclic Lipopeptides and Their Biological Activity

**Valery M. Dembitsky**

Centre for Applied Research, Innovation and Entrepreneurship, Lethbridge College, 3000 College Drive South, Lethbridge, AB T1K 1L6, Canada; valery.dembitsky@lethbridgecollege.ca or valery@dembitsky.com

**Abstract:** The study of lipopeptides and their related compounds produced by various living organisms from bacteria to marine invertebrates is of fundamental interest for medicinal chemistry, pharmacology, and practical clinical medicine. Using the principles of retrosynthetic analysis of linear and cyclic peptides, the pharmacological activity of unique, unusual, and rare fatty acids (FA) that are part of natural lipopeptides was investigated. To search for new biologically active natural metabolites from natural sources, more than 350 FA incorporated into linear and cyclic peptides isolated from bacteria, cyanobacteria, microalgae, marine invertebrates, fungal endophytes, and microorganisms isolated from sediments are presented. Biological activities have been studied experimentally in various laboratories, as well as data obtained using QSAR (*Quantitative Structure-Activity Relationships*) algorithms. According to the data obtained, several FA were identified that demonstrated strong antibacterial, antimicrobial, antifungal, or antitumor activity. Along with this, FA have been found that have shown rare properties such as antiviral, antidiabetic, anti-helmintic, anti-inflammatory, anti-psoriatic, anti-ischemic, and anti-infective activities. In addition, FA have been found as potential regulators of lipid metabolism, as well as agents for the treatment of acute neurological disorders, as well as in the treatment of atherosclerosis and multiple sclerosis. For 36 FA, 3D graphs are presented, which demonstrate their predicted and calculated activities.

**Keywords:** bacteria; cyanobacteria; microalgae; fungal endophytes; QSAR; lipopeptides; fatty acids

## 1. Introduction

It is known that peptides are natural biological molecules, are found in all living organisms living on planet Earth and apparently play a key role in all types of biological activity [1–9]. Linear and/or cyclic lipopeptides and depsipeptides of bacteria, cyanobacteria, marine invertebrates, and fungi and fungal endophytes are of great interest, both from the point of view of academic research and from a purely practical point of view, as well as their use in pharmacology and medicine [1,2,10–19]. Both types of peptides have shown different biological activities, which include antiviral, anti-inflammatory, antibacterial, antifungal, anti-tumor, and other activities [3,5,12,15,17]. Lipopeptides are molecules consisting of a lipid fragment connected to peptides with an ester bond, while depsipeptides are peptides in which one or more amide groups are replaced by a corresponding ester [20–25].

When scanning more than 30,000 peptide structures, especially lipopeptides and depsipeptides isolated from marine and terrestrial organisms, it was found that these compounds in the absolute majority (over 80%) contain fragments of saturated FA (C6:0–C26:0), about 15% contain *iso-*, *anteiso-* and methyl-branched chain FA (C6:0–C24:0), and about 4–5% contain unsaturated FA. A few exceptions to FA not included in this review are amino fatty (carboxylic) acids and those mentioned above. Unique, rare, and unusual FA make up less than one percent of the total peptides screening.

This review focuses on this rare group of FA that are part of lipopeptides or depsipeptides and are covalently linked. The lipopeptides used in this review were found and

isolated from bacteria, microalgae, cyanobacteria, marine invertebrates, fungal endophytes, and phytopathogenic microorganisms. The biological activity of many lipopeptides has not been studied. The biological activity of FA, as well as some amino acids that make up the peptides, was studied using the PASS software [26–28]. In addition, 3D graphs of the most interesting FA from the point of view of their pharmacological activity are presented.

## 2. Bacterial and Cyanobacterial Linear and Cyclic Peptides and Their Fatty Acids

It is known that cyanobacteria (or blue–green algae) belong to the division of Gram-negative bacteria capable of photosynthesis, accompanied by the release of oxygen, which have a blue–green color, and they are called cyanobacteria [29–33]. From the Archean rocks of Western Australia, fossils of cyanobacteria have been identified, the age of which is determined at 3.5 billion years [34,35].

Marine cyanobacteria have been attracting increasing attention for probe and drug discovery due to the high incidence of structurally novel bioactive secondary metabolites that complement those known from terrestrial sources [36–38]. These natural products are predominantly modified peptides and depsipeptides, polyketides, and peptide–polyketide hybrids, many of which are cyclic and oftentimes halogenated [39–41]. Cyanobacteria produce many bioactive compounds of various chemical structures, with about 40% of them being lipopeptides [36–45].

Using cytotoxic cyclic depsipeptides as an example, we want to show which FA are incorporated into peptides and are of interest to academic science for their unusual chemical structures. All the discovered cryptophycins can be retrosynthetically divided into four *sub*-fragments or subunits, namely A–D. Unit A represents the most exotic fragment, (5*S*,6*S*,*E*)-5-hydroxy-6-(2*R*,3*R*)-3-phenyloxiran-2-yl)hept-2-enoic acid. Fragment B can be derived from D-O-methyltyrosine and represents the (*R*)-2-amino-3-(3-chloro-4-methoxyphenyl)-propanoic acid. Unit C represents 3-aminopropanoic acid (known as β-alanine). Finally, (*S*)-2-hydroxy-4-methyl-pentanoic acid, also known as L-leucic acid (or 2-hydroxyisocaproic acid), constitutes the fragment D (Figure 1).

**Figure 1.** Molecular structure of cryptophycin-1 and retrosynthetic division on the subunits. Shortly after the discovery of cryptophycin-1, biological screening assays showed a high cytotoxicity against human cervical carcinoma (KB) and human breast adenocarcinoma (MCF-7).

Cryptophycin-1 is the most important member of the cryptophycins family. It was first isolated from *Nostoc* sp. ATCC 53787 by Merck scientists as an antifungal agent [46]. However, a detailed study of cryptophycin-1 showed that it was too toxic to be of practical use, at least as an antifungal agent. Subsequent studies have shown that cryptophycin-1 is an active microtubule depolymerization agent, showing excellent activity against a wide range of solid tumors implanted in mice, including multiple drug-resistant tumors [47]. In addition, cryptophycin-1 suppresses tubulin dynamics and induces apoptosis [48,49].

Considering the above experimental data, we tested the biological activity of cryptophycin-1 and its subunits A, B, C and D included in this depsipeptide. The data of the PASS analysis are shown in Table 1. The data obtained by various groups of scientists are fully confirmed by the PASS program. The dominant activity was antifungal with a confidence level of 84.5%, in addition, antitumor activity was found with a confidence level of 77%, antineoplastic

(solid tumors) activity with a reliability of 63%, and apoptosis agonist-62%. Thus, PASS fully confirmed the experimental data, and thus the program showed the correctness and quality of its work.

**Table 1.** Biological activity cryptophycin-1 and its subunits A, B, C and D.

| Fragment Name | Predicted Biological Activity, Pa * | Reported Activity [46–49] |
|---|---|---|
| Cryptophycin-1 | Antifungal (0.845) Antimitotic (0.784) Antineoplastic (0.771) Antineoplastic (solid tumors) (0.631) Apoptosis agonist (0.625) | Antifungal Anticancer Apoptosis |
| Subunit A (1) | Antineoplastic (0.856) Antileukemic (0.783) Antiviral (Arbovirus) (0.775) Antifungal (0.773) Cytoprotectant (0.675) Apoptosis agonist (0.669) Fibrinolytic (0.664) Antimitotic (0.634) Antithrombotic (0.624) | No published data |
| Subunit B | Preneoplastic conditions treatment (0.836) Antiviral (Arbovirus) (0.774) Acute neurologic disorders treatment (0.759) Antiviral (Picornavirus) (0.715) | No published data |
| Subunit C | Fibrinolytic (0.814) Preneoplastic conditions treatment (0.803) Antiviral (Arbovirus) (0.760) Antimutagenic (0.737) Anticonvulsant (0.676) Antiviral (Picornavirus) (0.662) | No published data |
| Subunit D | Anti-ischemic, cerebral (0.921) Sclerosant (0.871) Antihypertensive (0.756) Anti-hypoxic (0.741) Antiviral (Arbovirus) (0.740) | No published data |

\* Only activities with Pa > 0.5 are shown. The numbers in brackets show the level of biological activity. 100% activity level is 1.000.

Interestingly, the antifungal activity that is characteristic of cryptophycin-1 was found only in subunit A and was not found in other subunits B, C and D. From this it can be inferred that subunit A makes a significant contribution to the overall pool of activity of this depsipeptide (Table 1).

Cryptophycin-1, similar to other cryptophycins, is a class of 16-membered highly cytotoxic macrocyclic depsipeptides produced by the cyanobacterium from the strain *Nostoc* [46]. So, (5*S*,6*S*,*E*)-5-hydroxy-6-(2*R*,3*R*)-3-phenyloxiran-2-yl)hept-2-enoic acid (**1**, structure shown in Figure 2) is a fragment of the cryptophycins 1, 2, 16, 21, 23, 24, 28, 31, 38, 50, 52, 54, 176 and 326 [50,51], and (2*E*,5*S*,6*R*,7*E*)-5-hydroxy-6-methyl-8-phenylocta-2,7-dienoic acid (**2**) was detected in cryptophycins 3, 4, 17, 18, 19, 29, 31, 43, 45, 46, 49, 175 and 327 (Figure 3). (*S*,*E*)-5-Hydroxy-6-(2*R*,3*R*)-3-phenyloxiran-2-yl)hex-2-enoic acid (**3**) was found in the structure of cryptophycin 28, (*R*,2*E*,7*E*)-5-hydroxy-8-phenylocta-2,7-dienoic acid (**4**) was detected in cryptophycin 40, and (6*S*,7*S*,*Z*)-6-hydroxy-7-((2*R*,3*R*)-3-phenyloxiran-2-yl)oct-3-enoic acid (**5**) was isolated from cryptophycin 327.

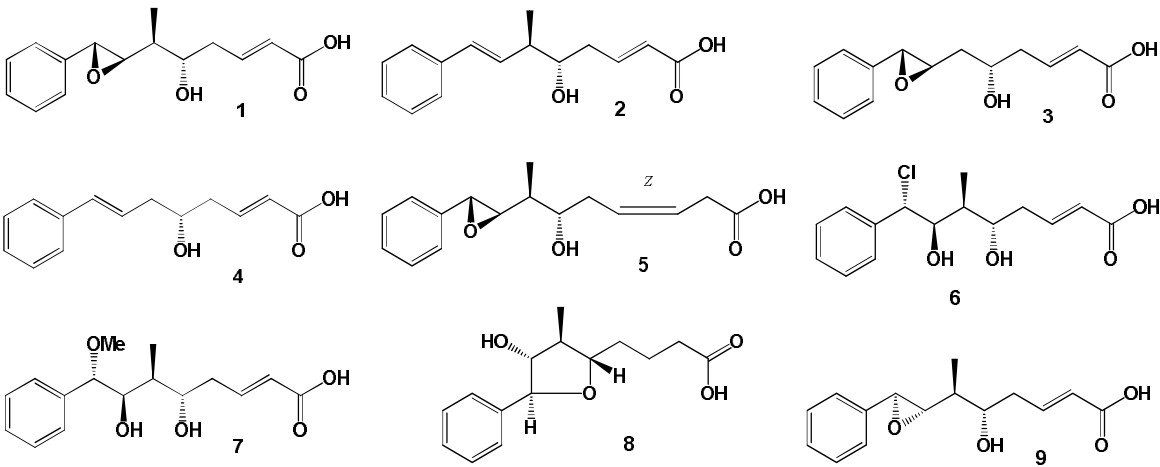

**Figure 2.** The genus of cyanobacteria *Nostoc*: (**a**), *Nostoc* sp.; (**b**), *Nostoc commune*; (**c**), *N. commune*; (**d**), *N. commune*, which inhabit various environments, such as the bottom of both fresh and salt lakes, and form colonies consisting of filaments of moniliform cells in a gelatinous membrane. Cyanobacteria of the genus Nostoc produce saturated and unsaturated fatty acids, lipopeptides, depsipeptides, oligopeptides and toxins. All photos are taken from sites where permission was granted for non-commercial use.

**Figure 3.** Chemical diversity of FA incorporated into cryptophycins.

Cryptophycin-38, -326, and -327 were isolated from the terrestrial cyanobacterium *Nostoc* sp. GSV 224 by Chaganty and co-workers [52], and cryptophycin-2, -21, -46, -175, and -176 have been identified from the MeCN-CH$_2$Cl$_2$ extract of the same blue–green algae [50,53,54].

(5$S$,6$S$,7$R$,8$S$,$E$)-8-Chloro-5,7-dihydroxy-6-methyl-8-phenyloct-2-enoic acid (**6**) has been detected in the cryptophycins 8 and 55 which were isolated from *Nostoc* sp. GSV, and (5$S$,6$S$,7$R$,8$S$,$E$)-5,7-dihydroxy-8-methoxy-6-methyl-8-phenyloct-2-enoic acid (**7**) was found in the cryptophycins 9 and 10 [55]. 4-(2$S$,3$R$,4$R$,5$S$)-4-Hydroxy-3-methyl-5-phenyltetrahydrofuran-2-yl)-butanoic acid (**8**) has been found in the linear cryptophycins 6 and 7, and (5$S$,6$S$,$E$)-5-hydroxy-6-((2$S$,3$S$)-3-phenyloxiran-2-yl)hept-2-enoic acid (**9**) was detected in cryptophycin-101. The anticancer activity of natural, semi-synthetic and synthetic cryptophycins has been studied in detail and summarized in several review articles [50–57]. The biological activity of the fatty acids incorporated into cryptophycins is shown in Table 2.

**Table 2.** Predicted biological activity of subunit A (FA) incorporated into cryptophycins.

| No. | Predicted Biological Activity, Pa * |
|:---:|:---:|
| 2 | Antiviral (Arbovirus) (0.780); Antineoplastic (0.766); Antifungal (0.707) Lipid metabolism regulator (0.705); Hypolipemic (0.694) Preneoplastic conditions treatment (0.667); Apoptosis agonist (0.664) |
| 3 | Antiviral (Arbovirus) (0.804); Antineoplastic (0.795); Antidiabetic (0.701) Antifungal (0.695); Anti-inflammatory (0.680); Cytoprotectant (0.661) Immunosuppressant (0.634); Anti-hypercholesterolemic (0.629) Antithrombotic (0.611) |
| 4 | Antiviral (Arbovirus) (0.833); Lipid metabolism regulator (0.827) Anti-inflammatory (0.765); Hypolipemic (0.759); Cytoprotectant (0.729) Anti-hypercholesterolemic (0.715); Antineoplastic (0.697) |
| 5 | Antineoplastic (0.850); Antileukemic (0.781); Antifungal (0.727) Anti-hypoxic (0.702); Antiviral (Arbovirus) (0.683); Cytoprotectant (0.638) |
| 6 | Antineoplastic (0.764); Apoptosis agonist (0.762) Antiviral (Arbovirus) (0.728); Antimitotic (0.664); Antifungal (0.560) |
| 7 | Antiviral (Arbovirus) (0.730); Antifungal (0.658); Antineoplastic (0.611) |
| 8 | Antineoplastic (0.776); Antifungal (0.694); Anti-helmintic (0.691) Antidiabetic (0.660); Acute neurologic disorders treatment (0.625) Antibacterial (0.624); Antiviral (Arbovirus) (0.621) Antiviral (Picornavirus) (0.608) |
| 9 | Antineoplastic (0.856); Antileukemic (0.783); Antiviral (Arbovirus) (0.775) Antifungal (0.773); Anti-hypercholesterolemic (0.742); Apoptosis agonist (0.669) |

* Only activities with Pa > 0.5 are shown. The numbers in brackets show the level of biological activity. 100% activity level is 1.000.

The analysis of the predicted biological activity of subunit A incorporated into cryptophycins, which is presented in Table 2, shows that all these fatty acids have three dominant properties such as antiviral (*Arbovirus*), antifungal and moderate antineoplastic activities with some variations in different fatty acids. The 3D graph of the activities of cryptophycin-1 and its four subunits A, B, C and D is shown in Figure 4.

It is known that natural compounds containing an acetylenic (triple) bond have been isolated from many species of plants, fungi, fungal endophytes, and various marine invertebrates [58–63]. Numerous studies have shown that many of these metabolites exhibit various biological activities, such as antibacterial, antimicrobial, antifungal, antitumor, and other medicinal properties [58,64–66]. Various species of freshwater and marine plants, macrophytes, microalgae, cyanobacteria and some other aquatic organisms produce a wide variety of different bioactive molecules containing acetylenic bonds [58,64–66].

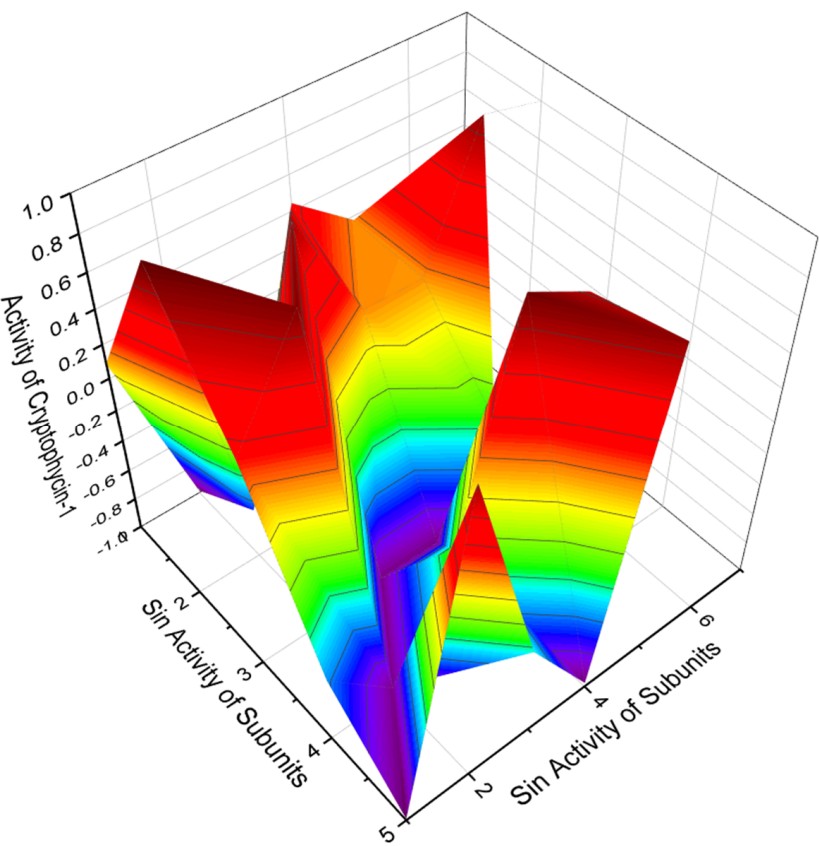

**Figure 4.** 3D graph showing the predicted and calculated activity of cryptophycin−1 and its four subunits A in Figure 1 ((5*S*,6*S*,*E*)-5-hydroxy-6-((2*R*,3*R*)-3-phenyloxiran-2-yl)-hept-2-enoic acid); B in Figure 1 ((*R*)-2-amino-3-(3-chloro-4-methoxyphenyl)-propanoic acid); C in Figure 1 (3-aminopropanoic acid) and D in Figure 1 ((*S*)-2-hydroxy-4-methylpentanoic acid), with the highest degree of confidence being more than 74%. Designations A, B, C, and D are shown in Figure 1. The depsipeptide named cryptophycin−1 was produced by the cyanobacterium from the strain *Nostoc*.

The genus *Lyngbya* is the most abundant and available cyanobacterial species and is distributed throughout the world in tropical and subtropical regions. The species *L. majuscule* (see Figure 5), *L. martensiana*, *L. aestuarii* and *L. wollei* are currently the most important species of their genus *Lyngbya* and synthesize many secondary metabolites including lipopeptides.

Through numerous studies, it has been established that the widespread tropical cyanobacterium *Lyngbya majuscula* synthesizes more than 30% of all natural products derived from all marine cyanobacteria [67], which exhibit various activities, including antiproliferative, antifeedant, anti-inflammatory, molluscicidal, and immunosuppressive properties. It has been established that more than half of the known secondary metabolites are either linear or cyclic lipopeptides, some of which contain an acetylene fragment [67]. The linear lipopeptides called apramides A, B, and G were found in the cytotoxic fraction of *L. majuscula* collected at Apra Harbor (Guam) [68], and apramide G showed strong cytotoxic activity against KB and LoVo cells, respectively [69]. (*R*)-2-methyloct-7-ynoic acid (**10**, see Figure 6 for structure, and biological activity shown in Table 3) was found in apramide A and G, and apramide B contained oct-7-ynoic acid (**11**).

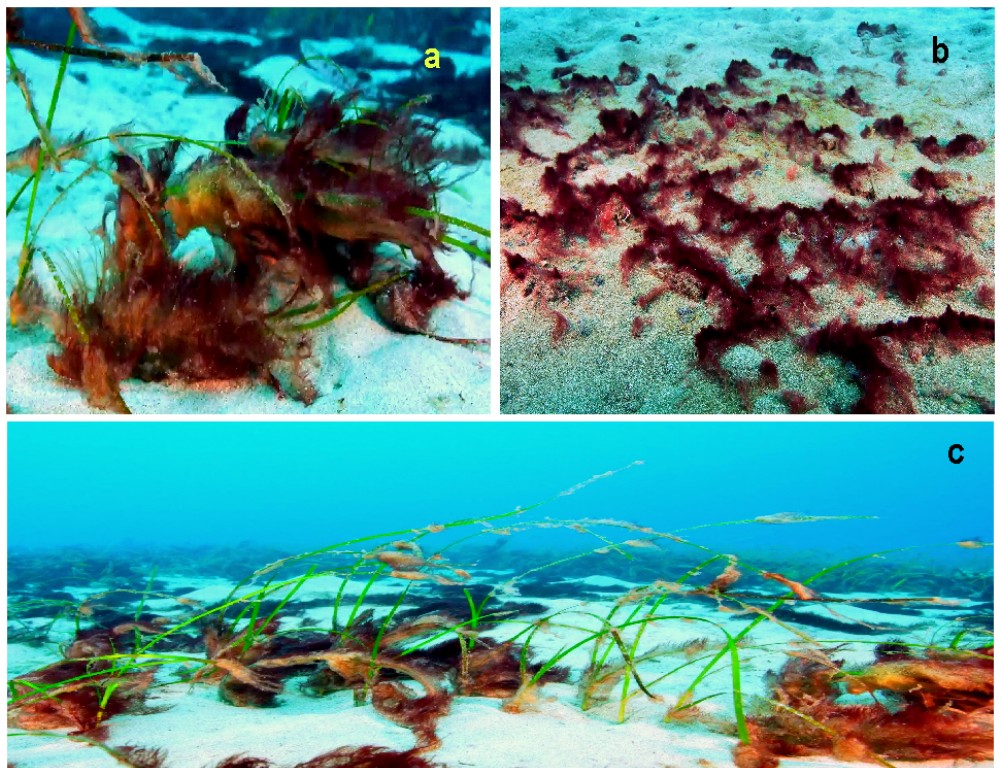

**Figure 5.** The unbranched filamentous cyanobacterium, *Lyngbya majuscula* (**a**–**c**), commonly referred to as "mermaids' hair" or "fireweed," is found in coastal tropical and subtropical marine and estuarine areas around the world. The cyanobacterium usually grows up to 30 m below the surface and attaches to rocks, sand, or algae (**a**,**c**), or can form large mats (**b**).

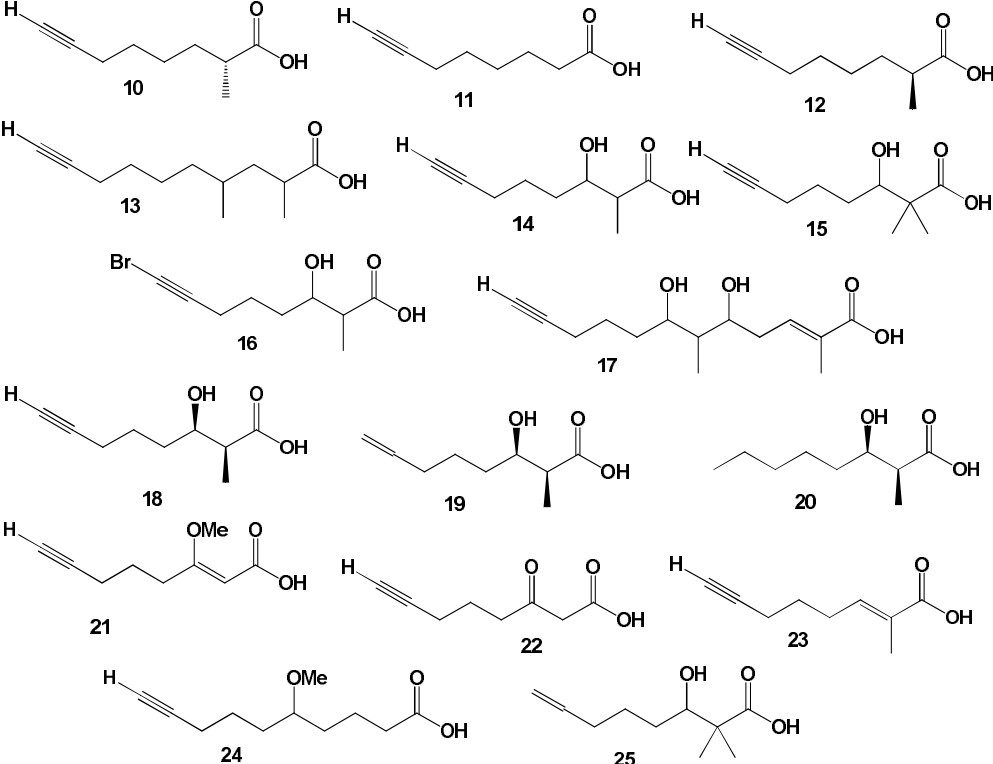

**Figure 6.** Fatty acids from peptide molecules produced by cyanobacteria belonging to the strain *Nostoc*.

Dragonamide, dysidenamide, nordysidenin, and pseudodysidenin were isolated from *L. majuscula* collected from the beach at Boca del Drago Beach, Bocas del Toro, Panama. Dragonamide containing (*S*)-2-methyloct-7-ynoic acid (**12**) showed strong cytotoxic activity against P-388, A-549, HT-29, and MEL-28 cancer cells [70], and was synthesized four years later after its discovery [71]. The n-BuOH extract of *L. majuscula* contained a cytotoxic, linear lipotetrapeptide named carmabin A, which contains 2,4-dimethyldec-9-ynoic acid (**13**) [72]. The depsipeptides named antanapeptins A–D, of which antanapeptin A and D contain 3-hydroxy-2-methyloct-7-ynoic acid (**14**), were isolated from *L. majuscula* of the Antany Mora collection (Madagascar), and both depsipeptides showed moderate cytotoxic activity against neuroblastoma-2A cells in mice [73].

Pitipeptolide A, which belongs to the cyclodepsipeptides, was isolated from extracts of the cyanobacterium L. majuscula, which lives around the Piti Bomb Holes (Guam Reefs), where marine blooms caused by these cyanobacteria occur with a certain periodicity. This lipopeptide containing a 2,2-dimethyl-3-hydroxy-7-octynoic acid (**15**) [74] exhibited weak cytotoxicity against LoVo cancer cells but possessed moderate antimycobacterial activity and stimulated elastase activity. Other pitipeptolides D, E, and F also contained this fatty acid. Two depsipeptides named yanucamides A and B were found in lipid extracts of two cyanobacteria, *L. majuscula* and *Schizothrix* sp. collected at Yanuca Island (Fiji) [75]. Both lipopeptides contained the same fatty acid (**15**), which has previously been described as a major component of kulolide-1 and kulokainalide-1 isolated from the marine mollusk *Philinopsis speciosa* [76].

This acetylenic acid (**15**) was also found in the depsipeptides called wewakpeptins A and C, and both peptides showed cytotoxic activity against H460 human lung tumor and the mouse neuroblastoma-2A cell lines [77–79]. Additionally, another cyclic depsipeptide called georgamide was isolated from a non-identified Australian cyanobacterium [79], containing two hydroxy carboxylic acids, 2(S)-hydroxy-3(R)-methyl-pentanoic acid and FA (**15**), which are also present in wewakpeptins A and B [79].

Cytotoxic depsipeptide, which was named onchidin B, was isolated from extracts of the pulmonate mollusk *Onchidium* sp. and contained the 3-hydroxy-2-methyl-7-octynoic acid (**14**) [80–82].

Cyclic depsipeptide, kulolide-1 was isolated from a cephalaspidean mollusk, *Philinopsis speciosa*, and contains two carboxylic acids, L-3-phenyllacetic acid and the unprecedented (*R*)-3-hydroxy-2,2-dimethyl-7-octynoic acid (**15**). The isolated depsipeptide showed activity against L-1210 leukemia cells and P388 murine leukemia cells and caused a morphological change in 3Y1 rat fibroblast cells [83,84]. In addition, this mollusk yielded a linear peptide, pupukeamide, and an unprecedented macrolide, tolytoxin-23-acetate, which contained this acetylenic acid [84]. Interestingly, the lipopeptides kulolide-2, kulolide-3, kulokainalide-1, kulomoopunalide-1, kulomoopunalide-2, and tolytoxin 23-acetate were found in combined extracts of ((EtOH and CHCl$_3$/MeOH (1:1))) mollusk *Ph. speciosa*. Kulokainalide-1 contains acetylenic acid (**15**), and kulomoopunalide-1 and kulomoopunalide-2 contains (*R*)-3-hydroxy-2-methyl-7-octynoic acid (**14**) [84]. Widely present in lipopeptides, acetylenic acid (**15**) was also found in structures such as in the depsipeptides mantillamide, and dudawalamide A, isolated from extracts of the marine cyanobacterium *Lyngbya* sp. Both peptides show anticancer and antimalarial activity [85]. A cyclic depsipeptide, guineamide G with FA (**15**), is produced by the marine cyanobacterium *L. majuscula*, collected from Papua New Guinea, and exhibits potent brine shrimp toxicity and moderate cytotoxicity to a mouse neuroblastoma cell line [86]. Additionally, the cocosamides A and B from the lipophilic extract of a collection of *L. majuscula* from Cocos Lagoon (Guam), demonstrated activity against HT-29 cells, and also contained FA (**15**) [87]. FA (**15**) was present in cyclic depsipeptides, and the viequeamides A–F, which were discovered from a shallow subtidal collection of *Rivularia* sp.; viequeamide A is active against the H460 human lung cancer cell line [88]. Other cyclic depsipeptides, named dudawalamides A−D, were isolated from Papua New Guinea from the cyanobacterium *Moorea* sp., and FA (**15**) was found in

dudawalamide A as recently reported [89]. The 3D graph of the activity of fatty acids (**15**) is shown in Figure 7.

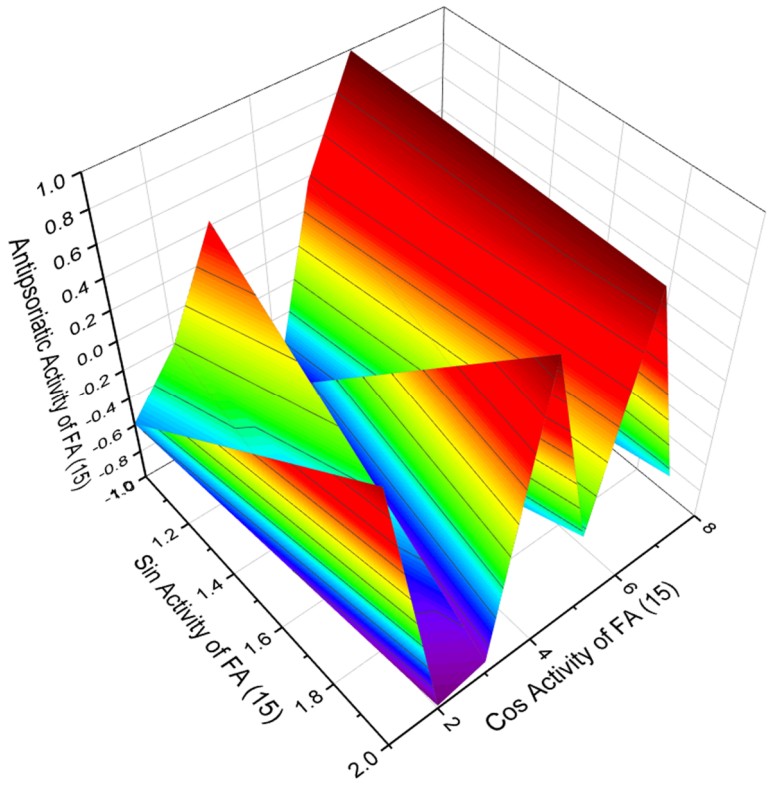

**Figure 7.** 3D Graph showing the predicted and calculated anti−psoriatic activity of a 2,2-dimethyl-3-hydroxy-7-octynoic acid (**15**). This FA is a subunit in many linear and cyclic lipopeptides that are synthesized by cyanobacteria belonging to the genus *Lyngbya*.

The Luesch group from the Florida University reported the isolation of cytotoxic cyclodepsipeptides, veraguamides A–F from a cyanobacterium *Symploca* cf. *hydnoides* at Cetti Bay, Guam [90], and veraguamides H, I–L from the marine cyanobacterium *Oscillatoria margaritifera* at the Coiba National Park, Panama [91], which contained 3-hydroxy-2-methyl-7-octynoic acid (**14**). Among them, veraguamides A and B are 8-bromo-3-hydroxy-2-methyl-7-octynoic acid (**16**)-containing cyclic peptides, while veraguamides K and L are acid (**16**)-containing linear peptides. The Okinawan marine cyanobacterium *Oceania* sp. produces a cytotoxic depsipeptide called odobromoamide containing an alkynyl bromide (**16**), and this demonstrated cytotoxic activity against HeLa S3 cells and broad-spectrum cytotoxicity against a panel of human cancer cell lines [92].

An alkynyl-containing cyclic depsipeptide, palauamide, containing 5,7-dihydroxy-2,6-dimethyldodec-2-en-11-ynoic acid (**17**), was extracted from a *Lyngbya* sp. from Palau. Palauamide showed strong cytotoxicity against KB cells with an $IC_{50}$ value of 13 nM [93]. Sitachitta and co-workers reported the isolation and identification of cyclic peptides, trungapeptins A, B, and C, containing 3-hydroxy-2-methyl-7-octynoic (**18**), 3-hydroxy-2-methyl-7-octenoic (**19**), and 3-hydroxy-2-methyl-7-octanoic acid (**20**), respectively [94]. Trungapeptin A exhibited potent brine shrimp toxicity and ichthyotoxicity at 10 ppm and 6.2 ppm, respectively.

In 2009, a hmoya-containing analog of hantupeptin A (**19**, 3-hydroxy-2-methyl-7-octynoic acid) was discovered from the marine cyanobacterium *Lyngbya majuscula* from Pulau Hantu Besar, Singapore [95]. Further, hantupeptin A afforded both brine shrimp toxicity at 10 ppm and strong cytotoxicity against the leukemia cell line MOLT-4 with an $IC_{50}$ value of 32 nM. The same FA (**19**) was detected in trungapeptin A, which was detected in the marine cyanobacterium *L. majuscula* collected from Trung Province, Thailand [96].

A sample of brown *Lyngbya polychroa* from Hollywood Beach, Fort Lauderdale, Florida yielded an impressive array of structurally diverse cytotoxic linear tetrapeptide–octynoates, the dragonamides A, B, C, D and E [97–99]. The dragonamides C and D showed weak activity, with $GI_{50}$ values of 56 and 59 μM against U2OS osteosarcoma cells, 22 and 32 μM against HT29 colon adenocarcinoma cells, and 49 and 51 μM against IMR-32 neuroblastoma cells, respectively. From isolated linear tetrapeptides, the dragonamides A and B contain FA (**12**), dragonamide C contains FA (**21**), dragonamide D contains FA (**22**) and dragonamide E contains FA (**23**).

**Table 3.** Predicted biological activity of FA from cyanobacteria of the strain *Nostoc*.

| No. | Predicted Biological Activity, Pa * |
|:---:|:---:|
| **10** | Neuroprotector (0.806); Sclerosant (0.779); Anticonvulsant (0.734) Acute neurologic disorders treatment (0.684); Anti-inflammatory (0.681) Antineoplastic (0.631); Preneoplastic conditions treatment (0.628); Anti-neurogenic pain (0.610) |
| **11** | Neuroprotector (0.827); Antineoplastic (0.708); Preneoplastic conditions treatment (0.672) Anticonvulsant (0.672); Antiviral (Arbovirus) (0.647); Psychostimulant (0.643) |
| **12** | Neuroprotector (0.806); Sclerosant (0.779); Acute neurologic disorders treatment (0.684) Antineoplastic (0.631); Preneoplastic conditions treatment (0.628) |
| **13** | Anticonvulsant (0.797); Hypolipemic (0.762); Acute neurologic disorders treatment (0.759) Neuroprotector (0.739); Sclerosant (0.727); Antineoplastic (0.638) |
| **14** | Sclerosant (0.767); Neuroprotector (0.748); Antineoplastic (0.666) Acute neurologic disorders treatment (0.645); Antiviral (Arbovirus) (0.595) |
| **15** | Anti-psoriatic (0.923); Antineoplastic (0.883); Neuroprotector (0.675) Antiviral (Arbovirus) (0.635); Neurodegenerative diseases treatment (0.620) Alzheimer's disease treatment (0.591) |
| **16** | Sclerosant (0.767); Antifungal (0.735); Antineoplastic (0.730) |
| **17** | Antineoplastic (0.812); Anti-inflammatory (0.763); Apoptosis agonist (0.691) |
| **18** | Sclerosant (0.767); Antineoplastic (0.666); Acute neurologic disorders treatment (0.645) |
| **19** | Acute neurologic disorders treatment (0.795); Sclerosant (0.754) Lipid metabolism regulator (0.749); Antiviral (Arbovirus) (0.704) |
| **20** | Sclerosant (0.910); Antiviral (Arbovirus) (0.784); Acute neurologic disorders treatment (0.747) Preneoplastic conditions treatment (0.714); Lipid metabolism regulator (0.667) |
| **21** | Antineoplastic (0.758); Neuroprotector (0.752); Antiviral (Arbovirus) (0.636) |
| **22** | Neuroprotector (0.756); Periodontitis treatment (0.744); Antineoplastic (0.680) Preneoplastic conditions treatment (0.655); Psychostimulant (0.545) |
| **23** | Antineoplastic (0.765); Neuroprotector (0.724); Lipid metabolism regulator (0.720) Apoptosis agonist (0.625); Acute neurologic disorders treatment (0.566) |
| **24** | Neuroprotector (0.800); Antineoplastic (0.738); Anticonvulsant (0.678) |
| **25** | Anti-psoriatic (0.924); Lipid metabolism regulator (0.889); Antineoplastic (0.867) |

* Only activities with Pa > 0.5 are shown.

Several linear alkynoic lipopeptides have been isolated from a Panamanian strain of the marine cyanobacterium *L. majuscula*, including carmabin A, dragomabin, and dragonamide A, which showed good antimalarial activity ($IC_{50}$ 4, 6, and 7.7 μM, respectively), whereas the non-aromatic analog, dragonamide B, was inactive [98]. The isolated linear lipopeptides dragomabin and dragonamide A and B contained fatty acid (**12**), while fatty acid (**13**) were determined in carmabin A.

A marine cyanobacterium *Oscillatoria nigro-viridis* from Panama area led to the isolation of two linear alkynoic lipopeptides, viridamides A and B with 5-methoxydec-9-ynoic acid (**24**). Viridamide A showed anti-trypanosomal activity with an $IC_{50} = 1.1$ μM, and anti-leishmanial activity with an $IC_{50} = 1.5$ μM [100]. An acetylene-containing lipopeptide, kurahyne with FA (**23**), was isolated from a cyanobacterial assemblage that mostly consisted of *Lyngbya* sp. Kurahyne inhibited the growth of human cancer cells and induced apoptosis in HeLa cells, and it seemed to localize in mitochondria [101].

Cyclic depsipeptides, the cocosamides A and B, containing FA (**15**) and (**25**), respectively, have been detected in the lipophilic extract of a collection of *L. majuscula* from Cocos Lagoon, Guam. Both metabolites showed weak cytotoxicity against MCF7 breast cancer and HT-29 colon cancer cells [102].

The marine benthic cyanobacteria *Oscillatoria nigroviridis* from the Colombian Caribbean Sea produces lipopeptides named almiramides B, D, E, F, H, and G with FA (**12**). Almiramides B and D show a strong activity against human tumor cell lines A549, MDA-MB231, MCF-7, HeLa and PC3 [103]. Pitipeptolides D, E and F with (**15**, FA) showing antimycobacterial cyclodepsipeptides were detected in the marine cyanobacterium *L. majuscula* from Piti Bomb Holes, Guam. Obtained compounds showed weak cytotoxicity against HT-29 colon adenocarcinoma and MCF7 breast cancer cells [86].

Cytotoxic cyclic depsipeptides, hantupeptins A and B, have been derived from the marine cyanobacterium *L. majuscula* from Pulau Hantu Besar, Singapore. Hantupeptin A with FA (**12**) showed strong cytotoxicity against leukemia cells and breast cancer MCF-7 cells (IC$_{50}$ values of 32 and 4.0 μM, respectively), while hantupeptin B with FA (**19**) displayed moderate cytotoxicity against MOLT-4 (leukemia) and MCF-7 cell lines [95,104].

The cyclic depsipeptide guineamide G with FA (**15**) has been extracted from *Lyngbya semiplena* and *L. majuscula*. The isolated lipopeptide exhibited brine shrimp toxicity and showed potent cytotoxicity against a mouse neuroblastoma cell line with an LC$_{50}$ value of 2.7 μM [105,106].

A slightly halophilic myxobacterial strain, SMH-27-4, was isolated from nearshore soil. This slowly-growing myxobacterium produced the novel antibiotic depsipeptides named miuraenamides A and B, which both contain (*R*,*E*) -9-hydroxy-6-methyldec-5-enoic acid (**26**, structure see Figure 8, and Table 4). Miuraenamide A exhibited potent and selective inhibition against a phytopathogenic microorganism, *Phytophthora* sp., And moderate inhibition against some fungi and yeasts. Both metabolites inhibited NADH oxidase at IC$_{50}$ values of 50 μM [107].

**Figure 8.** FA derived from peptide molecules produced by cyanobacteria belonging to the strain *Lyngbya*.

**Table 4.** Predicted biological activity of FA from cyanobacteria of the strain *Lyngbya*.

| No. | Predicted Biological Activity, Pa * |
|:---:|:---:|
| **26** | Antineoplastic (0.855); Lipid metabolism regulator (0.842) Apoptosis agonist (0.803) |
| **27** | Sclerosant (0.871); Acute neurologic disorders treatment (0.870) Antineoplastic (0.714) |
| **28** | Antineoplastic (0.906); Hypolipemic (0.902); Lipid metabolism regulator (0.879) Apoptosis agonist (0.811); Acute neurologic disorders treatment (0.670) Atherosclerosis treatment (0.662); Proliferative diseases treatment (0.562) |
| **29** | Cholesterol antagonist (0.649); Antianginal (0.628) Lipid metabolism regulator (0.532) |
| **30** | Cholesterol antagonist (0.649); Antianginal (0.628) Lipid metabolism regulator (0.532) |
| **31** | Iron antagonist (0.952); Antineoplastic (0.736) Microtubule formation stimulant (0.578) |
| **32** | Neurodegenerative diseases treatment (0.688); Bone diseases treatment (0.528) |
| **33** | Dermatologic (0.667); Anti-psoriatic (0.663) |

* Only activities with Pa > 0.5 are shown.

Cyanobacteria are known to require iron to grow, and they often inhabit iron-restricted habitats and produce several siderophores, including an unusual FA (**27**). Iron starvation triggered the synthesis of β-OH-Asp lipopeptides in cyanobacteria *Rivularia* sp. strain PCC 7116, *Leptolyngbya* sp. strain NIES-3755, and *Rubidibacter lacunae* strain KORDI 51-2 [108].

Unusual (2*E*,6*E*,8*E*)-12-hydroxy-14-methoxy-2,4,9-trimethylpentadeca-2,6,8-trienoic acid (**28**) was detected in cyclic peptide named alotamide A, which is produced by the marine cyanobacterium *Lyngbya bouillonii*. Alotamide A displays an unusual calcium influx activation profile in murine cerebrocortical neurons with an $EC_{50}$ of 4.18 μM [109]. The cyclic peptides apratoxins A–C possess the unprecedented (*E*)-3-((4*S*)-2-((2*S*,3*R*,5*S*)-3,8-dihydroxy-5,9,9-trimethyldecan-2-yl)-4,5-dihydrothiazol-4-yl)-2-methylacrylic acid (**29**) as the polyketide moiety, and apratoxin D with FA (**30**) showed potent in vitro cytotoxicity against H-460 human lung cancer cells with an $IC_{50}$ value of 2.6 nM. Apratoxin A possesses $IC_{50}$ values for in vitro cytotoxicity against human tumor cell lines, ranging from 0.3 to 0.5 nM; however, it was only marginally active in vivo against a colon tumor and ineffective against a mammary tumor. Apratoxins A–D have been isolated from the marine cyanobacteria *Lyngbya majuscula* and *Lyngbya sordida* [110,111].

The nNeuroactive cyclic depsipeptide hoiamide A was originally isolated from a consortium of two different filamentous cyanobacteria identified as *Lyngbya majuscula* and *Phormidium gracile*, and two related peptide metabolites, one a cyclic depsipeptide, hoiamide B, and the other a linear lipopeptide, hoiamide C, were isolated from two different collections of marine cyanobacteria obtained in Papua New Guinea. All the isolated hoiamides A–C contain unusual FA (**31**) [112].

The 3D graph demonstrating the predicted and calculated FA (**31**) activity is shown in Figure 9. A collection of the marine cyanobacterium *Lyngbya bouillonii* from Guam afforded the cytotoxic apratoxin E, which displayed stronger cytotoxicity than its closest analog, semisynthetic E-dehydroapratoxin A, against several cancer cell lines derived from colon, cervix, and bone, ranging from 21 to 72 nM. Both cyclic peptides contained FA (**32**) and (**33**), respectively [113]. A halogenated metabolite, (*S*)-7,7-dichloro-3-hydroxy-2,2-dimethyloctanoic acid (**34**) is incorporated into many lipopeptides that are produced predominantly by the marine cyanobacterium of the genus *Lyngbya*, as well as other cyanobacterial species, *L. bouillonii* and the Fijian marine cyanobacterium *Moorea producens* [114,115]. Thus, lyngbyabellin A, a significantly cytotoxic compound with unusual structural features, was isolated from a Guamanian strain of the marine cyanobacterium *Lyngbya majuscula* [116]. Lyngbyabellin A was shown to be a potent disrupter of the cellular

microfilament network. In addition, lyngbyabellin A showed anticancer activity against HT29, HeLA, KB, LoVo cancer cell lines [114,115]. Antitumor lipopeptides lyngbyabellin B, C, J and 27-deoxylyngbyabellin A have shown activity against HT29, HeLA, and MCF7 cancer cell lines [115,117,118].

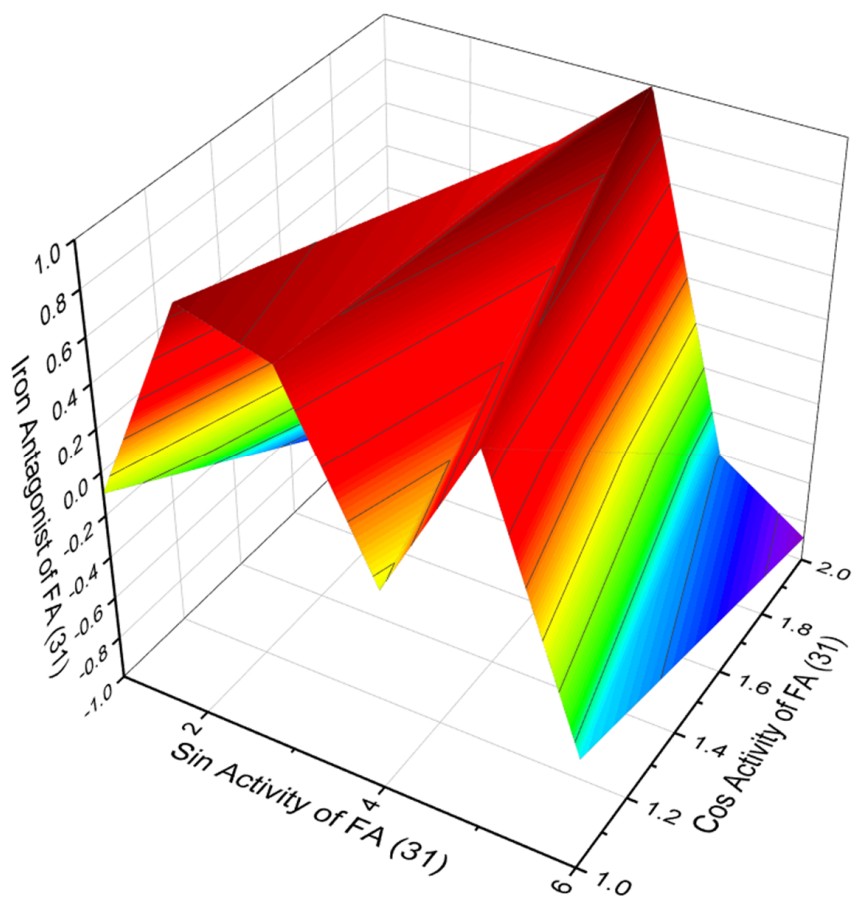

**Figure 9.** 3D graph showing the predicted and calculated Iron antagonist activity of unusual FA (**31**) with the highest degree of confidence being more than 95%. This acid is part of a neuroactive cyclic depsipeptide called hoiamide. These cyclic depsipeptides are produced by two different filamentous cyanobacteria, *Lyngbya majuscula* and *Phormidium gracila*. It is a very rare case that a fatty acid exhibits the surprising property of being an iron antagonist with a high degree of certainty.

A halogenated FA (**34**, see Figure 10) has also been found in the cytotoxic lipopeptide named hectochlorin which was isolated from marine isolates of *Lyngbya majuscula* collected from Hector Bay, Panama [119]. Another chlorine-containing metabolite, (2*S*,3*S*)-7,7-dichloro-3-hydroxy-2-methyl-octanoic acid (**35**) was detected in bioactive lipopeptides such as the lyngbyabellins E, F, G, H, I, K, M, N, O and P, which exhibit antimalarial, anti-cancer, and antifouling activities [117,118,120–122]. The above lyngbyabellins have been isolated from the marine cyanobacteria *Lyngbya majuscula*, *Lyngbya* sp., *L. bouillonii*, *Okeania* sp., *Moorea bouillonii*, and *M. producens*. (2*S*,3*S*,7*R*)-7-Chloro-3-hydroxy-2-methyloctanoic acid (**36**) was isolated from lipopeptides named lyngbyabellins K, L and 7-*epi*-lyngbyabellin L, which showed antitumor activity against H-460 cancer cell lines [123].

The cyclic lipopeptides named antillatoxins, ATx-A (**41**) and ATx-B (**42**) have been isolated from a marine cyanobacteria *Lyngbya majuscula* [124], and both toxins contained (4*S*,5*S*,6*E*,8*E*)-5-hydroxy-4,6,8,10,10-pentamethyl-3-methyleneundeca-6,8-dienoic acid (**37**), and acid (**40**). Antillatoxin B exhibited significant sodium channel-activating (EC$_{50}$ = 1.8 μM) and ichthyotoxic (LC$_{50}$ = 1 μM) properties [125]. It was shown that the natural product antillatoxin B (**42**) is 10 times less active than antillatoxin A (**41**), and synthetic stereoisomers (**43** and

**44**) of the cyclic depsipeptide of antillatoxin A were 20–55 times less active than the natural isomer [125,126].

**Figure 10.** FA derived from lipopeptides and the cyclic depsipeptides produced by *Moorea* and *Lyngbya*.

Both synthetic stereoisomers (**43** and **44**) contain different lipophilic fragments, (4*S*,5*S*,6*E*, 8*E*)-5-hydroxy-4,6,10,10-tetramethyl-3-methyleneundeca-6,8-dienoic acid (**38**) and (4*S*,5*S*),(*E*)-5-hydroxy-4,6,10,10-tetramethyl-3-methyleneundec-6-enoic acid (**39**), respectively, although the cytotoxicity (**44**) for Neuro-2a cells was shown to be 10 times more effective than the cytotoxicity of the molecule (**43**) as shown by Okura and co-workers [127].

The biological activities of FA **34–40** and the cyclic lipopeptides named antillatoxins **41–44** are shown in Table 5. Undoubtedly, (*S*)-7,7-dichloro-3-hydroxy-2,2-dimethyloctanoic acid (**34**) containing two chlorine atoms is of great interest since it demonstrates strong hypolipemic activity with a confidence level of more than 92%. The 3D graph of this acid is shown in Figure 11.

**Table 5.** Predicted biological activity of FA from peptides of *Lyngbya majuscula*.

| No. | Predicted Biological Activity, Pa * |
|:---:|:---:|
| 34 | Hypolipemic (0.921); Anti-psoriatic (0.913); Antidiabetic (0.897); Antineoplastic (0.799); Anti-obesity (0.783); Antihypertriglyceridemic (0.766); Lipid metabolism regulator (0.633) |
| 35 | Hypolipemic (0.910); Antidiabetic (0.902); Antihypertriglyceridemic (0.761) |
| 36 | Sclerosant (0.741); Inflammatory Bowel disease treatment (0.607); Antibacterial (0.588) |
| 37 | Antineoplastic (0.880); Hypolipemic (0.858); Lipid metabolism regulator (0.608) |
| 38 | Antineoplastic (0.822); Hypolipemic (0.801); Lipid metabolism regulator (0.700) |

**Table 5.** *Cont.*

| No. | Predicted Biological Activity, Pa * |
|---|---|
| 39 | Hypolipemic (0.890); Lipid metabolism regulator (0.813); Anti-hypercholesterolemic (0.704) Atherosclerosis treatment (0.657); Cholesterol synthesis inhibitor (0.542) |
| 40 | Hypolipemic (0.858); Antineoplastic (0.810); Lipid metabolism regulator (0.608) |
| 41 | Antineoplastic (0.897); Antifungal (0.787); Apoptosis agonist (0.696) |
| 42 | Antineoplastic (0.869); Antifungal (0.728); Apoptosis agonist (0.650) |
| 43 | Antineoplastic (0.895); Antifungal (0.802); Apoptosis agonist (0.734) |
| 44 | Antineoplastic (0.854); Antifungal (0.801); Antibacterial (0.708); Apoptosis agonist (0.653) |
| A | Hypolipemic (0.910); Antidiabetic (0.902); Antihypertriglyceridemic (0.761) |
| B | Anti-ischemic, cerebral (0.756); Neuroprotector (0.718); Antiviral (Arbovirus) (0.687); Genital warts treatment (0.648); Antineoplastic (liver cancer) (0.582); Antimetastatic (0.540) |
| C | Antiviral (Arbovirus) (0.732); Neuroprotector (0.726); Antineoplastic (liver cancer) (0.633); Acute neurologic disorders treatment (0.568); Antimitotic (0.567) |
| D | Antineoplastic (liver cancer) (0.923); Antineoplastic (0.685); Anti-ischemic, cerebral (0.664) |

* Only activities with Pa > 0.5 are shown.

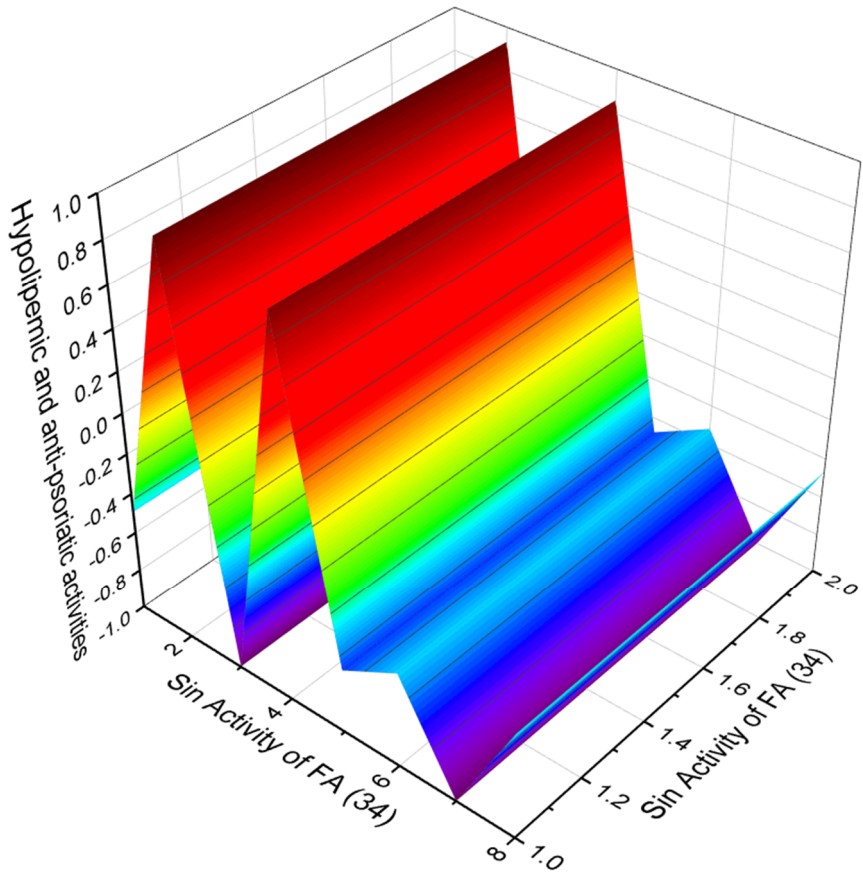

**Figure 11.** 3D Graph showing the predicted and calculated activities with the dominance of hypolipemic and anti-psoriatic properties (these are two highs in the red zone), as seen in (*S*)-7,7-dichloro-3-hydroxy-2,2-dimethyloctanoic acid (**34**). This FA was found in the cytotoxic lipopeptide hectochlorin which was isolated from the cyanobacterium *Lyngbya majuscula*.

It seems very interesting to compare the activities of (4*S*,5*S*,6*E*,8*E*)-5-hydroxy-4,6,8,10,10-pentamethyl-3-methyleneundeca-6,8-dienoic acid (**37**), two cyclic lipopeptides named antillatoxin A (**41**) and B (**42**), and a synthetic analogue (**43**). As shown by the PASS analysis, for all samples the dominant property is moderate antitumor activity with a reliability of about 90%. The 3D graph of this acid (**37**) and cyclic lipopeptides is shown in Figure 12.

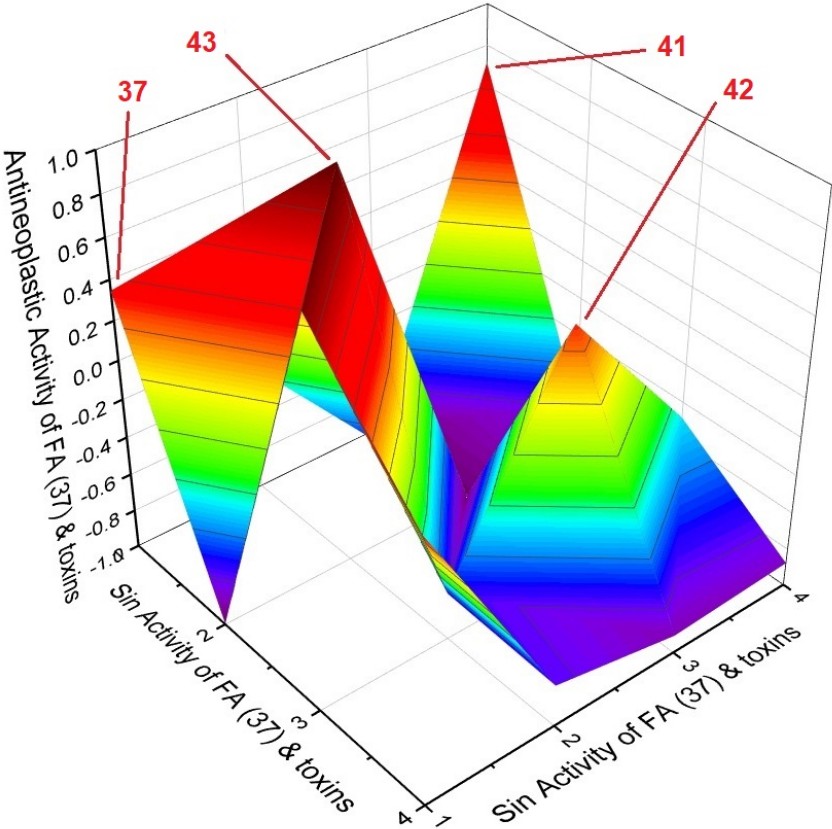

**Figure 12.** 3D Graph showing the predicted and calculated activity with the dominance of antineoplastic properties (these are four highs in the red zone) of (4*S*,5*S*,6*E*,8*E*)-5-hydroxy-4,6,8,10,10-pentamethyl-3-methyleneundeca-6,8-dienoic acid (**37**), two cyclic lipopeptides, antillatoxins A (**41**) and B (**42**), and a synthetic analogue (**43**). The cyclic lipopeptides, ATx-A (**41**) and ATx-B (**42**) have been isolated from a marine cyanobacteria *Lyngbya majuscula*.

Lyngbyabellin N is a bioactive lipopeptide (structure shown in Figure 10) that is produced by the filamentous marine cyanobacterium *Moorea bouillonii*. This metabolite possesses a leucine statin residue and showed strong cytotoxic activity against HCT116 colon cancer cell lines ($IC_{50}$ = 41 nM) and is highly cytotoxic to H-460 human lung cancer cells [117–123]. It was interesting to determine which of the four fragments (A, B, C and D) of lyngbyabellin N contributes to its overall activity. The actuality of these fragments is shown in Table 6. As can be seen from these PASS data, fragment D demonstrates antineoplastic (*liver cancer*) activity with a confidence level of 92.3%.

Neo FA, neo alkanes and their analogs and derivatives are quite rare lipid molecules that are found in marine invertebrates, algae, fungi, plants, and microorganisms, but they are not found in free form [128]. Several neo fatty (carboxylic) acids (**37–41**, **45–48**) have been incorporated into the lipopeptides produced by cyanobacteria. Two cytotoxic peptides with pivalic acid (**45**, activity see in Table 5), named bisebromoamide and norbisebromoamide, have been identified from the marine cyanobacterium *Lyngbya* sp. Bisebromoamide exhibit cytotoxicity against HeLa S3 cells ($IC_{50}$ = 0.04 μg/mL) and inhibit the phosphorylation of extracellular signal-regulated protein kinases in NRK cells, showing potent and selective inhibitory effects on protein kinases [129,130]. The antibiotic bottromycin

B with pivalic acid (**45**), produced by *Streptomyces* sp. strain No. 3668-L2, *Kitasatoa purpurea* strain KA-281, and *Micromonospora chalcea* strain FERM-P 1823 [131–134]. Bottromycin has shown antibacterial activity against six antibiotic-resistant strains of *Staphylococcus aureus*, *Streptococcus pyogenes*, *Micrococcus flavus*, *Bacillus subtilis*, *B. cereus*, *B. megaterium*, *B. anthracis*, *Corynebacterium xerosis*, and *Mycobacterium phlei* in concentrations of 0.03–3 µg/mL.

**Table 6.** Predicted biological activity of FA from peptides of cyanobacteria.

| No. | Predicted Biological Activity, Pa * |
|:---:|:---:|
| **45** | Lipid metabolism regulator (0.928); Anti-hypercholesterolemic (0.738) Preneoplastic conditions treatment (0.722); Antihypoxic (0.711) Hypolipemic (0.676) |
| **46** | Psychostimulant (0.768); Antiviral (0.766) Preneoplastic conditions treatment (0.678) |
| **47** | Psychostimulant (0.731); Antiviral (0.731) Preneoplastic conditions treatment (0.718) |
| **48** | Lipid metabolism regulator (0.885); Hypolipemic (0.772) Anti-hypercholesterolemic (0.735) |
| **49** | Lipid metabolism regulator (0.843); Preneoplastic conditions treatment (0.832) Antimutagenic (0.832); Acute neurologic disorders treatment (0.691) |
| **50** | Fibrinolytic (0.893); Preneoplastic conditions treatment (0.804) |
| **51** | Antidiabetic (0.886); Inflammatory Bowel disease treatment (0.852) |
| **52** | Antidiabetic (0.886); Inflammatory Bowel disease treatment (0.852) |
| **53** | Antidiabetic (0.886); Inflammatory Bowel disease treatment (0.852) |
| **54** | Antidiabetic (0.916); Antineoplastic (0.831); Immunosuppressant (0.681) |
| **55** | Hypolipemic (0.790); Lipid metabolism regulator (0.738) Atherosclerosis treatment (0.679) |
| **56** | Lipid metabolism regulator (0.908); Hypolipemic (0.881) Anti-hypercholesterolemic (0.765) Acute neurologic disorders treatment (0.734); Atherosclerosis treatment (0.722) |
| **57** | Antineoplastic (0.834); Lipid metabolism regulator (0.831) Apoptosis agonist (0.818) Acute neurologic disorders treatment (0.795); Hypolipemic (0.725); Atherosclerosis treatment (0.617) |
| **58** | Antineoplastic (0.703); Lipid metabolism regulator (0.678) Antiviral (Arbovirus) (0.643) |
| **59** | Lipid metabolism regulator (0.886); Hypolipemic (0.823) Anti-hypercholesterolemic (0.742) |
| **60** | Anti-infective (0.877); Lipid metabolism regulator (0.866) Antiviral (Arbovirus) (0.827) |

* Only activities with Pa > 0.5 are shown.

Pyrrolinone-containing lipopeptides named ypaoamide with (*Z*)- and (*E*)- double bonds, herbivore antifeedant metabolites, were isolated from the extract of a mixed cyanobacterial assemblage that was composed of *Schizothrix calcicola* and *Lyngbya majuscula* [135–137]. More than 20 years later, similar pyrrolinone-containing lipopeptides named ypaoamides B and C were isolated from marine cyanobacterium *Okeania* sp. collected in Okinawa. Both ypaoamides B and C stimulated glucose uptake in cultured rat L6 myotubes, and ypaoamide B showed potent activity and activated AMP-activated protein kinase. All four lipopeptides contained the rare 6,6-dimethylheptanoic acid (**46**) [138].

The unique polytheonamides A and B with 5,5-dimethyl-2-oxohexanoic acid (**47**) are highly cytotoxic polypeptides with 48 amino acid residues have been isolated from the marine sponge *Theonella swinhoei* [139].

Janadolide, obtained from cyanobacteria *Okeania* sp. (Janado, Japan) and a cyclic peptide polyketide hybrid possessing a rare *tert*-butyl moiety, (2R,7S,E)-7-hydroxy-2,5,8,8-tetramethylnon-5-enoic acid (**48**), showed potent activity towards *Trypanosoma brucei* (IC$_{50}$ 47 nM) which was superior to the commonly used therapeutic drug suramin. Furthermore, significant selectivity towards the trypanosome parasite was identified, since no in vitro cytotoxicity towards the human cell lines MRC-5, HL60 and HeLa cells was noted at 10 mM [140].

Compounds containing an aromatic ring in a molecule are widespread in nature. They are found in marine invertebrates, algae, fungi, microorganisms, and other living organisms [141–153]. Acyclic lipopeptides named hoshinoamides A and B with 4-(4-hydroxyphenyl)-butanoic acid (**49**) have been isolated from the marine cyanobacterium *Caldora penicillata*. Both compounds inhibited the in vitro growth of the malarial parasite *Plasmodium falciparum* (IC$_{50}$ = 0.5 and 1.0 μM, respectively) [154]. Cytotoxic depsipeptides, anaenamides A and B with 2-methoxy-6-pentylbenzoic acid (**50**), were discovered from a green filamentous cyanobacterium *Hormoscilla* sp. from Guam [155].

A linear lipopeptide aldehyde with 2-hydroxy-3-(4-hydroxyphenyl)-propanoic acid (**51**) was detected in a hydrophilic extract of the *Nostoc* sp. [156]. Two trypsin inhibitors called nostosin A and B with (S)-2-hydroxy-4-(4-hydroxyphenyl)-butanoic acid (**52**) and (R)-2-hydroxy-4-(4-hydroxyphenyl)-butanoic acid (**53**), respectively, were isolated from a hydrophilic extract of *Nostoc* sp. strain FSN, which was collected from a paddy field in the Golestan Province, Iran. Nostosins A and B exhibited IC$_{50}$ values of 0.35 and 55 μM against porcine trypsin, respectively, suggesting that the argininal aldehyde group plays a crucial role in the efficient inhibition of trypsin [157].

Nannocystin A, a lipopeptide with epoxy-containing FA (**54**), was isolated from a myxobacterium *Nannocystis* sp. [158,159]. The isolated compound has a strong antifungal effect against *C. albicans* and displays potent cell proliferation inhibitive properties by inducing apoptosis early in tested cell lines. Besides, nannocystin A has shown antiproliferative properties against 472 cancer cell lines in the nanomolar concentration range (IC$_{50}$ values ranging from 0.5 μM to 5 nM).

A cytotoxic and linear peptide was isolated from the marine cyanobacterium *Geitlerinema* sp. The structure of mitsoamide contains an unusual polyketide unit (3,7-dimethoxy-5-methyl-nonanedioic acid, **55**) incorporated into a homolysine residue, and possesses a highly unusual piperidine amino moiety. This peptide showed antitumor activity against NCI-H460 human lung tumor cells, IC$_{50}$ 460 nM [160].

A lipopeptide, minnamide A with unique (3S,5S,7R,9S,11R,13S,15R)-3,7,11,15-tetrahydroxy-5,9,13-trimethyloctadecanoic acid (**56**) from the marine cyanobacterium *Okeania hirsute* showed growth-inhibitory activity toward HeLa cells with an IC$_{50}$ value of 0.17 μM, and rapidly induced cell death at a concentration of 2 μM [161].

Rare (2E,4E,10E)-15-hydroxy-7-methoxy-2-methylhexadeca-2,4,10-trienoic acid (**57**) was incorporated into cyclic lipopeptide named malevamide E which was found in extracts of the marine cyanobacterium *Symploca laeteviridis* (see Figure 13) and showed store-operated Ca$^{2+}$ entry in thapsigargin-treated human embryonic kidney (HEK) cells with a dose-dependent inhibition (2–45 μM) [162].

The tropical marine cyanobacterium, *Moorea bouillonii* from New Britain, Papua New Guinea, yielded a cytotoxic cyclic depsipeptide, bouillonamide [163]. The obtained metabolite exhibited mild toxicity with an IC$_{50}$ of 6.0 μM against the neuron 2a mouse neuroblastoma cells. In addition, the cyclopeptide contained two unique polyketide-derived moieties, namely a 2-methyl-6-methylamino-hex-5-enoic acid (**58**) and 3-methyl-5-hydroxy-heptanoic acid (**59**).

The antifungal glycosylated lipopeptide, hassallidin A, was isolated from an epilithic cyanobacterium *Tolypothrix basionym* collected in Bellano, Italy. The isolated lipopep-

tide with 2,3-dihydroxytetradecanoic acid (**60**, see Figure 14) exhibited antifungal activity against *Aspergillus fumigatus* and *Candida albicans* [164]. The 3D graph demonstrating the predicted and calculated activity of FA (**51, 52, 53** and **54**) is shown in Figure 15.

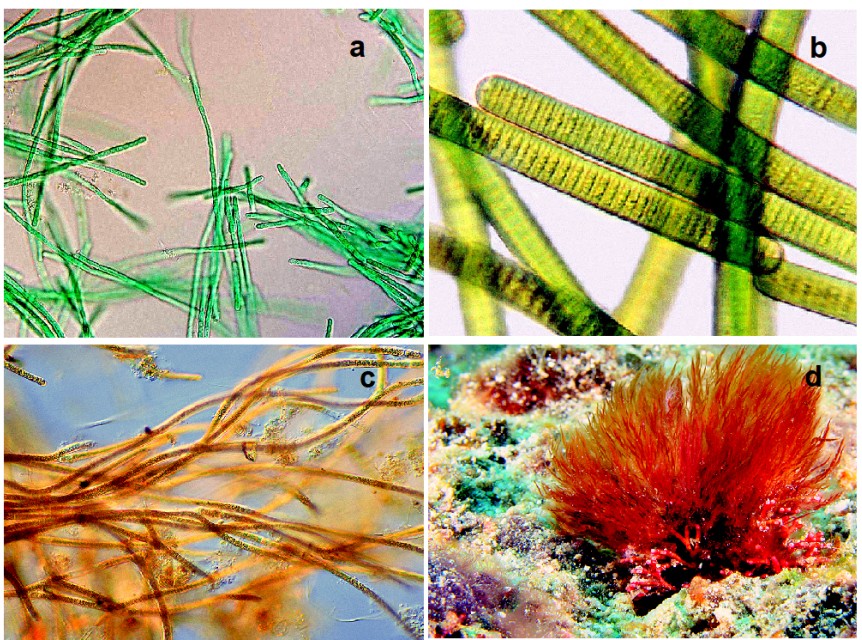

**Figure 13.** Cyanobacteria: (**a**), *Leptolyngbya* sp.; (**b**), *Oscillatoria* sp.; (**c**), *Tolypothrix* sp.; (**d**), *Symploca hydnoides*, found in marine and freshwater environments, typically develop on lake, pond, or marine sediments. All these cyanobacteria produce various bioactive metabolites, including linear and cyclic peptides and lipopeptides containing unusual FA.

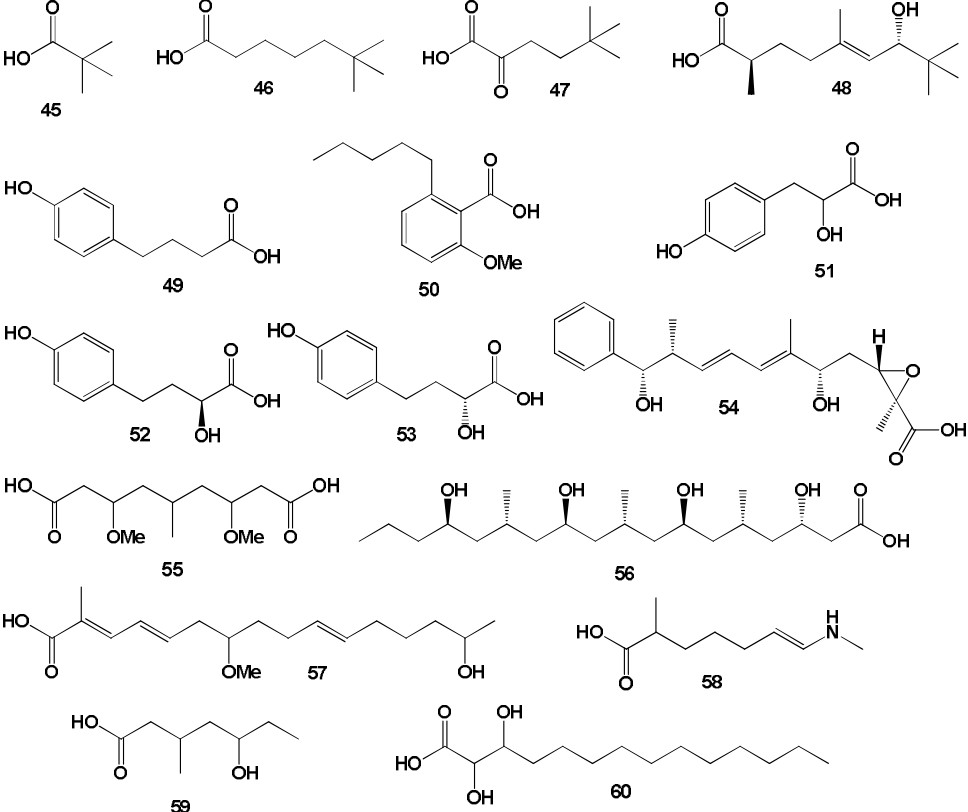

**Figure 14.** Neo, aromatic, and other FA derived from peptides produced by cyanobacteria.

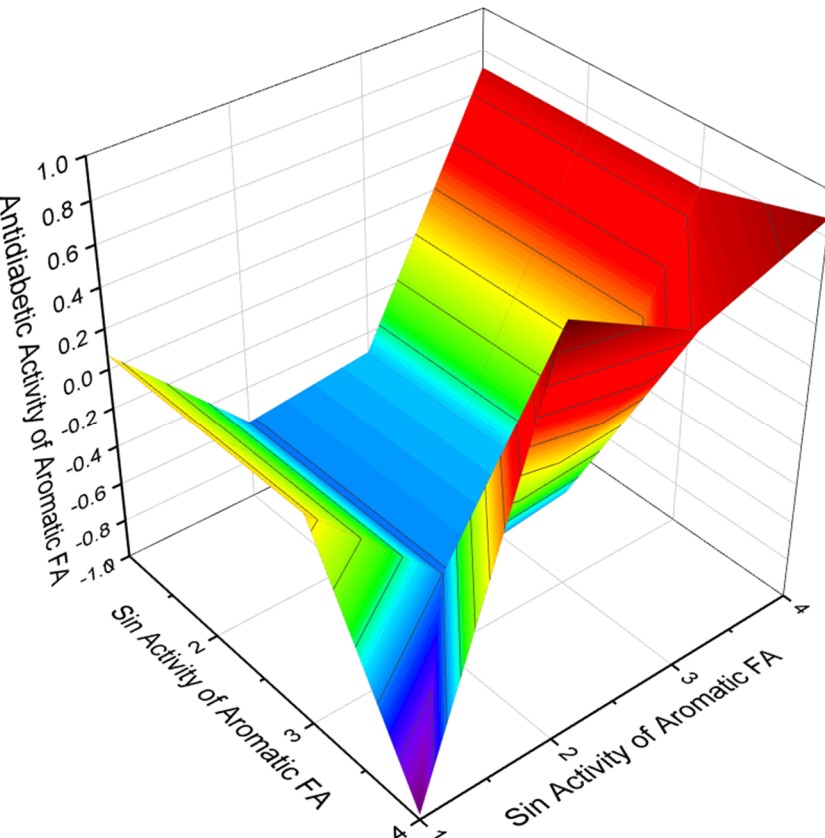

**Figure 15.** 3D graph showing the predicted and calculated with dominance of the antidiabetic activity of 2-hydroxy-3-(4-hydroxyphenyl)-propanoic acid (**51**), (**52**), (**53**), and (2*R*,3*S*)-3-((2*S*,3*E*,5*E*,7*R*,8*S*)-2,8-dihydroxy-3,7-dimethyl-8-phenylocta-3,5-dien-1-yl)-2-methyloxirane-2-carboxylic acid (**54**). These FA were derived from lipopeptides produced by the cyanobacterium *Nostoc* sp. and myxobacterium *Nannocystis* sp.

Yu and co-workers [165] isolated nine linear lipopeptides of the named microcolins E–M from the marine cyanobacteria *Moorea produns*, which showed significant cytotoxic activity against lung carcinoma, and they all contained (2*S*,4*S*)-2,4-dimethyloctanoic acid (**61**, for structure see Figure 16, and activity see in Table 7). Two linear lipopeptides, gageostatins B and C belonging to the heptapeptides were obtained from the fermentation broth of a marine-derived bacterium *Bacillus subtilis*. The isolated lipopeptides contain (3*R*)-3-hydroxy-9,11-dimethyltridecanoic (**62**) and (3*S*,*E*)-3-hydroxy-9,11-dimethyl-tridec-4-enoic (**63**) acids, respectively. The gageostatins exhibited good antifungal activities with MIC values of 4–32 µg/mL when tested against pathogenic fungi (*Rhizoctonia solani*, *Bacillus cinerea* and *Colletotrichum acutatum*) and both compounds shown moderate antibacterial activity against bacteria (*Bacillus subtilis*, *Staphylococcus aeureus*, *Salmonella typhi* and *Pseudomonas aeruginosa*) with MIC values of 8–64 µg/mL. Furthermore, gageostatins displayed cytotoxicity against six human cancer cell lines with GI$_{50}$ values of 4.6–19.6 µg/mL [166].

An antimalarial lipopeptide, ikoamide with (3*S*,5*R*)-3,5-dimethoxyoctanoic acid (**64**), was isolated from an *Okeania* sp. marine cyanobacterium. Ikoamide showed strong antimalarial activity with an IC$_{50}$ value of 0.14 µM without cytotoxicity against human cancer cell lines at 10 µM [167]. A malyngamide with (*E*)-7-hydroxytetradec-4-enoic acid (**65**) was isolated from the marine cyanobacterium *Moorea producens* collected in Hawaii. The compound showed cytotoxicity against the L1210 cell line at an IC$_{50}$ value of 2.9 mM and lethal toxicity against the shrimp *Palaemon paucidens* at an LD$_{100}$ value of 33.3 mg/kg [168].

The marine bacterium *Saccharomonospora* sp. CNQ-490 produced the chlorinated lipopeptide taromycin A, and taromycin B was detected in ethyl acetate extracts of *S. coelicolor* M1146-M1 cultures [169,170]. Both taromycins A and B display potent activity against

methicillin-resistant *Staphylococcus aureus* and vancomycin-resistant *Enterococcus faecium* clinical isolates. Both lipopeptides contain a rare (2*E*,4*E*)-octa-2,4-dienoic (**66**) and (2*E*,4*E*)-6-methylocta-2,4-dienoic (**67**) acids, respectively.

Cyclic lipopeptides named bananamides D-G were detected in the crude extract of *Pseudomonas* sp. COW3. Both bananamides D and G contain (Z)-3-hydroxydodec-4-enoic acid (**68**). COW3 displayed antagonistic activity and mycophagy against *Pythium myriotylum*, while it mainly showed mycophagy on *Pyricularia oryzae*. Purified bananamides D-G inhibited the growth of *P. myriotylum* and *P. oryzae* and caused hyphal distortion [171].

The cyclic lipopeptide named gageopeptin A with (3*S*)-3-hydroxy-12,14-dimethylhexa-decanoic acid (**69**) was obtained from the ethyl acetate extract of the fermentation broth of a marine-derived strain *Bacillus* sp. 109GGC020 and it exhibited moderate antibacterial and good antifungal activities [172], and two linear lipopeptides, gageopeptides C and D with (3*S*)-3-hydroxy-8,10-dimethyldodecanoic (**70**) and (3*R*)-3-hydroxy-9,11-dimethyltridecanoic (**71**) acids, were discovered from a marine *Bacillus subtilis* strain 109GGC020 [173].

**Figure 16.** FA derived from peptides produced by marine bacteria and cyanobacteria.

**Table 7.** Predicted biological activity of FA derived from cyanobacterial peptides.

| No. | Predicted Biological Activity, Pa * |
|---|---|
| **61** | Hypolipemic (0.861); Antineoplastic (0.854); Lipid metabolism regulator (0.849) |
| **62** | Lipid metabolism regulator (0.934); Hypolipemic (0.903); Sclerosant (0.869) <br> Acute neurologic disorders treatment (0.845); Anti-hypercholesterolemic (0.831) <br> Atherosclerosis treatment (0.705); Cholesterol synthesis inhibitor (0.519) |
| **63** | Lipid metabolism regulator (0.959); Anti-hypercholesterolemic (0.900); Hypolipemic (0.895) <br> Acute neurologic disorders treatment (0.893); Atherosclerosis treatment (0.696) <br> Multiple sclerosis treatment (0.548); Antibacterial (0.537); Cholesterol synthesis inhibitor (0.526) |
| **64** | Hypolipemic (0.878); Acute neurologic disorders treatment (0.780) <br> Lipid metabolism regulator (0.771); Atherosclerosis treatment (0.675) |
| **65** | Lipid metabolism regulator (0.941); Hypolipemic (0.829); Anti-hypercholesterolemic (0.792) <br> Acute neurologic disorder treatment (0.697); Atherosclerosis treatment (0.647) |
| **66** | Antiviral (Arbovirus) (0.947); Lipid metabolism regulator (0.884); Antimutagenic (0.793) <br> Antiviral (Picornavirus) (0.782); Hypolipemic (0.724); Anti-hypercholesterolemic (0.695) |
| **67** | Hypolipemic (0.835); Lipid metabolism regulator (0.830); Antiviral (Arbovirus) (0.821) <br> Antiviral (Picornavirus) (0.761); Preneoplastic conditions treatment (0.722) |
| **68** | Lipid metabolism regulator (0.953); Antiviral (Arbovirus) (0.885) <br> Hypolipemic (0.884); Anti-hypercholesterolemic (0.856); Leukopoiesis stimulant (0.826); Atherosclerosis treatment (0.692) |
| **69** | Lipid metabolism regulator (0.934); Hypolipemic (0.903); Anti-hypercholesterolemic (0.831) <br> Atherosclerosis treatment (0.705); Cholesterol synthesis inhibitor (0.519) |
| **70** | Lipid metabolism regulator (0.934); Hypolipemic (0.903); Anti-hypercholesterolemic (0.831) <br> Atherosclerosis treatment (0.705); Cholesterol synthesis inhibitor (0.519) |
| **71** | Lipid metabolism regulator (0.934); Hypolipemic (0.903); Anti-hypercholesterolemic (0.831) <br> Atherosclerosis treatment (0.705); Cholesterol synthesis inhibitor (0.519) |
| **72** | Vasoprotector (0.890); Lipid metabolism regulator (0.884); Antiviral (Arbovirus) (0.824) |
| **73** | Antipsoriatic (0.957); Antineoplastic (0.886); Antiviral (Arbovirus) (0.796) <br> Lipid metabolism regulator (0.792); Alzheimer's disease treatment (0.613) |
| **74** | Lipid metabolism regulator (0.929); Antiviral (Arbovirus) (0.891); Antimutagenic (0.866) <br> Anti-hypercholesterolemic (0.786); Hypolipemic (0.721); Atherosclerosis treatment (0.629) |

* Only activities with Pa > 0.5 are shown.

A lipopeptide antibiotic, stalobacin I was discovered from a culture broth of an unidentified Gram-negative bacterium. Stalobacin I had a unique chemical architecture composed of an upper and a lower half peptide sequence, which were linked via a hemiaminal methylene moiety. The sequence of one contained an unusual amino acid, carnosadine, 3,4-dihydroxyariginine, 3-hydroxy-isoleucine, and 3-hydroxyaspartic acid, and a novel cyclopropyl FA, (*E*)-2-hydroxy-4-((1*R*,2*R*)-2-((*Z*)-tridec-6-en-1-yl)-cyclopropyl)-but-3-enoic acid (**72**). This compound showed antibacterial activity against a broad range of drug-resistant Gram-positive bacteria and was much stronger than those of "last resort" antibiotics such as vancomycin, linezolid, and telavancin (MIC 0.004–0.016 μg/mL) [174].

*Lyngbya majuscula* from Papua New Guinea yielded the guineamides B and C with (*S*)-3-hydroxy-2,2-dimethylhexanoic acid (**73**), which possess moderate cytotoxicity to a mouse neuroblastoma cell line with IC$_{50}$ values of 15 and 16 µM, respectively [40]. The same acid (**73**) was detected in a cyclic depsipeptide palmyramide A that was isolated from a *Lyngbya majuscula–Centroceras* sp. association [175]. Pure palmyramide A showed sodium channel-blocking activity in neuro-2a cells and cytotoxic activity in H-460 human lung carcinoma cells. The 3D graph demonstrating the predicted and calculated activity of FA (**66**) is shown in Figure 17, and the 3D graph on Figure 18 shows the activity of FA (**62**), (**63**), (**68**), and (**70**).

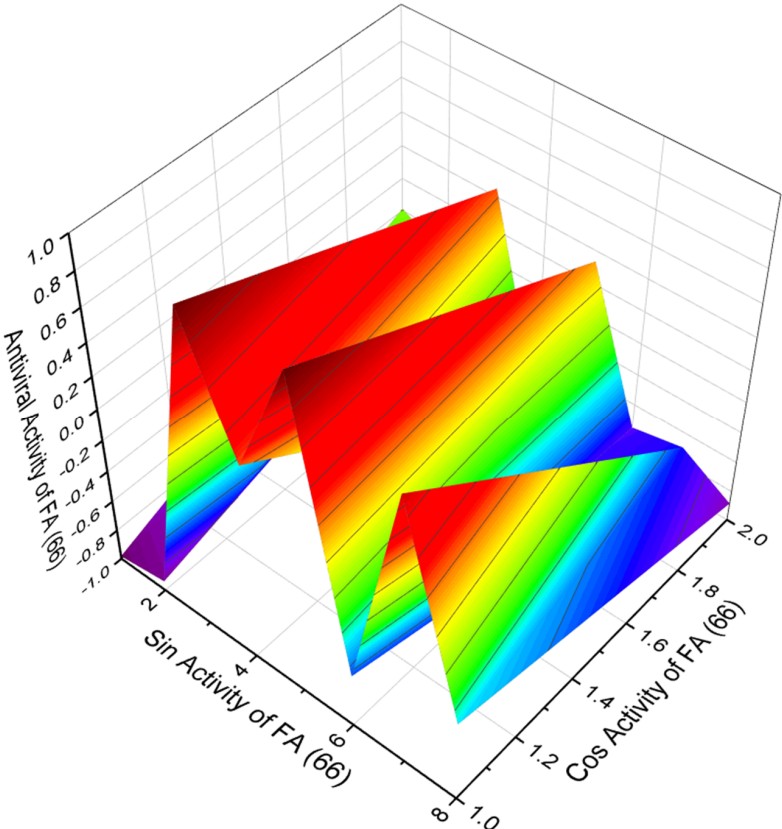

**Figure 17.** 3D Graph showing the predicted and calculated with dominance of antiviral activity of (2*E*,4*E*)-octa-2,4-dienoic (**66**) acid. This FA was found in lipopeptides, which are produced by the marine bacteria *Saccharomonospora* sp. and *Saccharomonospora coelicolor* and can be used as a strong antiviral agent.

A lipopeptide with (*Z*)-hexadec-9-enoic acid (**74**) named pseudoalteropeptide A was isolated from the marine bacterium *Pseudoalteromonas piscicida* SWA4_PA4. It showed moderate iron-chelating activity as well as cytotoxic activity against Jurkat human T lymphocyte cells [176].

Bioactive compounds of marine cyanobacteria from a Jamaican collection of *Lyngbya majuscula* led to the isolation of jamaicamides A–C. Two jamaicamides A and B are highly functionalized lipopeptides containing an alkynyl, vinyl chloride, α-methoxy eneone system, and pyrrolinone ring fragments [176A]. Three halogenated FA, (4*E*,9*E*)-14-bromo-9-(chloromethylene)-6-methyltetradec-4-en-13-ynoic acid (**75**, for structure see Figure 19, and activity see in Table 8), (4*E*,9*E*)-9-(chloromethylene)-6-methyltetradec-4-en-13-ynoic acid (**76**) and (4*E*,9*E*)-9-(chloromethylene)-6-methyltetradeca-4,13-dienoic acid (**77**) were found in the neurotoxins, jamaicamides A–C, respectively. Halogenated linear lipopeptides called vatiamides F and E have been found in the cyanobacterium *Moorea producens*. Both lipopeptides contained same FA (**75**) and (**76**), respectively [177].

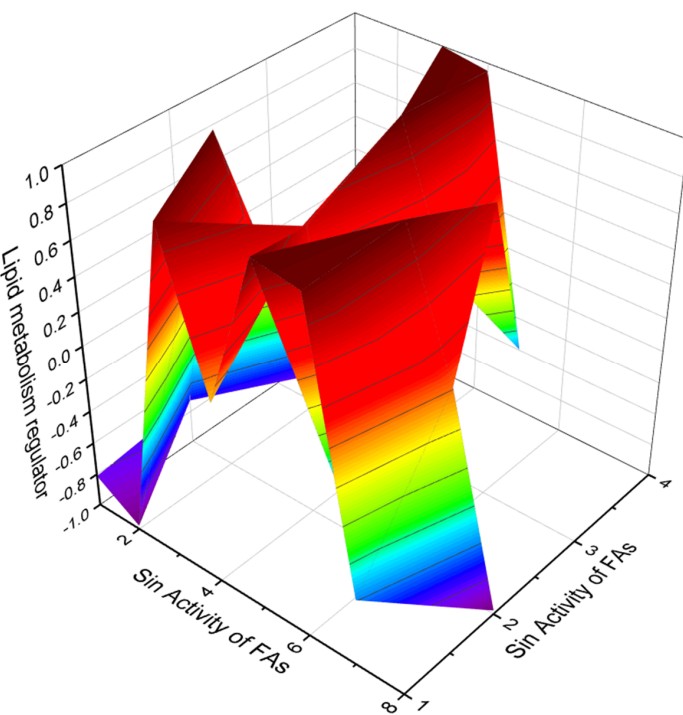

**Figure 18.** 3D graph showing the predicted and calculated activity, with the dominance of lipid metabolism-regulator properties, of FA (**62**), (**63**), (**68**), and (**70**), with the highest degree of confidence being more than 93%. These FA derived from lipopeptides produced by the marine cyanobacterium *Moorea produns*, bacteria *Pseudomonas* sp. and *Bacillus* sp.

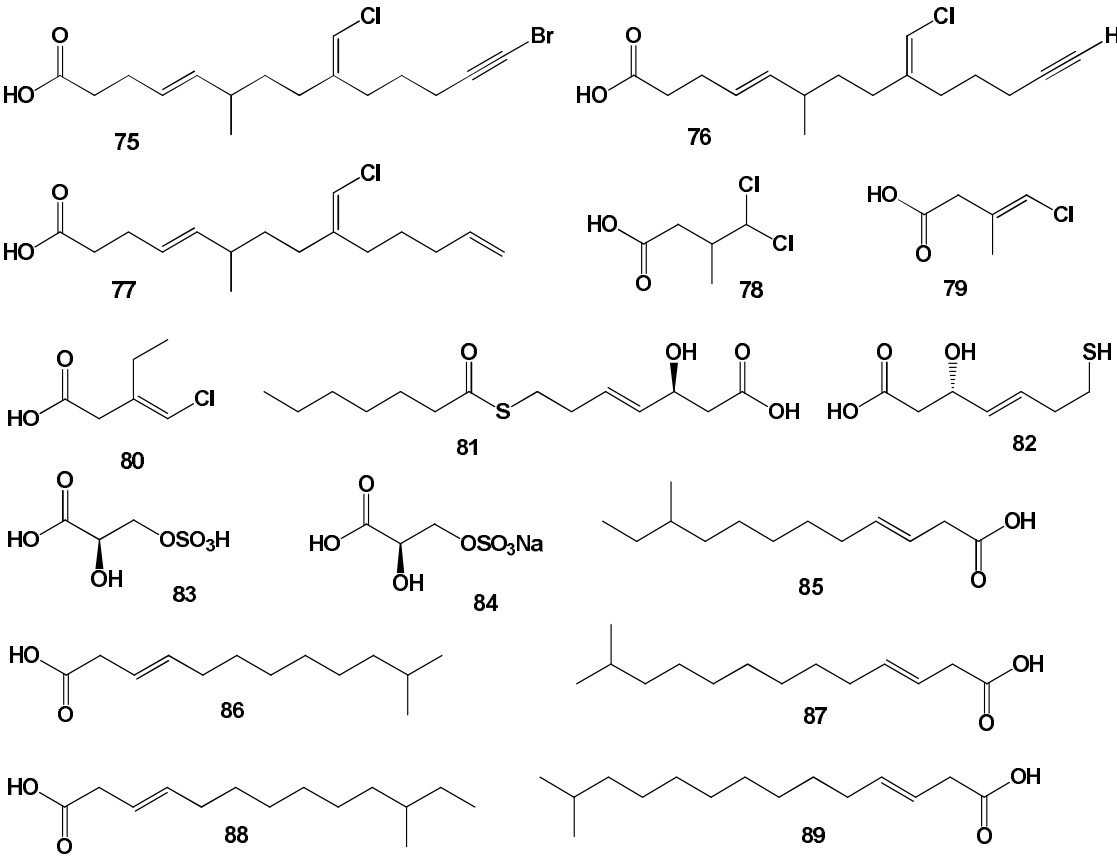

**Figure 19.** Chloro-containing, sulfur-containing and other FA derived from lipopeptides.

**Table 8.** Predicted biological activity of FA from cyanopeptides.

| No. | Predicted Biological Activity, Pa * |
|:---:|:---:|
| 75 | Antifungal (0.791); Antineoplastic (0.744) Lipid metabolism regulator (0.680) |
| 76 | Antieczematic (0.869); Anesthetic general (0.722) Neuroprotector (0.714) |
| 77 | Antieczematic (0.900); Lipid metabolism regulator (0.859) Antifungal (0.756) |
| 78 | Antineoplastic (0.960); Preneoplastic conditions treatment (0.661) Antiviral (Arbovirus) (0.618); Antiviral (Picornavirus) (0.520) |
| 79 | Cystic fibrosis treatment (0.861); Antiviral (Arbovirus) (0.717); Anesthetic general (0.702) |
| 80 | Cystic fibrosis treatment (0.850); Anesthetic general (0.733) Antiviral (Arbovirus) (0.687) |
| 81 | Anti-hypercholesterolemic (0.902); Apoptosis agonist (0.779); Antineoplastic (0.779) |
| 82 | Lipid metabolism regulator (0.888); Angiogenesis stimulant (0.869); Expectorant (0.715) |
| 83 | Acute neurologic disorders treatment (0.870); Anti-inflammatory (0.776) |
| 84 | Acute neurologic disorders treatment (0.870); Anti-inflammatory (0.776) |
| 85 | Lipid metabolism regulator (0.930); Anti-hypercholesterolemic (0.842) Hypolipemic (0.830) |
| 86 | Lipid metabolism regulator (0.848); Anti-hypercholesterolemic (0.770) Hypolipemic (0.760) |
| 87 | Lipid metabolism regulator (0.848); Anti-hypercholesterolemic (0.770) |
| 88 | Lipid metabolism regulator (0.930); Anti-hypercholesterolemic (0.842) Hypolipemic (0.830) |
| 89 | Lipid metabolism regulator (0.848); Anti-hypercholesterolemic (0.770) Hypolipemic (0.760) |

* Only activities with Pa > 0.5 are shown.

The cyanobacteria genus *Lyngbya* is an amazing source of chlorine-containing metabolites, and *Lyngbya majuscula* from Grenada has identified depsipeptides named itralamides A and B, which contain 4,4-dichloro-3-methylbutanoic acid (**78**) [178]. The 3D graph demonstrating the predicted and calculated activity of FA (**78**) is shown in Figure 20.

Cyanobacterial field collections from American Samoa and Palmyra Atoll yielded three cyclic peptides called tutuilamides A–C. Tutuilamides A–C show potent elastase inhibitory activity together with moderate potency in H-460 lung cancer cell cytotoxicity assays. The tutuilamides A and B contain (*E*)-4-chloro-3-methylbut-3-enoic acid (**79**) and (*E*)-3-(chloromethylene)-pentanoic acid (**80**) and were detected in tutuilamide [179].

The Floridian marine cyanobacterium, *Symploca* sp., produces a cytotoxic depsipeptide named largazole and contains a 4-methyl-thiazoline unit (**81**) and an unusual a 3-hydroxy-7-mercaptohept-4-enoic acid (**82**). This thioether-functional depsipeptide is a potent inhibitor of the growth of transformed human mammary epithelial cells (MDA-MB-231) and is less susceptible to non-transformed mouse mammary epithelial cells. In addition, largazole showed exceptional antiproliferative activity against transformed U2OS fibroblast osteosarcoma cells [180].

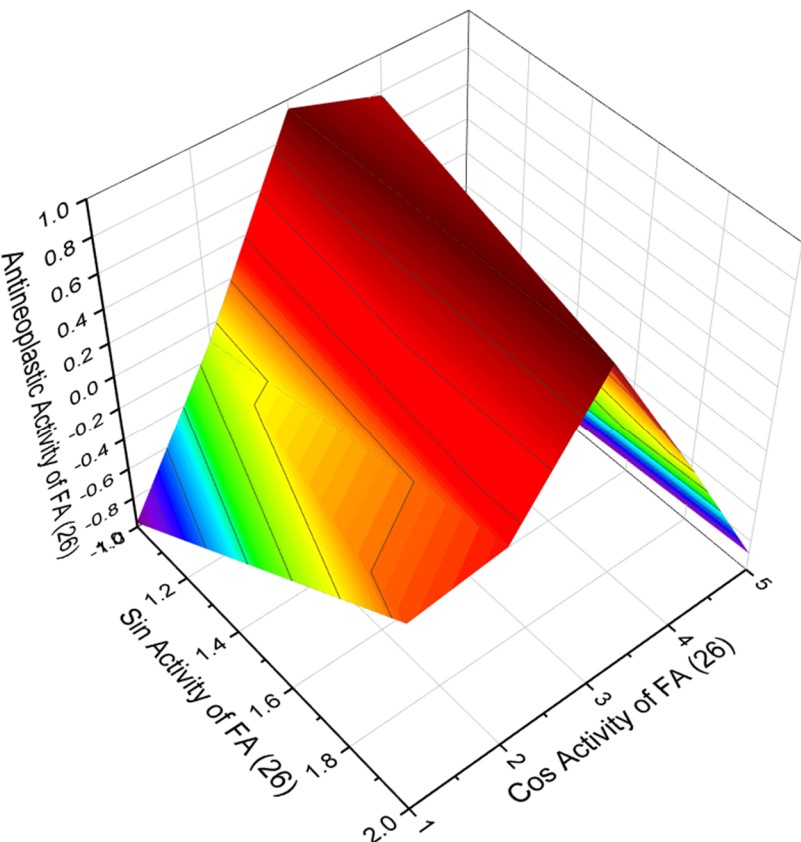

**Figure 20.** 3D graph showing the predicted and calculated antineoplastic activity of 4,4-dichloro-3-methylbutanoic acid (**78**) with the highest degree of confidence being more than 96%. This halogenated acid was detected in depsipeptides of the Eastern Caribbean collection of *Lyngbya majuscula*.

Three lipopetides called lyngbyastatins 4–6 have been identified from the marine cyanobacterium *Lyngbya confervoides* from the Florida Atlantic coast and South Florida. Lyngbyastatin 4 shows potent and selective inhibitory effects on elastase as well as chymotrypsin in vitro over other serine proteases with $IC_{50}$ values of 0.03 and 0.30 μM, respectively. Rare (*R*)-2-hydroxy-3-(sulfooxy)-propanoic acid (**83**) was detected in lyngbyastatin 4 and sodium (*R*)-2-carboxy-2-hydroxyethyl sulfate (**84**) was found in lyngbyastatin 6 [181,182].

A lipopeptide antibiotic called amphomycin A with (*E*)-10-methyldodec-3-enoic acid (**85**) was first isolated from extracts of the bacterium *Streptococcus canis* demonstrating antibacterial activity against Gram-positive pathogens [183,184]. A lipopeptide antibiotic called tsushimycin which was isolated from *Bacillus subtilis* contains three different FA in varying proportions: (*E*)-11-methyldodec-3-enoic (**86**), (*E*)-12-methyltridec-3-enoic (**87**) and (*E*)-11-methyltridec-3-enoic (**88**) acids [185].

Several lipopeptide antibiotics, friulimicins A, B, C, D, and the acidic lipopeptides of the amphomycin type that were also present in the culture fluid, compounds A-1437 A, B, E, and G, were isolated from cultures of *Actinoplanes friuliensis* HAG01 0964 after fermentation in different nutrient media. All eight lipopeptides possess an identical peptide macrocycle as their central element, linked via a diaminobutyric acid N-terminal either to an acylated asparagine residue or, in the case of the amphomycin series, to an acylated aspartic acid residue. Friulimicin A and A-1437 A contains cis-3-*iso*-13:1 acid (**87**), friulimicin B and A-1437 B contains *cis*-3-isotetradecenoic acid (**88**), friulimicin C and A-1437 E contains fatty acid (**89**), and friulimicin D and A-1437 G contains *cis*-3-anteisopentadecenoic acid (**90**, for structure see Figure 21, and activity in Table 9) [186,187].

**Figure 21.** Unsaturated and other FA derived from peptides.

Four cyclolipopeptides, glycinocins A to D, were isolated from the fermentation broth of an unidentified terrestrial Actinomycete species. The glycinocin antibiotics are structurally related to amphomycin that was originally reported as a linear lipopeptide with a C-terminal diketopiperazine moiety [188]. All isolated glycinocins contain rare double-bond FA in the second position. So, glycinocins A and D contain (*E*)-14-methylpentadec-2-enoic acid (**91**), B-(*E*)-15-methylhexadec-2-enoic acid (**92**) and C-(*E*)-13-methyltetradec-2-enoic acid (**93**), respectively.

A series of different lipopeptides containing the same FA fragment (**94–100**) were isolated from marine cyanobacteria living in different regions of the world's oceans. Thus, *Lyngbya majuscula* from Papua New Guinea led to the isolation of two lipopeptides, aurilides B and C. Both compounds with (2*E*,5*S*,6*S*,7*S*,8*E*)-5,7-dihydroxy-2,6,8-trimethylundeca-2,8-dienoic acid (**94**) showed in vitro cytotoxicity toward NCI-H460 human lung tumor and the neuro-2a mouse neuroblastoma cell lines, with LC$_{50}$ values between 0.01 and 0.13 μM, and aurilide B exhibited a high level of cytotoxicity against leukemia, renal, and prostate cancer cell lines [189].

**Table 9.** Predicted biological activity of FA from cyanobacterial peptides.

| No. | Predicted Biological Activity, Pa * |
|---|---|
| 90 | Lipid metabolism regulator (0.930); Anti-hypercholesterolemic (0.842) Hypolipemic (0.830) Atherosclerosis treatment (0.635); Multiple sclerosis treatment (0.507) |
| 91 | Lipid metabolism regulator (0.848); Anti-hypercholesterolemic (0.770); Hypolipemic (0.760) Acute neurologic disorders treatment (0.696); Atherosclerosis treatment (0.551) |
| 92 | Lipid metabolism regulator (0.848); Anti-hypercholesterolemic (0.770); Hypolipemic (0.760) Acute neurologic disorders treatment (0.696); Atherosclerosis treatment (0.551) |
| 93 | Lipid metabolism regulator (0.930); Anti-hypercholesterolemic (0.842); Hypolipemic (0.830) Acute neurologic disorders treatment (0.690); Atherosclerosis treatment (0.635) |
| 94 | Antineoplastic (0.859); Apoptosis agonist (0.765); Antimitotic (0.689) |
| 95 | Antineoplastic (0.778); Apoptosis agonist (0.724); Antiviral (Arbovirus) (0.695) Preneoplastic conditions treatment (0.551) |
| 96 | Antineoplastic (0.800); Apoptosis agonist (0.794); Lipid metabolism regulator (0.780) Antifungal (0.750); Preneoplastic conditions treatment (0.622); Spasmolytic (0.540) |
| 97 | Antineoplastic (0.877); Apoptosis agonist (0.818); Hypolipemic (0.809); Antifungal (0.775) Anti-inflammatory (0.748); Preneoplastic conditions treatment (0.540) |
| 98 | Antineoplastic (0.881); Apoptosis agonist (0.814); Hypolipemic (0.810) Antifungal (0.767); Anti-inflammatory (0.746); Lipid metabolism regulator (0.706) |
| 99 | Antineoplastic (0.871); Hypolipemic (0.815); Antifungal (0.775); Apoptosis agonist (0.764) Anti-inflammatory (0.729); Lipid metabolism regulator (0.643) |
| 100 | Antineoplastic (0.812); Anti-inflammatory (0.763); Immunosuppressant (0.712) Antifungal (0.695); Apoptosis agonist (0.691); Hypolipemic (0.667) |

* Only activities with Pa > 0.5 are shown.

Cyclic depsipeptides, lagunamides A and B were isolated from the marine cyanobacterium *L. majuscula* obtained from Pulau Hantu Besar (Singapore). Both lagunamides displayed significant antimalarial properties against *Plasmodium falciparum*, with IC$_{50}$ values of 0.2 and 0.9 μM, respectively. Lagunamides A and B contained (5*S*,6*S*,7*R*,8*R*,*E*)-5,7-dihydroxy-2,6,8-trimethyldec-2-enoic (**98**, for predicted activity see Figure 22) and (2*E*,5*S*,6*S*,7*S*,8*E*)-5,7-dihydroxy-2,6,8-trimethyldeca-2,8-dienoic (**99**) acids and possessed potent cytotoxic activity against P388 murine leukemia cell lines, with IC$_{50}$ values of 6.4 and 20.5 nM, respectively [190]. The same cyanobacterium from Singapore produced a cytotoxic cyclodepsipeptide named lagunamide C, which displayed potent cytotoxic activity against a panel of cancer cell lines, such as P388, A549, PC3, HCT8, and SK-OV3 cell lines, with IC$_{50}$ values ranging from 2.1 to 24.4 nM. This compound with (5*S*,6*R*,8*R*,9*S*,*E*)-5,8-dihydroxy-2,6,9-trimethylundec-2-enoic acid (**95**) also displayed significant antimalarial activity with an IC$_{50}$ value of 0.29 μM when tested against *Plasmodium falciparum*. In addition, lagunamide C exhibited weak anti-swarming activity when tested at 100 ppm against the Gram-negative bacterial strain, *Pseudomonas aeruginosa* PA01 [191].

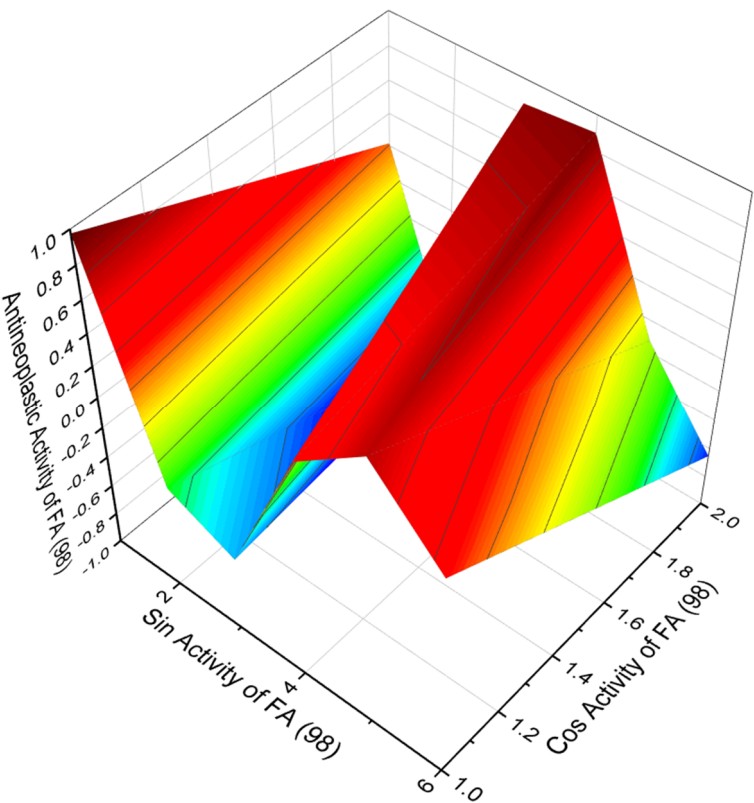

**Figure 22.** 3D graph showing the predicted and calculated antineoplastic activity of (5*S*,6*S*,7*R*,8*R*,*E*)-5,7-dihydroxy-2,6,8-trimethyldec-2-enoic acid (**98**) as a fragment of lagunamides A and B. These cyclic depsipeptides were isolated from the marine cyanobacterium *Lyngbya majuscula* obtained from Pulau Hantu Besar (Singapore).

Cytotoxic macrocyclic depsipeptides, lagunamide D and D, were discovered from a mixture containing marine cyanobacteria *Dichothrix* sp., *Lyngbya* sp. and *Rivularia* sp. from Florida. Both depsipeptides contain (5*S*,6*R*,7*R*,*E*)-5,7-dihydroxy-2,6-dimethylundec-2-enoic acid (**96**) [192]. In addition, lagunamide A and D exhibited antiproliferative activity even in the low-nanomolar range against A549 human lung adenocarcinoma cells with an IC$_{50}$ value of 6.7 and 7.1 nM, respectively.

Okinawan marine cyanobacterium *Okeania* sp. led to the isolation of the cyclodepsipeptide named odoamide. Notably, this compound containing (5*S*,6*S*,7*R*,8*S*,*E*)-5,7-dihydroxy-2,6,8-trimethylundec-2-enoic acid (**97**) showed potent cytotoxicity against HeLa S3 human cervical cancer cells with an IC$_{50}$ value of 26.3 nM [193].

The extract of a species of *Lyngbya* sp. from Palau has yielded the cyclodepsipeptide palauamide with (5*S*,6*R*,7*R*,*E*)-5,7-dihydroxy-2,6-dimethyldodec-2-en-11-ynoic acid (**100**), which had an IC$_{50}$ value of 13 nM against the KB tumor cell line [93].

The Baltic Sea cyanobacterium *Anabaena cylindrica* Bio33, cultivated in the laboratory, has provided the antifungal lipopeptides balticidins A–D. An unusual, chlorinated FA (**101**, for structure see Figure 23, and activity see in Table 10) was detected in balticidins A and B, and dechlorinated acid (**102**) was found in balticidins C and D. Antifungal activity with these compounds is also observed against *Candida albicans*, *C. krusei*, *Aspergillus fumigatus*, *Microsporum gypseum*, *Mucor* sp., and *Microsporum canis* [194,195]. Antibiotic lipopeptides from *Pseudomonads* (see Figure 24), brabantamides A–C, were isolated from plant-associated *Pseudomonas* sp. SH-C52. Brabantamides A–C displayed moderate to high in vitro activities against Gram-positive bacterial pathogens. Brabantamide B contains unsaturated fatty acid (**103**), and brabantamide A and C contains saturated FA (**104**) and (**105**), respectively [196].

**Figure 23.** Bioactive glycosidic FA derived from cyanobacterial peptides.

**Table 10.** Predicted biological activity of FA derived from cyanopeptides.

| No. | Predicted Biological Activity, Pa * |
|-----|-------------------------------------|
| 101 | Anti-hypercholesterolemic (0.936); Antifungal (0.848); Antibacterial (0.798) Hypolipemic (0.723); Anti-infective (0.718); Atherosclerosis treatment (0.522) |
| 102 | Anti-infective (0.936); Anti-hypercholesterolemic (0.928); Antifungal (0.853) Antioxidant (0.848); Antineoplastic (0.833); Antidiabetic (0.807) Antibacterial (0.774); Hypolipemic (0.774) Acute neurologic disorders treatment (0.702) Proliferative diseases treatment (0.690); Atherosclerosis treatment (0.603) |
| 103 | Vasoprotector (0.970); Anti-infective (0.966); Hemostatic (0.950) Neuroprotector (0.942) Anti-hypercholesterolemic (0.915); Lipid metabolism regulator (0.903) Acute neurologic disorders treatment (0.845); Hypolipemic (0.770) Atherosclerosis treatment (0.624); DNA synthesis inhibitor (0.584) Dementia treatment (0.582) |
| 104 | Anti-infective (0.961); Vasoprotector (0.960); Neuroprotector (0.905) Anti-hypercholesterolemic (0.875); Antithrombotic (0.850) Hypolipemic (0.757); Atherosclerosis treatment (0.627); DNA synthesis inhibitor (0.590) |

**Table 10.** *Cont.*

| No. | Predicted Biological Activity, Pa * |
|-----|-------------------------------------|
| **105** | Anti-infective (0.961); Vasoprotector (0.960); Neuroprotector (0.905); Sclerosant (0.891) Anti-hypercholesterolemic (0.875); Antithrombotic (0.850); Lipid metabolism regulator (0.807) Hypolipemic (0.757); Atherosclerosis treatment (0.627); DNA synthesis inhibitor (0.590) |
| **106** | Anti-infective (0.966); Vasoprotector (0.953); Anti-hypercholesterolemic (0.890) Antihypoxic (0.881); Lipid metabolism regulator (0.853); Antineoplastic (0.835) Hypolipemic (0.751); Acute neurologic disorders treatment (0.684); DNA synthesis inhibitor (0.566) |
| **107** | Vasodilator (0.868); Anti-infective (0.865); Antifungal (0.829); Antineoplastic (0.809) Anti-hypercholesterolemic (0.609); Hypolipemic (0.566); Antimycobacterial (0.545) |

* Only activities with Pa > 0.5 are shown.

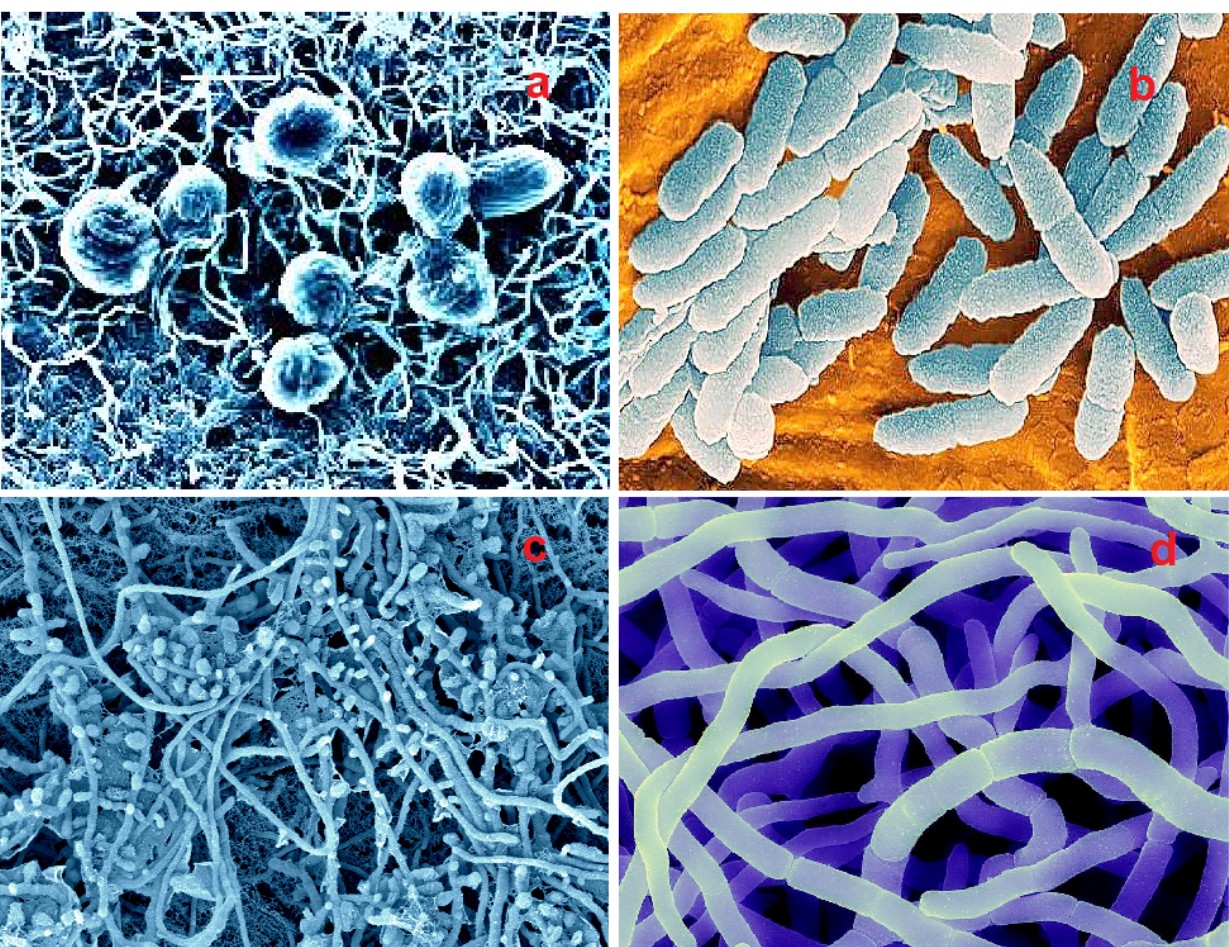

**Figure 24.** Photos of different types of bacteria: (**a**), *Actiniplanus* sp.; (**b**), *Pseudomonas aeruginosa*; (**c**), *Saccharomonospora viridis*; (**d**), *Streptomyces* sp., which inhabit various environments and produce lipopeptides with rare and unusual FA.

The cyanobacterium *Hassallia* sp. produces a family of bioactive compounds which exhibits a broad spectrum of antifungal activities. One of the bioactive glycolipopeptides is hassallidin B, which contains glycosidic FA (**106**, activity of this glycosidic FA sees in Figure 25) [197], and hassallidin D, which contains glycosidic FA (**107**) [198].

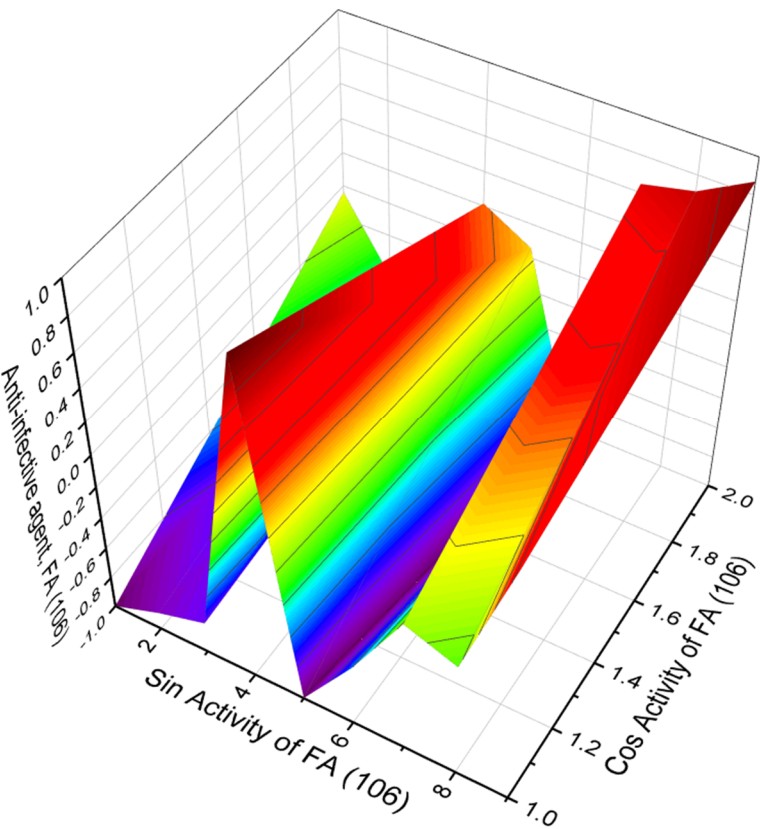

**Figure 25.** 3D graph showing the predicted and calculated activity of an anti-infective agent of glycosidic FA (**106**). This acid is a fragment of the glycolipopeptide hassallidin B, which was isolated from the cyanobacterium *Hassallia* sp.

Characterizing the fatty acids of lipopeptides of bacteria and cyanobacteria, some conclusions can be drawn. So, these lipopeptides are characterized by fatty acids containing an aromatic ring, as measured by cryptophycins. Acetylene-containing FA are a hallmark of bacterial lipopeptides produced by *Nostoc* species. For bacterial lipopeptides, fatty acids with oxazole and thiazole rings, as well as sulfur- and chlorine-containing FA, are an interesting feature.

## 3. Linear and Cyclic Peptides Derived from Seaweeds and Invertebrates

Many algae and invertebrate species have long been used as human food, animal fodder and sources of valuable substances, including lipids. Marine seaweeds and invertebrates are rich in unusual lipids, steroids, triterpenoids, phospholipids, glycolipids, and polyunsaturated FA and are of potential value as sources of essential FA, important in the nutrition of humans and animals [199–218]. In addition, proteins of marine algae and invertebrates, which are natural reservoirs of bioactive peptides, are of great interest [219–225].

### 3.1. Fatty Acids Derived from Seaweed Lipopeptides

Marine and freshwater algae are a phylogenetically heterogeneous group of aquatic plants that belong to three main taxonomic groups: green (Chlorophyta), brown (Phaeophyta), and red (Rhodophyta) [226]. Since ancient times, seaweeds have been of great practical interest since they contain bioactive elements such as iodine, bromine or chlorine, and metabolites, steroids, carotenoids, fatty acids, lipopeptides, alkaloids, and other organic molecules that have antimicrobial, antiviral, anti-inflammatory and immunotropic properties [227–231].

Two cyclic lipopeptides, mebamamide A and B, containing FA (**108**, for structure see Figure 26, and activity see in Table 11) and (3*R*,8*S*)-3,8-dihydroxy-9-methyldecanoic acid (**109**), respectively, were isolated from the green alga *Derbesia marina* [232].

**Figure 26.** Graphical display of the chemical structure of the green alga *Derbesia marina lipopeptide* and the free FA formed by hydrolysis of the amide bond.

The sacoglossan mollusc *Elysia rufescens* is known to use the green algae *Bryopsis pennata* and *B. plumosa* as its main diet [233–237]. Analysis of lipid extracts from molluscs and algae showed that they contain biologically active cyclic depsipeptides, (kahalalides A–F, iso-KF, 5-OHKF, K, O–S, R′, S′, W, and Y) and five linear depsipeptides (kahalalides G, H, J, V, and X), which exhibit cytotoxic, antitumour, antimicrobial, antileishmanial and immunosuppressive activities [233].

The (*R*)-2-methylbutanoic acid (**110**, for structures see Figure 27, and biological activity is shown in Table 11) was found in kahalalide A and 5-methylhexanoic acid (**111**) was included in the structure of the kahalalides B, F, G, O, R2 and S2. 3-Hydroxy-9-methyldecanoic acid (**112**) was present in kahalalides E, H, J, K, and Y. (*R*)-4-Methylhexanoic acid (**113**), 5-hydroxy-5-methylhexanoic acid (**114**), (*S*)-2-hydroxy-9-methyldecanoic acid (**115**), 5-hydroxy-7-methyloctanoic acid (**116**), and (*R*)-3-hydroxy-7-methyloctanoic acid (**117**) were incorporated into the lipopeptides *iso*-kahalalide F, 5-OH-kahalalide F, kahalalide P and Q, kahalalide R1 and S1, and kahalalide V, respectively. As specimens, Figure 28 shows the green alga *Bryopsis pennata*, *B. plumosa* and the sacoglossan mollusc *Elysia rufescens*.

According to the PASS data, FA (**115**) showed properties as a cerebral anti-ischemic agent with a confidence level of more than 94%. This FA is found in cyclic depsipeptides, kahalalides P and Q, and a 3D graph of its predicted and calculated cerebral anti-ischemic activity is shown in Figure 29.

**Table 11.** Predicted biological activity of FA from peptides of seaweeds and molluscs.

| No. | Predicted Biological Activity, Pa * |
|:---:|:---:|
| **108** | Hypolipemic (0.858); Lipid metabolism regulator (0.835); Anti-hypercholesterolemic (0.634) Antifungal (0.648); Atherosclerosis treatment (0.603) |
| **109** | Hypolipemic (0.858); Lipid metabolism regulator (0.835); Anti-hypercholesterolemic (0.634) Antifungal (0.648); Atherosclerosis treatment (0.603) |
| **110** | Preneoplastic conditions treatment (0.779); Hypolipemic (0.775); Anesthetic general (0.772) Lipid metabolism regulator (0.768); Acute neurologic disorders treatment (0.694) |
| **111** | Preneoplastic conditions treatment (0.805); Acute neurologic disorders treatment (0.723) Antiviral (Arbovirus) (0.716); Anti-inflammatory (0.650); Antiviral (Picornavirus) (0.649) |
| **112** | Lipid metabolism regulator (0.890); Hypolipemic (0.870); Anti-hypercholesterolemic (0.802) Atherosclerosis treatment (0.692); Cholesterol synthesis inhibitor (0.511) |
| **113** | Lipid metabolism regulator (0.895); Preneoplastic conditions treatment (0.778) Anti-hypercholesterolemic (0.777); Hypolipemic (0.758); Atherosclerosis treatment (0.683) |
| **114** | Mucositis treatment (0.886); Anesthetic general (0.852); Lipid metabolism regulator (0.842) Autoimmune disorders treatment (0.798); Transplant rejection treatment (0.795) |
| **115** | Anti-ischemic, cerebral (0.943); Acute neurologic disorders treatment (0.797) Anticonvulsant (0.702); Anti-hypercholesterolemic (0.642); Antihypertensive (0.627) |
| **116** | Lipid metabolism regulator (0.822); Vasodilator, peripheral (0.803); Vasoprotector (0.793) Hypolipemic (0.757); Anti-hypercholesterolemic (0.677); Atherosclerosis treatment (0.647) |
| **117** | Lipid metabolism regulator (0.890); Hypolipemic (0.870); Anti-hypercholesterolemic (0.802) Atherosclerosis treatment (0.692); Cholesterol synthesis inhibitor (0.511) |

* Only activities with Pa > 0.5 are shown.

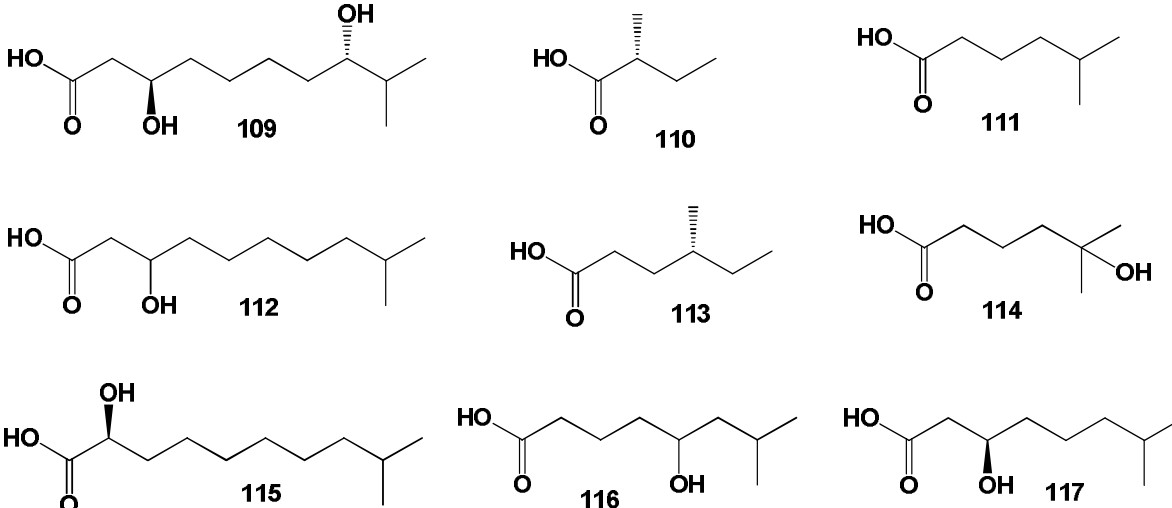

**Figure 27.** FA incorporated into lipopeptides derived from lipid extracts of the green algae *Bryopsis pennata, B. plumosa* and the sacoglossan mollusc *Elysia rufescens*.

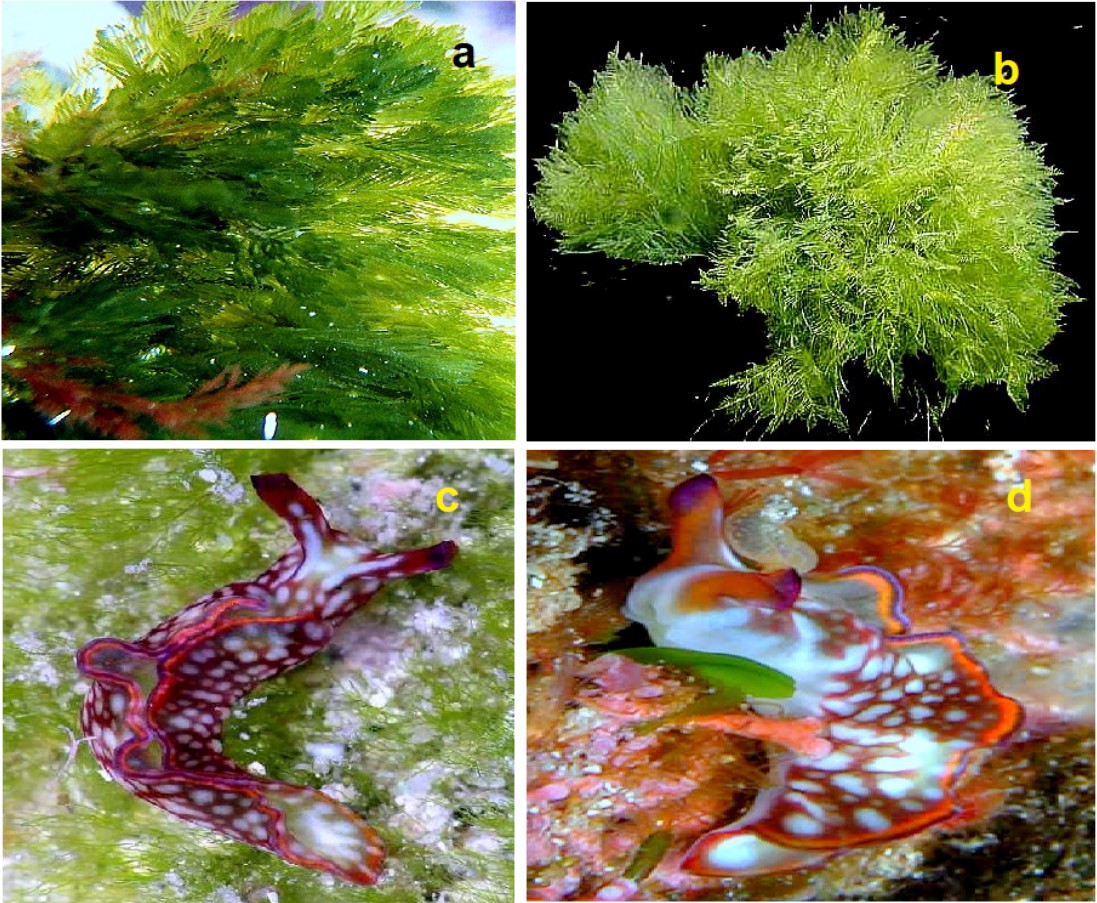

**Figure 28.** The green algae *Bryopsis plumosa* (**a**) and *B. pennata* (**b**) are the staple food for the sea slug, *Elysia rufescens* (**c,d**). This mollusk is similar to nudibranch, but is not classified in this order of gastropods, but belongs instead to a closely related clade, Sacoglossa. These molluscs synthesize a class of cyclic depsipeptides called kahalalides.

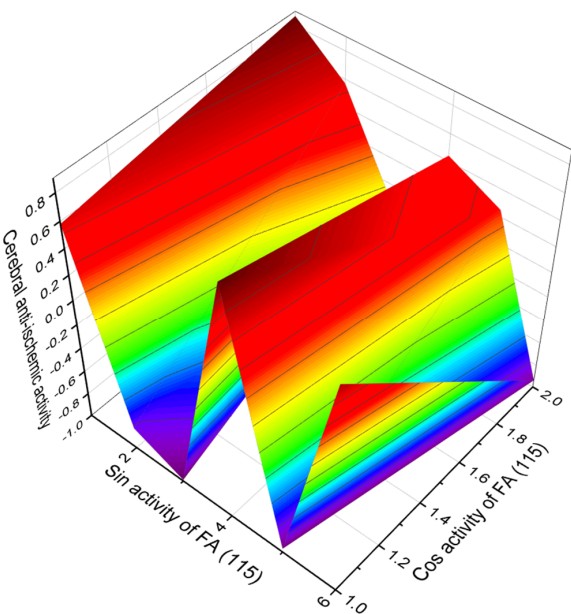

**Figure 29.** 3D Graph showing the predicted and calculated cerebral anti-ischemic activity of FA (**115**). This acid is a fragment of cyclic depsipeptides, kahalalides P and Q.

### 3.2. Fatty Acids Incorporated into Lipopeptides of Marine Sponges

Marine and freshwater sponges (class Demospongiae) are known to be home to many symbiotic microorganisms, including fungal endophytes, bacteria, and some unicellular organisms. Sponges, including their symbiotic microorganisms, synthesize many secondary metabolites such as steroids, terpenoids, carotenoids, halogenated and unusual fatty acids, alkaloids, and of course cyclic and linear lipopeptides [1,4,6,17,23,58,60,61,65,66,128,201–215].

The marine sponges of the genus Theonella synthesize a wide variety of lipopeptides, and in the sponge *Theonella* aff. *mirabilis*, a pentapeptide was found that contained a rare (2*R*,3*R*)-aziridine-2,3-dicarboxylic acid (**118**, see Figure 30) [238]. The isolated lipopeptide inhibits the proteolytic activity of trypsin-like serine proteases, papain-like cysteine proteases, and pepsin-like aspartyl proteases [239]. Previously, this FA (**118**) was found and isolated from the ascomycete *Streptomyces* sp. MD 398-A1 [240]. A similar lipopeptide was isolated from the Red Sea sponge *Theonella swinhoei* (order Lithistida, see Figure 31) and is a potent inhibitor of cathepsin B, protease, and HIV [241]. The aziridine-containing compounds are powerful immuno-modulatory and anticancer agents and are of practical interest to pharmacologists [242,243].

**Miraziridine A, a pentapeptide**

**118** (2R,3R)-aziridine-2,3-dicarboxylic acid

**Figure 30.** Graphical display of the chemical structure of *lipopeptide* isolated from the marine sponge *Theonella* aff. *mirabilis* and the free FA formed by hydrolysis of the amide bond.

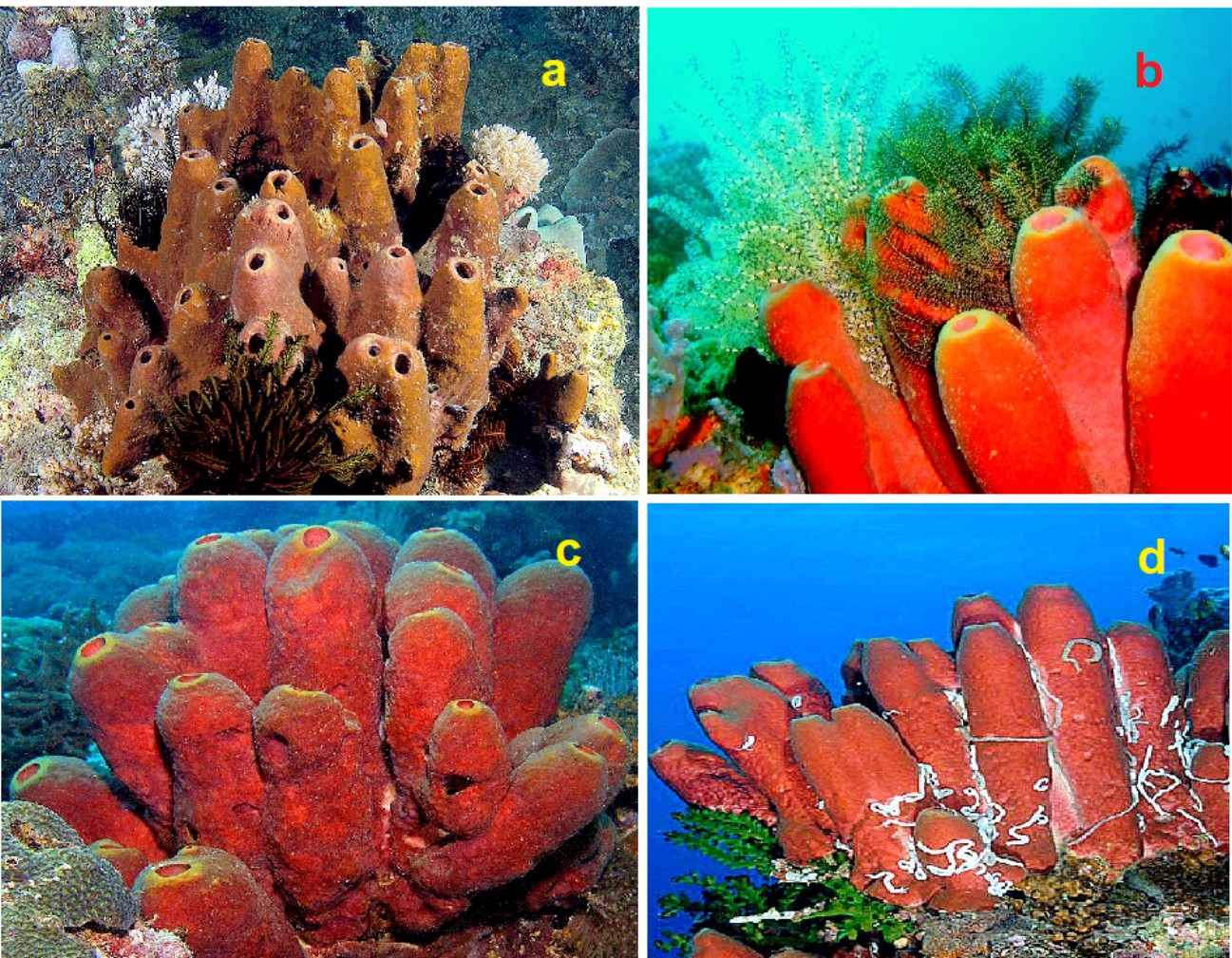

**Figure 31.** Samples of marine sponges: (**a**), *Theonella cylindrica;* (**b**), *T. swinhoei;* (**c**), *T. swinhoei;* (**d**), *T. swinhoei.* It is known that sea sponges from the genus Theonella are home to many associated bacteria that occupy up to 40% of their body volume. *Entotheonella* sp. (Tectomicrobia) is a filamentous symbiont that produces almost all known biologically active compounds derived from the sponge *Theonella swinhoei.*

### 3.2.1. Saturated, Methyl-Branched, and Unsaturated Fatty Acids

We have previously mentioned that neo fatty (carboxylic) acids have been isolated from cyanobacteria, microalgae, and some marine invertebrates [128]. Highly cytotoxic polypeptides named polytheonamides A and B are found in extracts of the marine sponge *Theonella swinhoei* [244–247]. Both polypeptides are quite unusual in that one peptide molecule contains nine amino acids with *tert*-butyl units, and both *linear* polypeptides contain a rare neo-FA, 5,5-dimethyl-2-oxohexanoic acid (**119**, for structure see Figure 32, and activity see in Table 12).

Linear peptides named yakuamides A and B were found and isolated from extracts of the Japanese sponge *Ceratopsia* sp. and they showed activity against P388 murine leukemia cells [245] and both contained 2,2,4,6-tetramethyl-3-oxoheptanoic acid (**120**).

**Figure 32.** Branched, saturated, neo-, and unsaturated FA isolated from sponge lipopeptides.

**Table 12.** Predicted biological activity of FA from peptides of marine sponges.

| No. | Predicted Biological Activity, Pa * |
|---|---|
| 118 | Antineoplastic (0.883); Lipid metabolism regulator (0.836); Anti-inflammatory (0.845) Apoptosis agonist (0.847); Acute neurologic disorders treatment (0.795); Antifungal (0.793) |
| 119 | Phobic disorders treatment (0.859); Psychostimulant (0.731); Antiviral (0.731) Acute neurologic disorders treatment (0.586); Neuroprotector (0.574) |
| 120 | Antiarthritic (0.805); Preneoplastic conditions treatment (0.730); Sclerosant (0.726) Acute neurologic disorders treatment (0.696); Anti-inflammatory (0.641) |
| 121 | Antineoplastic (0.813); Antiviral (Arbovirus) (0.748); Lipid metabolism regulator (0.693) Cytoprotectant (0.668); Antiviral (Picornavirus) (0.585); Hypolipemic (0.575) |
| 122 | Lipid metabolism regulator (0.880); Antineoplastic (0.863); Hypolipemic (0.816) Anti-hypercholesterolemic (0.672); Atherosclerosis treatment (0.590) |
| 123 | Lipid metabolism regulator (0.924); Antineoplastic (0.873); Hypolipemic (0.839) Anti-hypercholesterolemic (0.642); Atherosclerosis treatment (0.592) |

**Table 12.** *Cont.*

| No. | Predicted Biological Activity, Pa * |
|---|---|
| 124 | Hypolipemic (0.911); Lipid metabolism regulator (0.829); Anti-inflammatory (0.765) Anti-hypercholesterolemic (0.718); Acute neurologic disorders treatment (0.715) |
| 125 | Lipid metabolism regulator (0.929); Hypolipemic (0.908) Anti-hypercholesterolemic (0.825); Atherosclerosis treatment (0.680) |
| 126 | Antineoplastic (0.788); Hypolipemic (0.754); Acute neurologic disorders treatment (0.687) |
| 127 | Dermatologic (0.909); Anti-psoriatic (0.888); Anti-eczematic (0.856); Antifungal (0.605) |
| 128 | Anti-psoriatic (0.862); Antineoplastic (0.846); Antifungal (0.625); Anti-eczematic (0.601) |
| 129 | Hypolipemic (0.880); Antineoplastic (0.821); Antifungal (0.707); Antiviral (Arbovirus) (0.654) |
| 130 | Hypolipemic (0.880); Antineoplastic (0.821); Antifungal (0.707); Antibacterial (0.555) |
| 131 | Hypolipemic (0.880); Antineoplastic (0.821); Antifungal (0.707); Antibacterial (0.555) |
| 132 | Antifungal (0.688); Antiprotozoal (Plasmodium) (0.570); Antibacterial (0.514) |
| 133 | Antineoplastic (0.876); Antifungal (0.771); Lipid metabolism regulator (0.763); Hypolipemic (0.707) |

* Only activities with Pa > 0.5 are shown.

The marine sponge *Poecillastra* sp. (Bahamas) yielded the potently cytotoxic poecillastrins A–C, which are related to the chondropsin D, and the closely related cytotoxic poecillastrin D was isolated from *Jaspis serpentina* (Oshimashinsone, Japan) [248]. All the above mentioned lipopeptides contain (*E*)-7-hydroxy-4,4,6,8-tetramethyl-5-oxonon-2-enoic acid (**121**).

Rare cyclic lipodepsipeptides, lipodiscamides A and C, were found in extracts of the marine sponge *Discodermia kiiensis*. Lipodiscamides A and C contain (3*S*,5*R*,6*E*,8*E*,11*Z*)-3-hydroxy-5-methoxy-2,2,15-trimethylhexadeca-6,8,11-trienoic acid (**122**), and lipodiscamide B contains FA (**123**) [249].

Two "head-to-side-chain" depsiundecapeptides named stellatolide A and B were present in lipid extracts from the marine sponge *Ecionemia acervus*. Both compounds showed strong antiproliferative activity against three human cancer cell lines (Lung-NSCLC A549, Colon HT-29 and Breast MDA-MB-231). (3*S*,6*S*,*Z*)-3-Hydroxy-6,8-dimethylnon-4-enoic acid (**124**) was isolated from stellatolide A, and (3*S*,6*S*,*Z*)-3-hydroxy-6-methylnon-4- enoic acid (**125**) was found in stellatolide B [250].

HIV-inhibitory cyclic depsipeptides known as neamphamide A–C were isolated from Papua New Guinea in the marine sponge *Neamphius huxleyi*. All lipopeptides contain 2*R*,3*R*,4*R*)-3-hydroxy-2,4,6-trimethylheptanoic acid (**126**) [251].

Cyclic depsipeptides named halipeptin A and B were found in extracts from the marine sponge *Haliclona* sp. (see Figure 33). (3*R*,4*R*,7*S*)-3,7-Dihydroxy-2,2,4-trimethyldecanoic acid (**127**) was present in halipeptin B and C, and (3*R*,4*R*,7*S*)-3-hydroxy-7-methoxy-2,2,4-trimethyldecanoic acid (**128**) was found in halipeptin A and D [252].

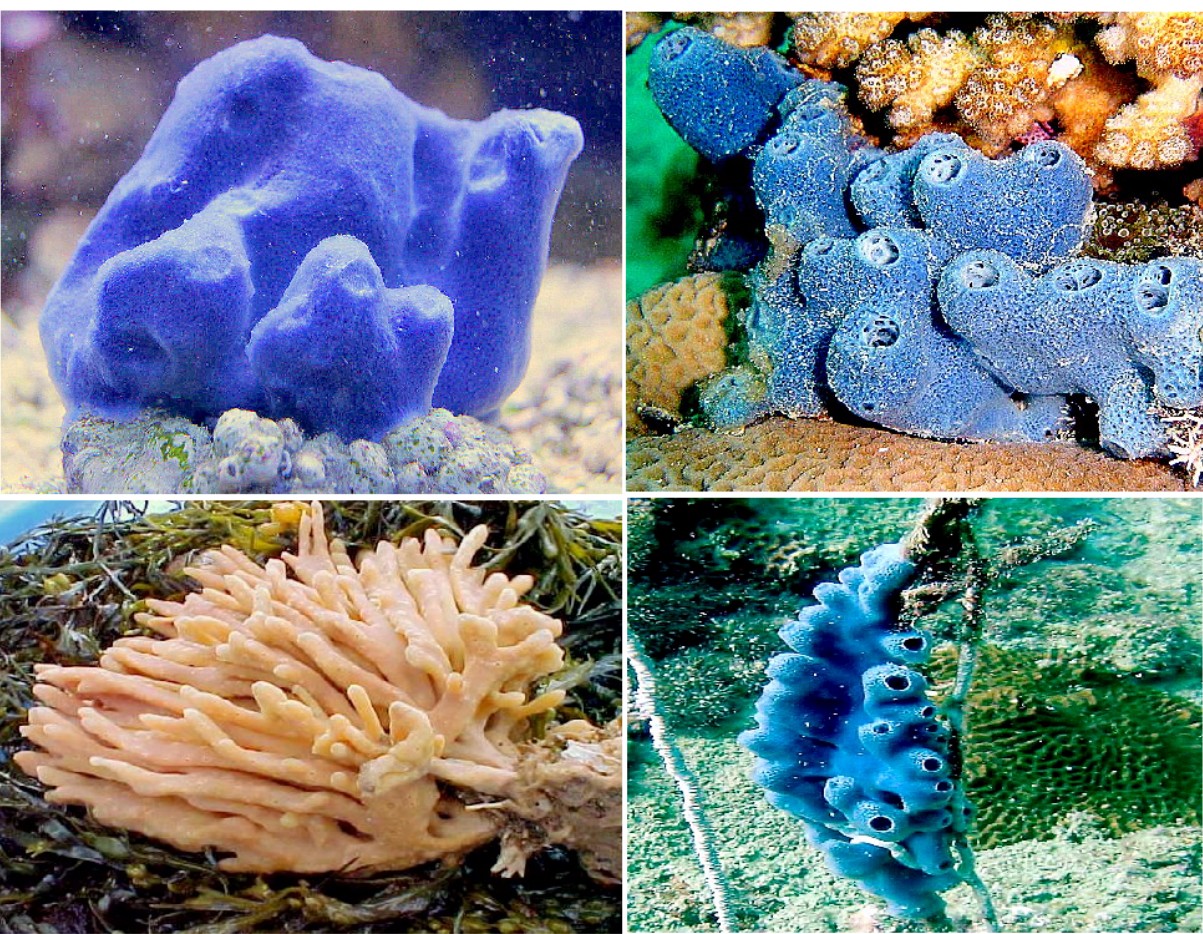

**Figure 33.** The marine sponges belonging to the genus *Haliclona* contain more than fifty species of actinobacteria belonging to the genera *Streptomyces*, *Nocardiopsis*, *Micromonospora* and *Verrucosispora*. Members of this genus produce large amounts of bioactive metabolites such as lipids, steroids, FA, lipopeptides and amino acids.

Cytotoxic depsipeptides, seragamides A–F, containing (2*R*,6*S*,8*R*,*E*)-8-Hydroxy-2,4,6-trimethylnon-4-enoic acid (**129**) were detected in the lipid extracts in the Okinawan sponge *Suberites japonicus*. The same FA has been found in jasplakinolide D, M, Q, and R1, as well as in cyclic depsipeptides named geodiamolides J, P, and R, which have been isolated from the marine sponge *Cymbastela* sp. and found in geodiamolides A and B from the sponge *Geodia* sp., and geodiamolide D from the sponge *Pseudoaxinyssa* sp. [253].

A cyclic depsipeptide, Jaspamide (jasplakinolide), containing (2*R*,6*S*,8*S*,*E*)-8-hydroxy-2,4,6-trimethylnon-4-enoic acid (**130**) was found in the lipid fraction of Fijian sponges of the genus *Jaspis* [254], and a similar peptide was found in other types of sponges [255]. The cytotoxic peptides, jaspamide and geodiamolide TA containing (*E*)-8-hydroxy-2,4,6-trimethylnon-4-enoic acid (**131**), found in the lipid fraction of the sponge *Hemiasterella*, while lipopeptides the geodiamolides J, K, and jaspamide B containing (2*R*,6*S*,8R)-8-hydroxy-2,6-dimethyl-4-methylene-5-oxononanoic acid (**132**) were isolated as minor metabolites from the sponge *Cymbastela* sp. [256].

Homophymines A–E and A1–E1 are a series of cyclodepsipeptides isolated from *Homophymia* sp. collected from shallow waters off the east coast of New Caledonia [257,258]. They are similar in structure to the previously published antiviral marine cyclodepsipeptides callipeltin A, neamphamide A, papuamides, theopapuamides, and mirabamides [259–265]. Homophymine A was cytotoxic against uninfected PBMC cells with an IC$_{50}$ of 1.19 μM, but it was almost sixteen times more effective against infected cells and exhibited potent cytotoxicity with IC$_{50}$ values ranging from 2 to 100 nM. These compounds were the most

potent against the PC3 human prostate adenocarcinoma and the SK-OV3 human ovarian adenocarcinoma cell lines [257–265]. (2*S*,3*S*,4*S*,6*S*)-3-Hydroxy-2,4,6-trimethyloctanoic acid (**134**, for structure see Figure 34, and activity is shown in Table 13) was isolated from the homophymines A and A1, (2*S*,3*S*,4*S*)-3-hydroxy-2,4,6-trimethylheptanoic acid (**135**) from the homophymines B and B1, (2*S*,3*S*,4*S*,6*S*)-3-hydroxy-2,4,6-trimethylnonanoic acid (**136**) from the homophymines C and C1, (2*S*,3*S*,4*S*,6*S*)-3-hydroxy-2,4,6,8-tetramethylnonanoic acid (**137**) from the homophymines D and D1, and (2*S*,3*S*,4*S*,6*S*,8*S*)-3-hydroxy-2,4,6,8-tetramethyldecanoic acid (**138**) was isolated from the homophymines E and E1.

**Figure 34.** Branched, unsaturated and glycosidic FA derived from sponge lipopeptides.

**Table 13.** Predicted biological activity of FA derived from sponge peptides.

| No. | Predicted Biological Activity, Pa * |
|---|---|
| 134 | Sclerosant (0.834); Hypolipemic (0.825); Antineoplastic (0.779); Anti-inflammatory (0.731) |
| 135 | Sclerosant (0.835); Antineoplastic (0.788); Hypolipemic (0.754); Anti-inflammatory (0.716) |
| 136 | Sclerosant (0.853); Hypolipemic (0.824); Antineoplastic (0.775); Anti-inflammatory (0.734) |
| 137 | Sclerosant (0.815); Hypolipemic (0.807); Antineoplastic (0.781); Anti-inflammatory (0.730) |
| 138 | Sclerosant (0.834); Hypolipemic (0.825); Antineoplastic (0.779); Anti-inflammatory (0.731) |
| 139 | Lipid metabolism regulator (0.903); Hypolipemic (0.848); Antineoplastic (0.805); Antifungal (0.782) |
| 140 | Sclerosant (0.834); Hypolipemic (0.825); Antineoplastic (0.779); Anti-inflammatory (0.731) |
| 141 | Restenosis treatment (0.827); Sclerosant (0.738); Neurodegenerative diseases treatment (0.722) |
| 142 | Sclerosant (0.834); Hypolipemic (0.825); Antineoplastic (0.779); Anti-inflammatory (0.731) |
| 143 | Antineoplastic (0.880); Antiviral (Arbovirus) (0.829); Apoptosis agonist (0.804) Hypolipemic (0.794); Antiprotozoal (Coccidial) (0.684); Antiviral (Picornavirus) (0.599) |
| 144 | Antineoplastic (0.885); Antiviral (Arbovirus) (0.814); Apoptosis agonist (0.800) Hypolipemic (0.794); Antiprotozoal (Coccidial) (0.621); Antiviral (Picornavirus) (0.598) |
| 145 | Anti-infective (0.934); Anti-hypercholesterolemic (0.916); Vasodilator (0.915) Antineoplastic (0.911); Vasoprotector (0.864); Lipid metabolism regulator (0.856) |
| 146 | Anti-infective (0.934); Anti-hypercholesterolemic (0.916); Vasodilator (0.915) Antineoplastic (0.911); Vasoprotector (0.864); Lipid metabolism regulator (0.856) |
| 147 | Anti-infective (0.934); Anti-hypercholesterolemic (0.916); Vasodilator (0.915) Antineoplastic (0.911); Vasoprotector (0.864); Lipid metabolism regulator (0.856) |

* Only activities with Pa > 0.5 are shown.

According to PASS data, among methyl-branched FA (**119–133**), of particular interest is FA (**127**). A rare feature of this acid that has been shown to be anti-psoriatic and anti-eczematic under the general concept of dermatologic activity with a high certainty of over 90%. The 3D graph of this methyl-branched FA (**127**) is shown in Figure 35.

(4*E*,6*E*)-2,3-Dihydroxy-2,6,8-trimethyldeca-4,6-dienoic acid (**139**) has been found in the depsipeptides papuamide A–D, which are produced by the sponge *Theonella* [262,263]. HIV-inhibitory depsipeptides, mirabamides A–D, contain (2*R*,3*R*,4*R*)-3-hydroxy-2,4,6-trimethyloctanoic acid (**140**) and were extracted from *Siliquariaspongia mirabilis*, while the cyclic depsipeptide neamphamide D also contains this FA and was found in the Australian marine sponge *Neamphius huxleyi* [266].

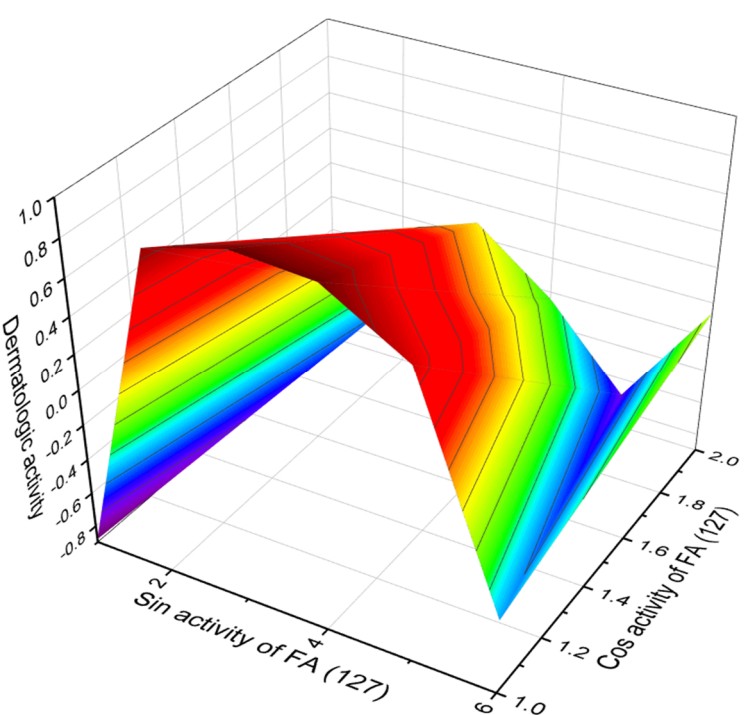

**Figure 35.** 3D graph showing the predicted and calculated dermatologic activity of methyl-branched FA (**127**). This acid is incorporated into the cyclic depsipeptide halipeptin A, which is found in the marine sponge *Haliclona* sp.

It is known that depsipeptides called didemnins, which are cytotoxins and immunosuppressive agents, were first isolated over 40 years ago from the Caribbean tunicate *Trididemnum solidum*, and contain (2*S*,4*S*)-4-hydroxy-2,5-dimethyl-3-oxohexanoic acid (**141**) [267]. However, recent data indicate that didemnins do not synthesize tunicate, but rather the symbiotic bacteria *Tistrella mobilis* [268]. These the symbiotic bacteria of the genus *Tistrella* have been found in marine sponges and appear to synthesize depsipeptides like didemnins [269].

Cytotoxic undecapeptides, theopapuamides and celebesides A–C from the sponge *Theonella swinhoei,* showed anticancer activity against HCT–116 cells (colon cancer) [270]. (2*R*,3*R*)–3–Hydroxy–2,4,6–trimethyloctanoic acid (**142**) was present in the undecapeptides theopapuamide A–D, (2*E*,4*E*,7*S*,8*R*,9*S*,10*R*)–7,9–dihydroxy–8,10-dimethyltrideca–2,4–dienoic acid (**143**) was isolated from celebeside A and C, and (2*E*,4*E*,7*S*,8*R*,9*S*,10*R*)–7,9–dihydroxy–8,10–dimethyldodeca–2,4–dienoic acid (**144**) was found in celebeside B [264]. Cytotoxic cyclic peptides, aciculitins A–C, were found in the active lipid fraction of the lithistid sponge *Aciculites orientalis*. Aciculitin A, containing FA (**145**) and FA (**146**), was present in aciculitin B, and aciculitin C contains FA (**147**) [271].

According to PASS data, among the group of fatty acids (**134–147**), glycosidic FA (**145**, **146** and **147**) are of the greatest interest, which demonstrate anti-infective and antineoplastic activities with a high degree of confidence, more than 93%. Figure 36 demonstrates the 3D graph of the activities of these acids.

A potent cytotoxin, psymberin, also known as irciniastatin A, is found in the sponge *Psammocinia* sp. with (2*S*)-2-hydroxy-3-methoxy-5-methylhex-5-enoic acid (**148**, for structure see Figure 37, and predicted activity is shown in Table 14) [272–275] and the keto analogue irciniastatin B was isolated from *Ircinia ramosa* (Borneo) and contains 2-hydroxy-3-methoxy-5-methylhex-5-enoic acid (**149**) [276,277].

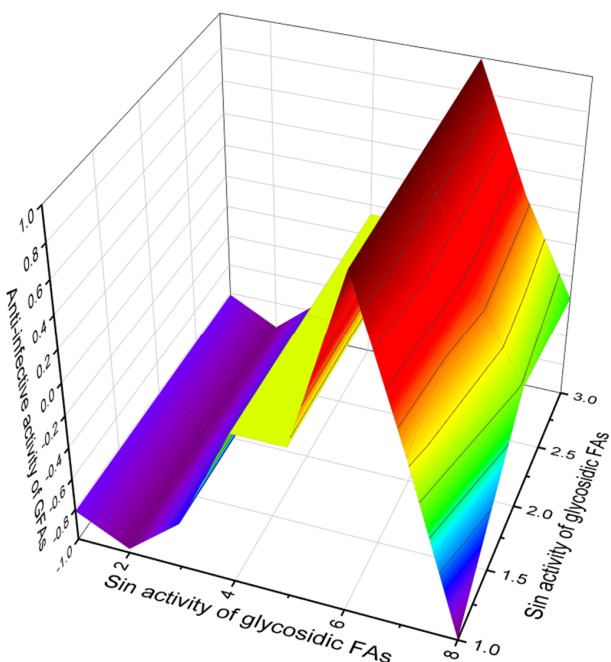

**Figure 36.** 3D graph showing the predicted and calculated anti-infective and antineoplastic activities of glycosidic FA (**145**, **146** and **147**). These acids are incorporated into cyclic peptides called aciculitins A–C and are produced by the lithistid sponge *Aciculites orientalis*.

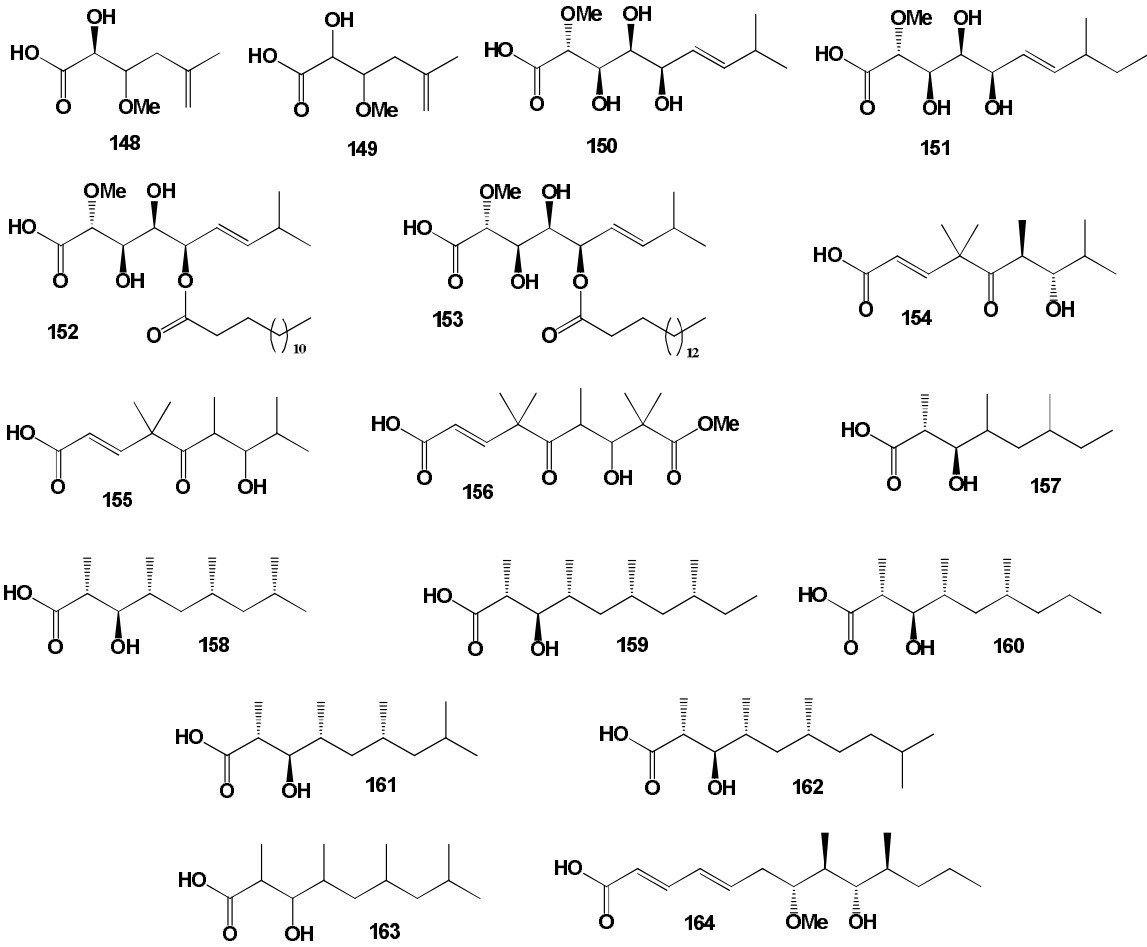

**Figure 37.** Branched, saturated, and unsaturated FA derived from sponge lipopeptides.

Marine sponges belonging to the Jaspidae family produce related bioactive lipopeptides, bengamides [278–282]. Thus, bengamides AE, G, H, J, L, M, O, Y, and Z contain (2*R*,3*R*,4*S*,5*R*,*E*)-3,4,5-trihydroxy-2-methoxy-8-methylnon-6-enoic acid (**150**), bengamides E' and F' contain (2*R*,3*R*,4*S*,5*R*,*E*)-3,4,5-trihydroxy-2-methoxy-8-methyldec-6-enoic acid (**151**), bengamides P and Q contain (2*R*,3*R*,4*R*,5*R*,*E*)-3,4-dihydroxy-2-methoxy-8-methyl-5-(tetradecanoyloxy)non-6-enoic acid (**152**), and (2*R*,3*R*,4*R*,5*R*,*E*)-3,4-dihydroxy-2-methoxy-8-methyl-5-(palmitoyloxy)non-6-enoic acid (**153**, activity see in Figure 38) was found in bengamide R [278–283].

(6*S*,7*S*,*E*)-7-Hydroxy-4,4,6,8-tetramethyl-5-oxonon-2-enoic acid (**154**) was incorporated into a macrocyclic lactam, mirabalin, which was found and isolated from extracts of *Siliquariaspongia mirabilis*. Mirabalin is known to inhibit the growth of the HCT-116 cell line [284–286]. Poecillastrin C and D are isolated from the deep-sea sponge, *Japsis serpentine* [287,288]. These compounds showed potent cytotoxicity against various tumor cell lines, and both poecillastrins contain (*E*)-7-hydroxy-4,4,6,8-tetramethyl-5-oxonon-2-enoic acid (**153**) [287].

**Table 14.** Predicted biological activity of FA derived from peptides of marine sponges.

| No. | Predicted Biological Activity, Pa * |
|---|---|
| **148** | Antineoplastic (0.752); Lipid metabolism regulator (0.715); Antiviral (Arbovirus) (0.642) |
| **149** | Antineoplastic (0.752); Lipid metabolism regulator (0.715); Antiviral (Arbovirus) (0.642) |
| **150** | Cell adhesion molecule inhibitor (0.883); Lipid metabolism regulator (0.801); Apoptosis agonist (0.752) |
| **151** | Cell adhesion molecule inhibitor (0.866); Hypolipemic (0.816); Antineoplastic (0.780) |
| **152** | Lipid metabolism regulator (0.931); Antineoplastic (0.826); Apoptosis agonist (0.634) |
| **153** | Lipid metabolism regulator (0.931); Antineoplastic (0.826); Apoptosis agonist (0.634) |
| **154** | Antineoplastic (0.813); Antiviral (Arbovirus) (0.748); Lipid metabolism regulator (0.693) |
| **156** | Antineoplastic (0.813); Antiviral (Arbovirus) (0.748); Lipid metabolism regulator (0.693) |
| **157** | Antineoplastic (0.779); Acute neurologic disorders treatment (0.681); Antiviral (Arbovirus) (0.680) |
| **158** | Sclerosant (0.815); Antineoplastic (0.781); Acute neurologic disorders treatment (0.722) |
| **159** | Sclerosant (0.834); Antineoplastic (0.799); Acute neurologic disorders treatment (0.725) |
| **160** | Sclerosant (0.835); Antineoplastic (0.795); Acute neurologic disorders treatment (0.731) |
| **161** | Sclerosant (0.815); Antineoplastic (0.781); Acute neurologic disorders treatment (0.722) |
| **162** | Sclerosant (0.835); Antineoplastic (0.781); Acute neurologic disorders treatment (0.764) |
| **163** | Sclerosant (0.815); Antineoplastic (0.781); Acute neurologic disorders treatment (0.722) |
| **164** | Antineoplastic (0.881); Hypolipemic (0.797); Antifungal (0.793); Antimitotic (0.787) |

* Only activities with Pa > 0.5 are shown.

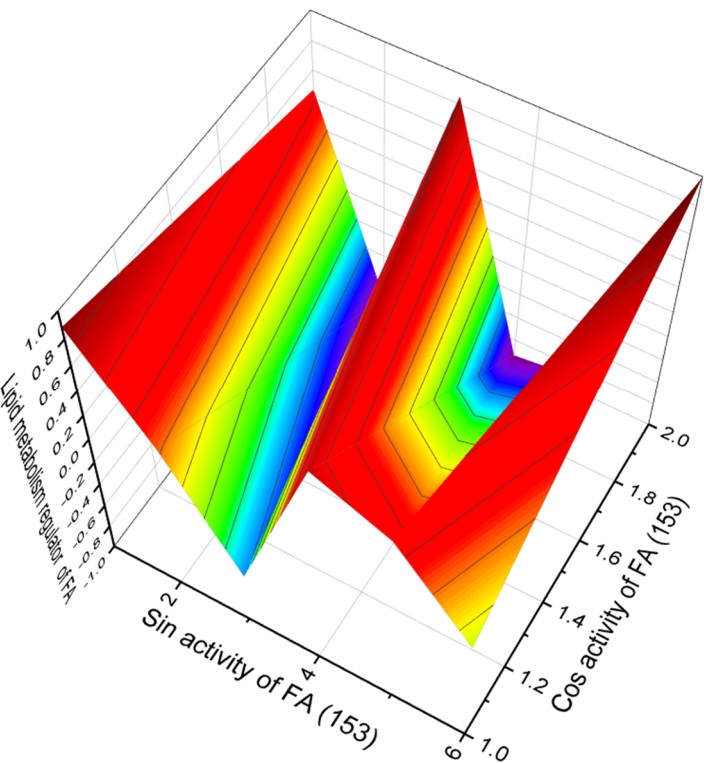

**Figure 38.** 3D graph showing the predicted and calculated activity of a regulator lipid metabolism of FA (**153**). Acid **153** has a similar activity since the structures of both metabolites are similar. These acids are found in lipopeptides, bengamides P and Q, and are produced by marine sponges belonging to the Jaspidae family.

An aqueous extract of the marine sponge *Chondropsis* sp. contains several macrolides called chondropsins [289]. Chondropsin A, B, and D and deoxychondropsin A contain (*E*)-7-hydroxy-9-methoxy-4,4,6,8,8-pentamethyl-5,9-dioxonon-2-enoic acid (**156**), and (*E*)-7-hydroxy-4,4,6,8-tetramethyl-5-oxonon-2-enoic acid (**155**) was found in chondropsin C [289].

Cytotoxic peptides, theopapuamides A–D, which contain (2*R*,3*R*)-3-hydroxy-2,4,6-trimethyloctanoic acid (**157**) were obtained from *Theonella swinhoei* sponge extracts [290], while geodiamolide TA was isolated from the marine sponge *Hemiasterella minor*, which also contained the same FA [291].

The cyclodepsipeptides named homophymines A–E and A1–E1 were obtained from lipid extracts of the marine sponge *Homophymia* sp. living in the island of Barneo [257,258]. All the members described so far exhibit potent cytotoxic activity. (2*R*,3*R*,4*R*,6*R*)-3-Hydroxy-2,4,6,8-tetramethylnonanoic acid (**158**) is found in homophymines B and B1, (2*R*,3*R*,4*R*,6*R*,8*R*)-3-hydroxy-2,4,6,8-tetramethyldecanoic acid (**159**) in homophymines A and A1. Homophymines C and C1 contain (2*R*,3*R*,4*R*,6*R*)-3-hydroxy-2,4,6-trimethylnonanoic acid (**160**), homophymines D and D1 contain (2*R*,3*R*,4*R*,6*R*)-3-hydroxy- 2,4,6,8-tetramethylnonanoic acid (**161**), and homophymines E and E1 contain (2*R*,3*R*,4*R*,6*R*)-3-hydroxy-2,4,6,9-tetramethyldecanoic acid (**162**) [257,258].

The sponge *Homophymia lamellosa* from the coast of Madagascar yielded cytotoxic cyclic depsipeptides, pipecolidepsins. Both pipecolidepsins A and B contain FA (**134**), and 3-hydroxy-2,4,6,8-tetramethylnonanoic acid (**163**) is found in pipecolidepsin C [292].

(2*E*,4*E*,7*R*,8*S*,9*S*,10*S*)-9-Hydroxy-7-methoxy-8,10-dimethyltrideca-2,4-dienoic acid (**164**) is incorporated into depsipeptide nagahamide A, which demonstrated antibacterial properties and was found in a water–methanol extract of the marine sponge *Theonella swinhoei* [293].

### 3.2.2. Chlorinated Fatty Acids Derived from Sponge Lipopeptides

It is known that chlorinated fatty acids are widely distributed in nature and are part of neutral lipids, phospholipids, and glycolipids, and are also found in natural lipopeptides of marine invertebrates [203]. Polychlorinated peptides from *Lamellodysidea herbacea* such as dysidin and dysidenin contain (*S*)-4,4,4-trichloro-3-methylbutanoic acid (**165**, for structure see Figure 39, and activity is shown in Table 15) [294]. Five dysideaprolines A–F contain FA (**166** and **167**), and barbaleucamides A and B, which contain (*E*)-6,6,6-trichloro-3-methoxy-5-methylhex-2-enoic acid (**168**), were obtained from the Philippines sponge *Dysidea* sp. (*E*)-6,6,6-Trichloro-5-methylhex-2-enoic acid, **169**) or herbacic acid is the major trichloroleucine metabolite of herbaceamide A in the sponge *Dysidea herbacea* [294,295] and FA (**170**) was isolated from chlorinated lipopeptides found in the marine sponge *Dysidea* sp.

**Figure 39.** Chlorinated FA derived from sponge lipopeptides.

**Table 15.** Predicted biological activity of FA from lipopeptides of *Dysidea* species.

| No. | Predicted Biological Activity, Pa * |
|:---:|:---:|
| **165** | Antineoplastic (0.841); Preneoplastic conditions treatment (0.689); Antiprotozoal (0.586) |
| **166** | Antineoplastic (0.841); Preneoplastic conditions treatment (0.689); Antiprotozoal (0.586) |
| **167** | Antineoplastic (0.960); Anti-infective (0.613); Acute neurologic disorders treatment (0.572) |
| **168** | Antineoplastic (0.774); Preneoplastic conditions treatment (0.690); Antiprotozoal (0.557) |
| **169** | Antineoplastic (0.850); Antiviral (Arbovirus) (0.765); Acute neurologic disorders treatment (0.574) |
| **170** | Antineoplastic (0.965); Anti-infective (0.628); Acute neurologic disorders treatment (0.589) |

* Only activities with Pa > 0.5 are shown.

Analyzing the PASS data, all chlorine-containing FA (**165–170**) show the dominant property of anticancer activity with varying degrees of reliability. Strong antitumor activity is characteristic of **165** and **170** acids. Figure 40 demonstrates the 3D graph which shows the predicted and calculated antitumor activity of fatty acid (**170**).

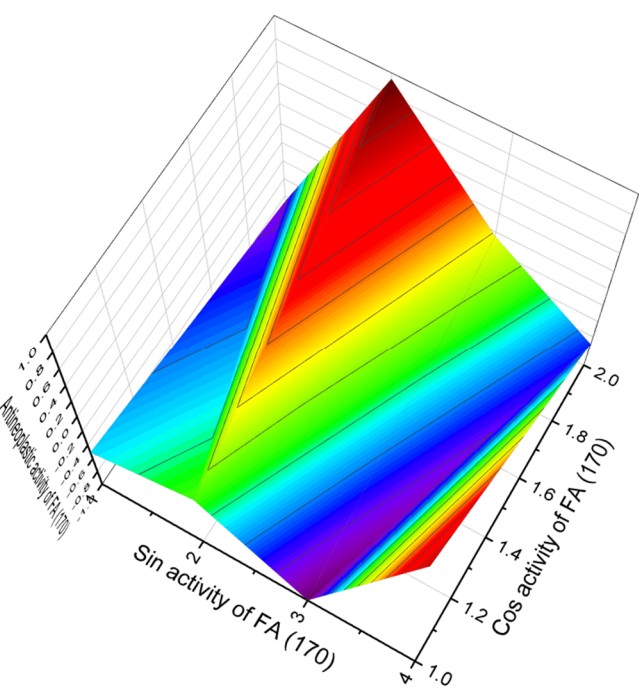

**Figure 40.** 3D graph showing the predicted and calculated antineoplastic activity of chlorinated FA (**170**). The figure shows that a single peak (red zone) dominates, which corresponds to the strong antitumor activity of (*S*)−4,4−dichloro−3−methylbutanoic acid.

### 3.2.3. Miscellaneous Fatty Acids Incorporated into Sponge Lipopeptides

A cytotoxic cyclic didepsipeptide named arenastatin A containing (5*S*,6*S*,7*S*,*E*)-6-hydroxy-5,6-dimethyl-7-(3-phenyloxiran-2-yl)oct-2-enoic acid (**171**, for structure see Figure 41, and activity is shown in Table 16) was found in a chloroform–methanol extract of the Okinawan marine sponge *Dysidea arenaria* [296–299]. This cyclodepsipeptide has an extremely strong cytotoxic activity against KB 3-1 cells (human epidermoid carcinoma cell line) [300–303].

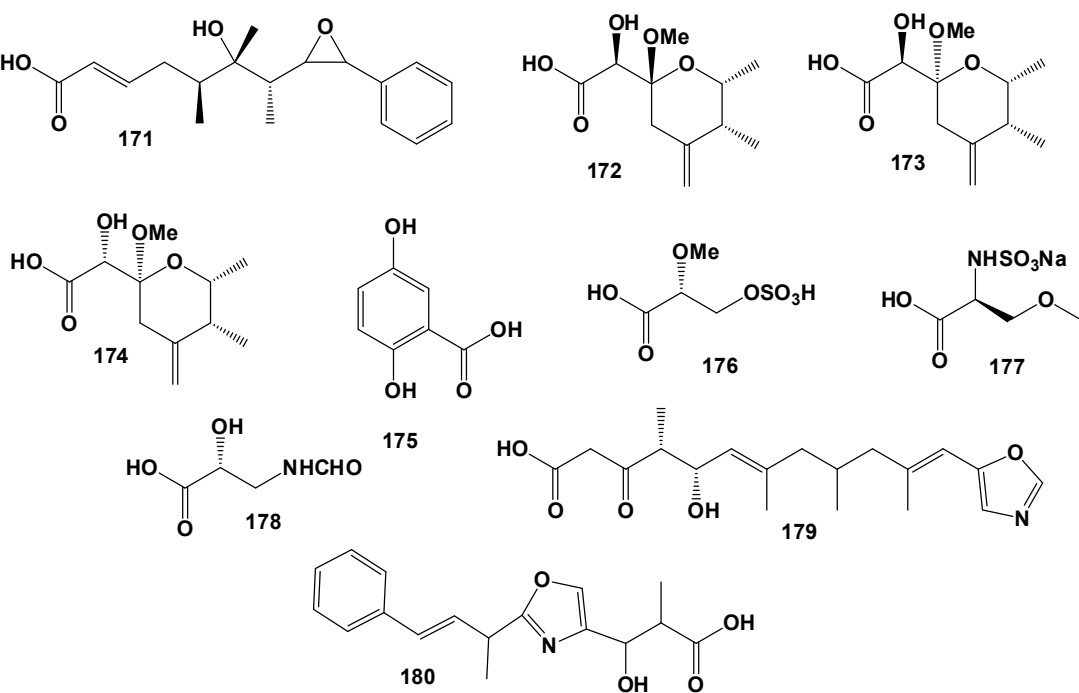

**Figure 41.** Miscellaneous FA derived from sponge lipopeptides.

The cytotoxic compounds onnamide A, B, and C were obtained from marine sponge *Theonella* sp. Onnamide A contains (*S*)-2-hydroxy-2-((2*R*,5*R*,6*R*)-2-methoxy-5,6-dimethyl-4-methylene-tetrahydro-2H-pyran-2-yl)acetic acid (**172**), onnamide B contains (*S*)-2-hydroxy-2-((2*S*,5*R*,6R)-2-methoxy-5,6-dimethyl-4-methylenetetrahydro-2H-pyran-2-yl)acetic acid (**173**), and onnamide C contains (*R*)-2-hydroxy-2-((2*S*,5*R*,6*R*)-2-methoxy-5,6-dimethyl-4-methylenetetrahydro-2H-pyran-2-yl)acetic acid (**174**). Onnamide A analogues, 21,22-dihydroxyonnamides A1–A4, containing FA (**172**), were isolated from an Okinawan collection of *Theonella swinhoei* [304–306]. A cyclic peptide oriamide with 2,5-dihydroxybenzoic acid (**175**), was detected in the marine sponge *Theonella* sp. collected in Sodwana Bay [307].

It is known that dysinosin A is an inhibitor of Factor VIIa and thrombin and is produced by the Australian sponge of the family Dysideidae, and contains a sulfated glyceric acid, (*R*)-2-methoxy-3-(sulfooxy)-propanoic acid (**176**), as its analogues dysinosins B and C contain this FA (**176**) [308,309].

**Table 16.** Predicted biological activity of FA from sponge lipopeptides.

| No. | Predicted Biological Activity, Pa * |
|:---:|:---:|
| **171** | Anti-hypercholesterolemic (0.873); Lipid metabolism regulator (0.713); Atherosclerosis treatment (0.559) |
| **172** | Antineoplastic (0.962); Apoptosis agonist (0.955); Antiparasitic (0.703); Antiprotozoal (0.590) |
| **173** | Antineoplastic (0.962); Apoptosis agonist (0.955); Antiparasitic (0.703); Antiprotozoal (0.590) |
| **174** | Antineoplastic (0.962); Apoptosis agonist (0.955); Antiparasitic (0.703); Antiprotozoal (0.590) |
| **175** | Antiseptic (0.945); Antiinfective (0.900); Preneoplastic conditions treatment (0.818) |
| **176** | Acute neurologic disorders treatment (0.858); Antidiabetic (0.818); Antidiabetic (type 2) (0.645) |
| **177** | Atherosclerosis treatment (0.857); Sweetener (0.635); Restenosis treatment (0.602) |
| **178** | Antihypertensive (0.765); Antidiabetic (0.757); Antithrombotic (0.522) |
| **179** | Antineoplastic (0.855); Transplant rejection treatment (0.591); Autoimmune disorders treatment (0.574) |
| **180** | Antineoplastic (0.667); Angiogenesis stimulant (0.566); Antidiabetic (0.531) |

* Only activities with Pa > 0.5 are shown.

The cyclic peptide, scleritodermin A with sodium (*S*)-(1-carboxy-2-methoxyethyl)-sulfamate (**177**) inhibited tubulin polymerization and showed significant in vitro cytotoxicity against human tumor cell lines [310] and was isolated from the lithistid sponge *Scleritoderma nodosum*. The bioactive hexapeptide, keramamide A, from the Okinawan marine sponge *Theonella* sp. contains (*R*)-3-formamido-2-hydroxypropanoic acid (**178**), which was also found in keramamides A, J, K, H, and G [311].

(4*R*,5*S*,6*E*,11*E*)-5-Hydroxy-4,7,9,11-tetramethyl-12-(oxazol-5-yl)-3-oxododeca-6,11-dienoic acid (**179**) is incorporated into a cytotoxic lipodepsipeptide named taumycin A, which was obtained from the Madagascar sponge *Fascaplysinopsis* sp. [256], and the sponge *Discodermia kiiensis* yielded the cyclic depsipeptides, discokiolide A–C, with (*E*)-3-hydroxy-2-methyl-3-(2-(4-phenylbut-3-en-2-yl)-oxazol-4-yl)-propanoic acid (**180**) [312].

According to PASS data, among FA (**171–180**), tetrahydro-2H-pyran-containing FA (**172**, **173** and **174**) are of the greatest interest, which demonstrate antineoplastic activity with a high degree of certainty, more than 96%. Figure 42 shows the 3D graph of FA (**173**) activity, and a single peak in the red area corresponds to strong antitumor activity.

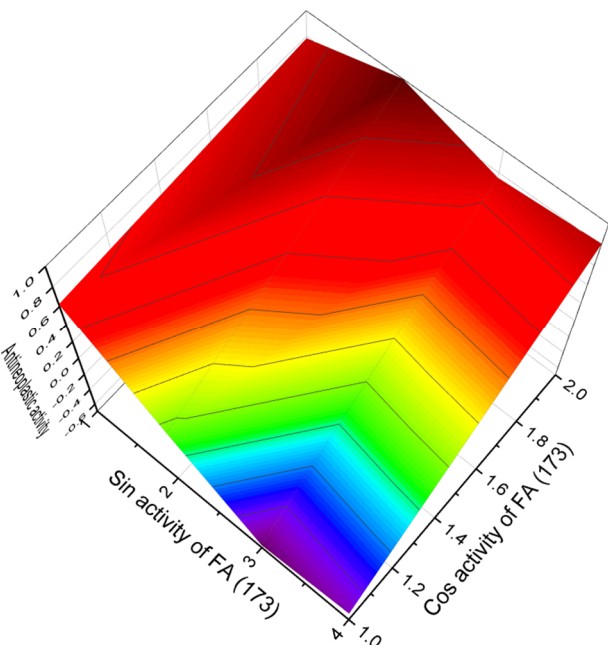

**Figure 42.** 3D graph showing the predicted and calculated antineoplastic activity of tetrahydro−2H−pyran−containing FA (**173**). This acid is part of the onnamide B lipopeptide, which exhibits highly cytotoxic activity against the P388 cell line.

Marine sponges, like their freshwater relatives, often contain dense and diverse microbial communities or symbionts, with many microorganisms specific to sponge hosts. Symbiont microorganisms can include bacteria, archaea, and unicellular eukaryotes (fungi and microalgae), and account for up to 40% of the volume of the sponge. These symbionts synthesize a wide variety of organic molecules, including lipopeptides, which can have a profound effect on the biology of the host sponge. To date, there is no definite answer as to who the true producer of certain organic molecules isolated from the body of sponges is. Therefore, when we say that lipopeptides are isolated from sea sponges, this does not mean that these lipopeptides were synthesized by the sponge; they can be synthesized, for example, by fungal endophytes, microalgae, or bacteria.

It is very difficult to characterize the FA composition of lipopeptides in algae and invertebrates, and especially in marine and freshwater sponges. This is since sponges and other invertebrates contain a huge pool of various symbiotic bacteria and fungi. And it is not easy to determine what contribution the symbionts of invertebrates make. It should be noted that most invertebrate lipopeptide FA contain similar fragments of bacterial FA, such as chlorine-containing, oxirane, polymethyl- or phenyl-containing FA. But there are fragments of FA that are not found in bacterial lipopeptides, such as aziridine and tetrahydro-2H-pyran-containing FA.

## 4. Fatty Acids Derived from Mollusca Lipopeptides

Numerous scientific publications have shown that gastropods are a rich source of bioactive compounds that include steroids, terpenoids, polyketides, FA, and lipopeptides. Many of the drugs found demonstrate anticancer, antibacterial, and antifungal properties [313–319].

(2*E*,5*S*,6*S*,7*S*,8*E*)-5,7-Dihydroxy-2,6,8-trimethyldeca-2,8-dienoic acid (**181**, for structure see Figure 43, activity is shown in Table 17, and picture of this mollusc is shown in Figure 43) is found in a cytotoxic depsipeptide, kulokekahilide-2, which is derived from a cephalaspidean mollusc, *Philinopsis speciosa* [320]. Kulolide-1 contains (*S*)-3-hydroxy-2,2-dimethyloct-7-enoic acid (**182**), kulolide-2 contains (*S*)-3-hydroxy-2,2-dimethyloct-7-enoic acid (**183**) [321,322] and kulomoopunalide-2 contains (2*R*,3*S*)-3-hydroxy-2-methyloct-7-ynoic acid (**184**) [323]. All depsipeptides were obtained from *Ph. Speciosa* [321–323]. Cyto-

toxic depsipcptides, onchidin A and B, were found in extracts of the ulmonated mollusc *Onchidium* sp. and contain (*S*)-2-hydroxy-3-methylbutanoic acid (**185**) and 3-hydroxy-2-methyloct-7-ynoic acid (**186**), respectively [324,325].

**Figure 43.** FA derived from mollusc lipopeptides.

**Table 17.** Predicted biological activity of FA from lipopeptides of molluscs.

| No. | Predicted Biological Activity, Pa * |
|---|---|
| **181** | Antineoplastic (0.871); Apoptosis agonist (0.764); Anti-inflammatory (0.729) |
| **182** | Anti-psoriatic (0.923); Anti-eczematic (0.721); Alzheimer's disease treatment (0.591) |
| **183** | Anti-psoriatic (0.924); Anti-eczematic (0.879); Antifungal (0.603) |
| **184** | Anti-inflammatory (0.751); Antiviral (Arbovirus) (0.595); Antifungal (0.579) |
| **185** | Anti-hypoxic (0.768); Antihypertensive (0.730); Antiviral (Picornavirus) (0.613) |
| **186** | Anti-inflammatory (0.751); Antiviral (Arbovirus) (0.595); Antifungal (0.579) |
| **187** | Lipid metabolism regulator (0.822); Hypolipemic (0.757); Anti-hypercholesterolemic (0.677) |
| **188** | Lipid metabolism regulator (0.889); Hypolipemic (0.685); Anti-hypercholesterolemic (0.644) |
| **189** | Antineoplastic (0.859); Apoptosis agonist (0.765); Atherosclerosis treatment (0.640) |
| **190** | Antidiabetic (0.916); Leukopoiesis stimulant (0.822); Atherosclerosis treatment (0.568) |
| **191** | Antineoplastic (0.834); Apoptosis agonist (0.818); Acute neurologic disorders treatment (0.795) |
| **192** | Anesthetic general (0.759); Antiviral (Arbovirus) (0.737); Antitoxic (0.728); Antihypoxic (0.726) |

**Table 17.** *Cont.*

| No. | Predicted Biological Activity, Pa * |
|---|---|
| 193 | Antihypoxic (0.758); Antiviral (Arbovirus) (0.735); Antiviral (Picornavirus) (0.671) |
| 194 | Antineoplastic (0.796); Apoptosis agonist (0.719); Antiparasitic (0.711); Antifungal (0.675) |
| 195 | Acute neurologic disorders treatment (0.754); Antiviral (Arbovirus) (0.733); Antineurotic (0.555) |
| 196 | Lipid metabolism regulator (0.809); Hypolipemic (0.758); Antihypertensive (0.741) Anti-hypercholesterolemic (0.616); Atherosclerosis treatment (0.535) |
| 197 | Hypolipemic (0.796); Antineoplastic (0.614); Preneoplastic conditions treatment (0.547) |
| 198 | Hypolipemic (0.804); Antineoplastic (0.689); Preneoplastic conditions treatment (0.588) |

* Only activities with Pa > 0.5 are shown.

Cyclic depsipeptides, kahalalides R, S and kahalalides F, D, were isolated from the mollusc *Elysia grandifolia* [326], and kahalalide F was also found in the molluscs *Elysia rufescens*, the bivalve mollusc *Spisula polynyma*, and from the green alga *Bryopsis* sp. [327]. Kahalalide S contains 5-hydroxy-7-methyloctanoic acid (**187**), kahalalide F contains 5-methylhexanoic acid, and kahalalide D contains 3-hydroxy-7-methyloctanoic acid (**188**).

A 26-membered cyclodepsipeptide, aurilide, with (2*E*,5*R*,6*R*,7*S*,8*E*)-5,7-dihydroxy-2,6,8-trimethylundeca-2,8-dienoic acid (**189**), has been isolated from the Japanese sea hare *Dolabella auricularia* (Figure 44) [328], and an antineoplastic agent, dolastatin 13 with 3-hydroxy-2-methoxypropanoic acid (**190**) was found in the same sea hare [329,330].

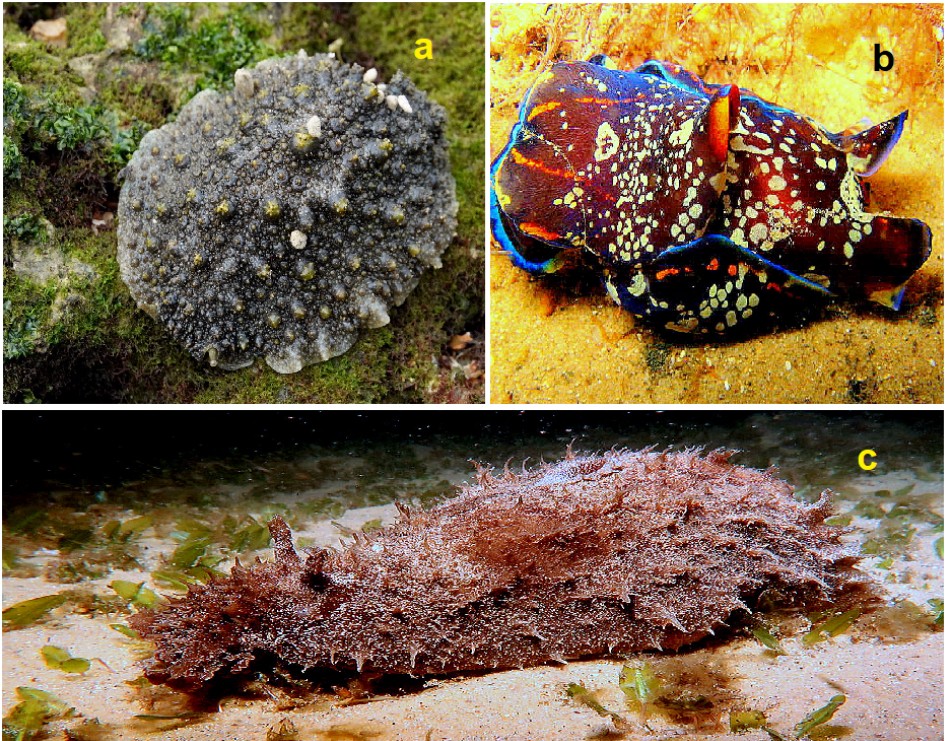

**Figure 44.** Marine molluscs: (**a**), The pulmonate mollusc *Onchidium* sp.; (**b**), a cephalaspidean mollusc, *Philinopsis speciosa*; (**c**), Sea hare *Dolabella auricularia.* All these marine molluscs share the ability to synthesize lipopeptides, although it is possible that they can obtain related lipopeptides from algae.

(2*E*,4*Z*,10*E*)-15-Hydroxy-7-methoxy-2-methylhexadeca-2,4,10-trienoic acid (**191**) was incorporated into a cytostatic depsipeptide, dolastatin 14 from the Indian Ocean shell-less mollusc *Dolabella auricularia* [331], while dolastaine C was found in the Japanese sea hare *D. auricularia* and contains (2*S*,3*R*)-2-(dimethylamino)-3-methylpentanoic acid (192) [332]. Dolastatin H, isodolastatin H, and dolastatin 10 contain (*S*)-2-(dimethylamino)-3-methylbutanoic acid (**193**) [333]. Two 35-membered depsipeptides, dolastatin G and nordolastatin G, demonstrated strong cytotoxicity against HeLa cells. Both the depsipeptides contain (2*Z*,4*E*,7*R*,8*S*)-8-hydroxy-3-methoxy-4,7-dimethylnona-2,4-dienoic (**194**) and (2*R*,3*R*,7*S*)-3,7-dihydroxy-2,8-dimethyl-nonanoic (**195**) acids [334]. Two antineoplastic cyclic depsipeptides, designated dolastatin 11 and dolastatin 12, were isolated from the Indian Ocean *D. auricularia*, and both depsipeptides contain (3*S*)-2-hydroxy-3-methylpentanoic acid (**196**) [330,335,336], dolastatin 18 contain 2,2-dimethyl-3-oxohexanoic acid (**197**) [337–339], and the cytotoxic cyclic depsipeptide, (-)-doliculide, similar to jasplakinolide, contains (2*S*,3*S*,5*S*,6*S*,8*S*)-6,8-dihydroxy-2,3,5,9-tetramethyl-decanoic acid (**198**) [340].

According to the PASS data, among FA (**181–198**) isolated from mollusc lipopeptides, the most interesting is FA (**183**), which demonstrates anti-psoriatic activity with a high degree of certainty, more than 92%. Figure 45 shows the 3D graph of the activity of acid **183**; two peaks in the red area correspond to strong anti-psoriatic and anti-eczematic activities.

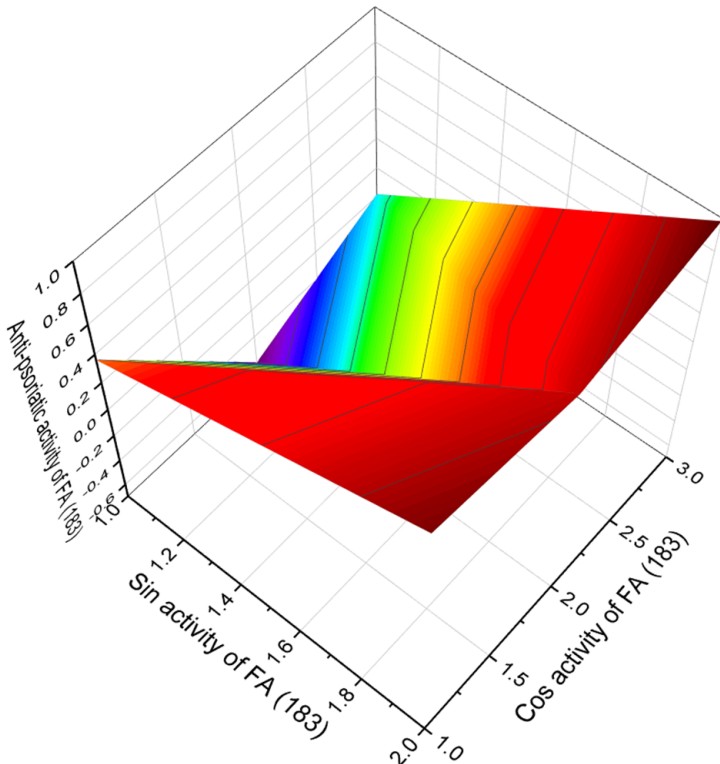

**Figure 45.** 3D graph showing the predicted and calculated anti-psoriatic activity of FA (**183**). (*S*)−3−hydroxy−2,2−dimethyloct−7−enoic acid (**183**) was incorporated into the lipopeptide kulolide 2 from cephalaspidean molluscs, *Philinopsis speciosa*.

Lipopeptide FA isolated from marine or freshwater molluscs are not very diverse. It is known that molluscs tend to eat microalgae, macrophytes, and algae residues where there may be various bacterial communities. The mollusc lipopeptides found in their bodies are likely to be ingested, since FA do not fundamentally differ in structure from bacterial FA.

## 5. Fatty Acids Derived from Tunicate Lipopeptides

Marine ascidians are considered one of the richest sources of biologically active substances, including lipopeptides [341–350]. Many biologically active compounds of marine ascidians are already at different stages of clinical and preclinical research. Various lipopeptides have antitumor, antihypertensive, antioxidant and antimicrobial properties [58,351–354]. Below we present some data on lipopeptides that are the most interesting from the point of view of medicine and pharmacology.

Sagittamide A and B have been isolated from a tropical tunicate, *Dolabella auricularia* (Micronesia) [355,356]; other minor congeners, sagittamides C–F were isolated from *Didemnidae ascidia* [355–357]. According to published data, sagittamides A-F each have different fatty acids (**199–204**, for structures see Figure 46, activity see in Table 18, and the same samples of tunicate are shown in Figure 47).

**Figure 46.** FA derived from Tunicata lipopeptides.

**Table 18.** Predicted biological activity of FA from lipopeptides of tunicates.

| No. | Predicted Biological Activity, Pa * |
|---|---|
| **199** | Neuroprotector (0.748); Immunosuppressant (0.710); Acute neurologic disorders treatment (0.645) |
| **200** | Lipid metabolism regulator (0.901); Macrophage stimulant (0.829); Antineoplastic (0.777) |
| **201** | Neuroprotector (0.748); Immunosuppressant (0.710); Acute neurologic disorders treatment (0.645) |
| **202** | Lipid metabolism regulator (0.850); Antidiabetic symptomatic (0.740) Atherosclerosis treatment (0.654); Hypolipemic (0.653); Antidiabetic (0.605) |
| **203** | Lipid metabolism regulator (0.850); Antidiabetic symptomatic (0.740) Atherosclerosis treatment (0.654); Hypolipemic (0.653); Antidiabetic (0.605) |
| **204** | Lipid metabolism regulator (0.850); Antidiabetic symptomatic (0.740) Atherosclerosis treatment (0.654); Hypolipemic (0.653); Antidiabetic (0.605) |
| **205** | Lipid metabolism regulator (0.857); Apoptosis agonist (0.729); Immunosuppressant (0.689) |
| **206** | Lipid metabolism regulator (0.879); Immunosuppressant (0.760); Apoptosis agonist (0.744) |
| **207** | Hypolipemic (0.880); Lipid metabolism regulator (0.861); Anti-hypercholesterolemic (0.683) Atherosclerosis treatment (0.634); Neurodegenerative diseases treatment (0.610) |
| **208** | Restenosis treatment (0.827); Neurodegenerative diseases treatment (0.722); Antidiabetic (0.655) |
| **209** | Restenosis treatment (0.827); Neurodegenerative diseases treatment (0.722); Antidiabetic (0.655) |
| **210** | Restenosis treatment (0.827); Neurodegenerative diseases treatment (0.722); Antidiabetic (0.655) |
| **211** | Mucositis treatment (0.803); Antimutagenic (0.727); Cytoprotectant (0.719) |

* Only activities with Pa > 0.5 are shown.

Cyclic polyether lipopeptides named bistramides B, C, D, and K, which were related to bistramide A from the ascidian *Lissoclinum bistratum*, were found in the extracts and their structures were determined. Bistramides A–C contain FA (**205**), bistramide D contains FA (**206**), and bistramide K contains FA (**207**) [358–361].

Cyclic depsipeptide from a tunicate of the genus *Trididemnum*, didemnins A, B and D, contain (2*R*,4*R*)-4-hydroxy-2,5-dimethyl-3-oxohexanoic (**208**), (2*S*,4*R*)-4-hydroxy-2,5-dimethyl-3-oxohexanoic [209], and (2*R*,4*S*)-4-hydroxy-2,5-dimethyl-3-oxohexanoic (**210**) acids, respectively [362,363], and two other cyclic peptides named Eudistomides A and B, which are derived from a Fijian ascidian *Eudistoma* sp. contain identical FAs (**211**) [364].

Sea squirts are sedentary and filter water containing plankton, microalgae, and bacteria. For sea squirts, the obligate cyanobacterial symbionts are *Prochloron* spp. Apparently, ascidian metabolites are organic molecules, including lipopeptides, which are synthesized by microorganisms associated with ascidians. Therefore, if we are talking about ascidian lipopeptides, then we mean organic metabolites isolated from the body of ascidians and nothing more.

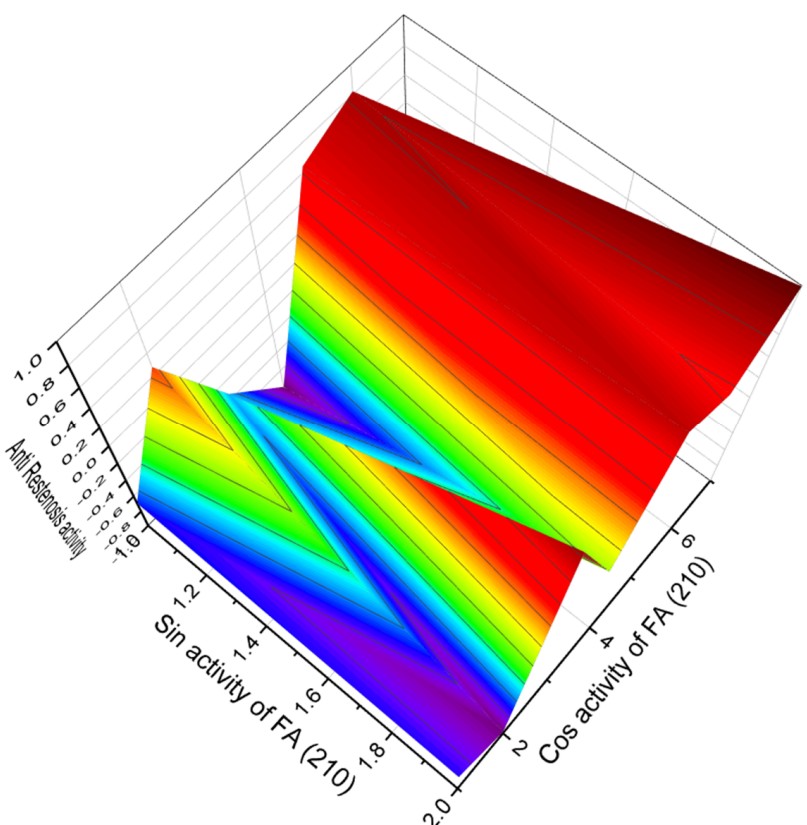

**Figure 47.** 3D graph showing the predicted and calculated FA (**208–210**) activities as an anti-restenosis agent. Recurrence, called restenosis or vasoconstriction, is rare. Various drugs, mechanical devices such as stents, genetic treatments such as gene transfer or stem cell infusion, or combinations of the above are commonly used to treat restenosis. For FA, this property is apparently described for the first time.

## 6. Fatty Acids Incorporated into Actinomycete and Fungal Lipopeptides

Marine fungi, fungal endophytes and fungi growing in other ecosystems synthesize linear and/or cyclic lipopeptides. These organisms are an inexhaustible source of new biologically active compounds. These compounds are unique because the aquatic environment requires many specific and potent biologically active molecules. Various lipopeptides have been discovered with a wide spectrum of biological activity, including antimicrobial, antitumor and antiviral activity, and toxins [365–368].

An Australian marine-derived fungus, *Acremonium* sp. (MST-MF588a), yielded a family of lipodepsipeptides, acremolides A–D [369]. 3,5,11-trihydroxy-2,6-dimethyl-dodecanoic FA (**212**, for structures see Figure 48, activity is shown in Table 19) was incorporated into acremolides A, C, and D, and 3,5-dihydroxy-2,6-dimethyl-11-oxododecanoic FA (**213**) was found in the structure of acremolide B.

As inhibitors of topoisomerases, the cyclic lipopeptides fusaristatins A and B were isolated from rice cultures of endophytic fungus *Fusarium* sp. YG-45. An unusual (8*E*,10*E*)-3-hydroxy-2,6,10,14-tetramethyl-7-oxoicosa-8,10-dienoic acid (**214**) was found in cyclic lipopeptides of fusaristatin A and B, as well as the linear lipopeptide YM 170320. Figure 49 shows the 3D graph of the predicted and calculated activity of this FA [370,371].

**Figure 48.** Unusual FA derived from fungal lipopeptides.

**Table 19.** Predicted biological activity of FA from fungal lipopeptides.

| No. | Predicted Biological Activity, Pa * |
|---|---|
| **212** | Hypolipemic (0.845); Antihypertensive (0.622); Lipid metabolism regulator (0.621) |
| **213** | Lipid metabolism regulator (0.791); Hypolipemic (0.778); Antihypertensive (0.643) |
| **214** | Acute neurologic disorders treatment (0.944); Lipid metabolism regulator (0.868) |
| **215** | Natural killer cell stimulant (0.785); Antineoplastic (0.785); Leukopoiesis stimulant (0.733) Acute neurologic disorders treatment (0.701); Immunosuppressant (0.697); Neuroprotector (0.664) |
| **216** | Natural killer cell stimulant (0.785); Antineoplastic (0.785); Leukopoiesis stimulant (0.733) Acute neurologic disorders treatment (0.701); Immunosuppressant (0.697); Neuroprotector (0.664) |

**Table 19.** *Cont.*

| No. | Predicted Biological Activity, Pa * |
|:---:|:---|
| **217** | Hypolipemic (0.807); Acute neurologic disorders treatment (0.722); Immunosuppressant (0.710) Leukopoiesis stimulant (0.687); Natural killer cell stimulant (0.685); Erythropoiesis stimulant (0.551) |
| **218** | Natural killer cell stimulant (0.795); Leukopoiesis stimulant (0.784); Hypolipemic (0.765) Immunosuppressant (0.702); Acute neurologic disorders treatment (0.696); Neuroprotector (0.684) |
| **219** | Hypolipemic (0.851); Acute neurologic disorders treatment (0.844); Anticonvulsant (0.733) Natural killer cell stimulant (0.715); Leukopoiesis stimulant (0.713) |
| **220** | Hypolipemic (0.851); Acute neurologic disorders treatment (0.844); Anticonvulsant (0.733) Natural killer cell stimulant (0.715); Leukopoiesis stimulant (0.713) |
| **221** | Natural killer cell stimulant (0.795); Leukopoiesis stimulant (0.784); Hypolipemic (0.765) Immunosuppressant (0.702); Acute neurologic disorders treatment (0.696); Neuroprotector (0.684) |
| **222** | Hypolipemic (0.828); Acute neurologic disorders treatment (0.774); Natural killer cell stimulant (0.746) Leukopoiesis stimulant (0.736); Immunosuppressant (0.726); Lipid metabolism regulator (0.685) |
| **223** | Lipid metabolism regulator (0.881); Hypolipemic (0.851); Leukopoiesis stimulant (0.822) Natural killer cell stimulant (0.795); Anti-hypercholesterolemic (0.728) |
| **224** | Lipid metabolism regulator (0.881); Hypolipemic (0.851); Leukopoiesis stimulant (0.822) Natural killer cell stimulant (0.795); Anti-hypercholesterolemic (0.728) |
| **225** | Platelet antagonist (0.800); Acute neurologic disorders treatment (0.701); Fibrinolytic (0.700) Natural killer cell stimulant (0.696); Erythropoiesis stimulant (0.619); Immunosuppressant (0.612) |
| **226** | Leukopoiesis stimulant (0.780); Antineoplastic (0.773); Hypolipemic (0.748) Natural killer cell stimulant (0.744); Lipid metabolism regulator (0.650) |
| **227** | Lipid metabolism regulator (0.809); Leukopoiesis stimulant (0.776); Hypolipemic (0.758) Antihypertensive (0.741); Natural killer cell stimulant (0.730); Antineoplastic (0.683) |
| **228** | Antidiabetic (0.781); Leukopoiesis stimulant (0.725); Bone diseases treatment (0.679) |

* Only activities with Pa > 0.5 are shown.

Rakicidins A–D are the 15-membered cytotoxic depsipeptides produced by the actinomycetes *Micromonospora* and *Streptomyces* [372,373], while rakicidin D was isolated from the culture broth of an actinomycete strain of the genus *Streptomyces* sp. MWW064 [374]. 3-Hydroxy-2,4,15-trimethylhexadecanoic acid (**215**) was found in rakicidin A, 3-hydroxy-2,4,16-trimethylheptadecanoic acid (**216**)—in rakicidin B, 3-hydroxy-2,4,6,8-tetramethyl-nonanoic acid (**217**)—in rakicidin C, and 3-hydroxy-2,4-dimethyldecanoic acid (**218**) was detected in rakicidin D.

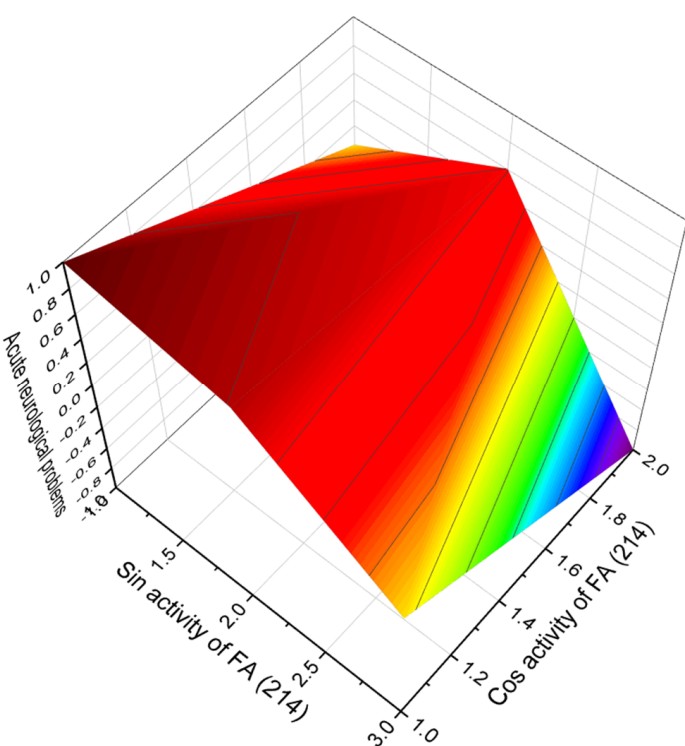

**Figure 49.** 3D graph showing the predicted and calculated activity of an acute neurological disorder agent, FA (**214**). This acid is part of the cyclic lipopeptides of fusaristatin A and B, as well as the linear lipopeptide YM-170320.

The cyclic depsipeptides tumescenamides A and B were isolated from the fermentation broth of a marine bacterium, *Streptomyces tumescens* YM23-260 [375], and tumescenamide C was detected in a culture broth of an actinomycete *Streptomyces* sp. KUSC F05 [376]. (2*S*,4*S*)-2,4-dimethylheptanoic acid (**219**) was found in tumescenamide A and C, and (2*S*,4*S*)-2,4,6-trimethylnonanoic acid (**220**) was isolated from tumescenamide B.

In a mixture of cultures of fungal *Emericella* sp. and the actinomycete *Salinispora arenicola*, two cyclic depsipeptides, emericellamides A and B, were found. It is known that *Salinispora arenicola* was isolated from marine sediments and exhibited weak cytotoxicity against human colon cancer cells HCT-116 [377]. (2*R*,3*R*,4*S*)-3-hydroxy-2,4-dimethyl-decanoic acid (**221**) found in emericellamide A, and (2*R*,3*R*,4*S*,6*S*)-3-hydroxy-2,4,6-trimethyldodecanoic acid (**222**)—in emericellamides B. The cyclic hexadepsipeptides arenamides A and B are characterized by a 19-membered macrocycle with six subunits—Phe, Ala, Val, Gly, Leu, were isolated from the marine actinomycete *S. arenicola*. These two compounds, having an aromatic amino acid phenyl alanine in the molecule, inhibited NO production in a dose-dependent manner (2–10 µM), besides displaying weak activity against HCT116 cells [378]. The arenamide A and B blocked TNF-induced activation in a dose- and t-dependent manner with IC$_{50}$ values of 3.7 and 1.7 µM respectively. Two different FA, (3*R*,4*R*)-3-hydroxy-4-methyldecanoic (**223**), and (3*S*,4*S*)-3-hydroxy-4-methyldecanoic (**224**), were isolated from arenamide A (A1) and arenamide B (A2), respectively.

Bicyclic depsipeptide antibiotics, salinamides A and B, with anti-inflammatory properties, were produced by fermentation of a specific marine actinomycete, a *Streptomyces* sp. (CNB-091) in saltwater-based media [379]. (2*S*,3*S*)-3-hydroxy-2,4-dimethylpentanoic acid (**225**) was present in both the bicyclic depsipeptides salinamides A and B.

Vinylamycin, a depsipeptide antibiotic, was isolated from the culture broth of a *Streptomyces* sp. It has showed antimicrobial activities against Gram-positive bacteria including methicillin-resistant *Staphylococcus aureus* [380]. 3-hydroxy-2-(2-hydroxyethyl)-4-methyl-decanoic acid (**226**) was found in vinylamycin. A cyclic depsipeptide antibiotic NA30851A useful for insecticides or microbicides, is manufactured by culturing *Streptomyces* sp. NA30851A (FERM P-16214) [381,382]. Two 2-hydroxy-3-methylpentanoic (**227**) and 3,4-dihydroxy-2,2-dimethyl-5-phenyl-pentanoic (**228**) FA were present in cyclic depsipeptide antibiotic NA 30851A.

A metabolite of antimycin family, JBIR-06, was isolated from *Streptomyces* sp. ML55, and it inhibited the expression of GRP78 induced by 2-deoxyglucose at the IC$_{50}$ value of 250 nM [383]. An actinomycete, *Streptomyces* sp. ML55, produced the antibiotic JBIR-52 and containing the FA 4-Hydroxy-2,2-dimethyl-3-oxo-nonanoic acid (**229**, for structure see Figure 50, and activity see in Table 20) was found in JBIR-06, and 4-hydroxy-2-methyl-3-oxo-nonanoic acid (**230**) was present in JBIR-52 [384].

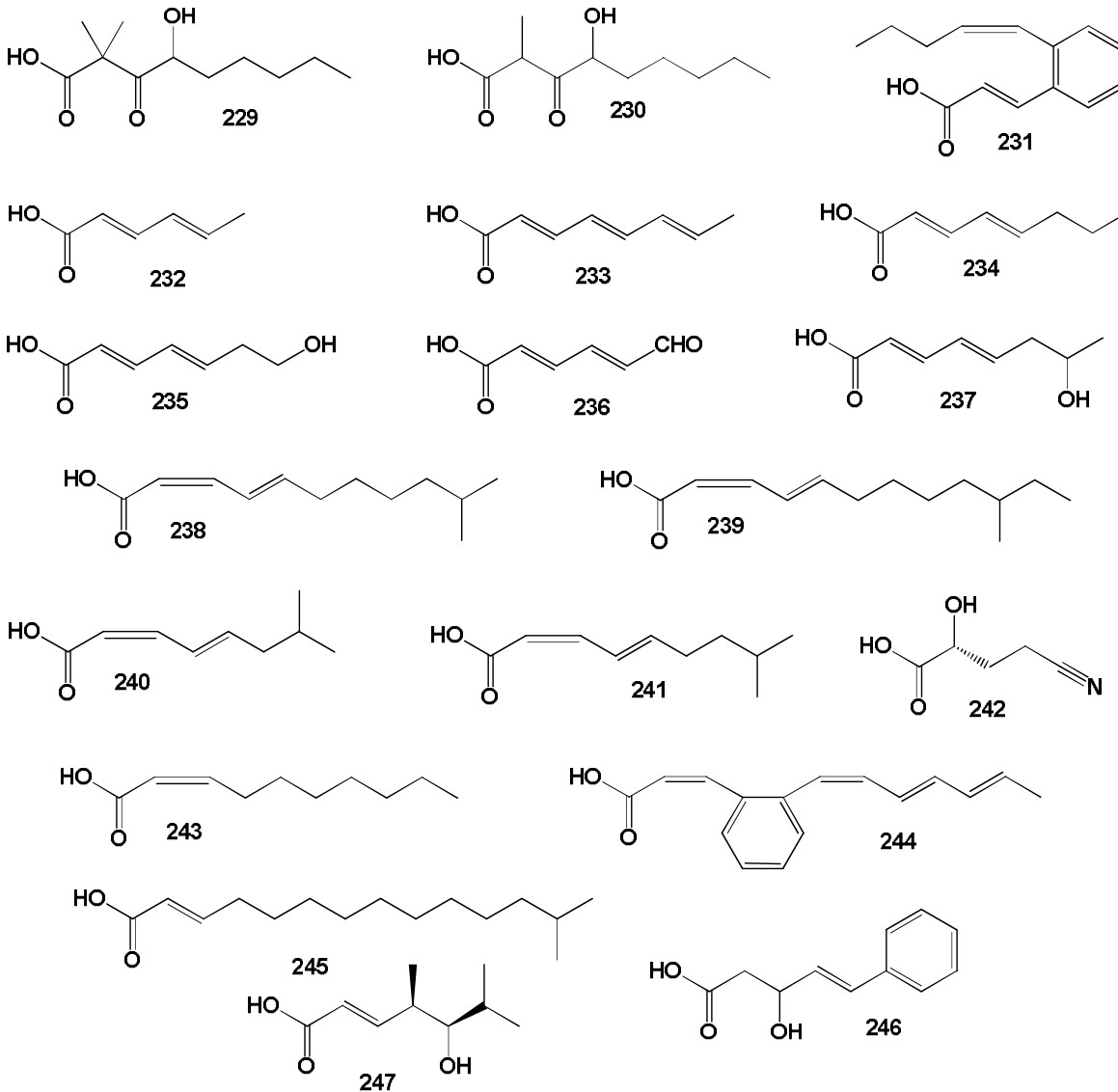

**Figure 50.** Saturated, unsaturated, and phenolic FA derived from fungal lipopeptides.

**Table 20.** Predicted biological activity of FA derived from fungal lipopeptides.

| No. | Predicted Biological Activity, Pa * |
|---|---|
| 229 | Lipid metabolism regulator (0.861); Leukopoiesis stimulant (0.840); Hypolipemic (0.685) |
| 230 | Fibrinolytic (0.760); Natural killer cell stimulant (0.746); Leukopoiesis stimulant (0.746) |
| 231 | Antiviral (Arbovirus) (0.834); Antimutagenic (0.827); Mucositis treatment (0.812) Preneoplastic conditions treatment (0.766); Lipid metabolism regulator (0.718) |
| 232 | Antiviral (Arbovirus) (0.861); Lipid metabolism regulator (0.844); Antiviral (Picornavirus) (0.776) |
| 233 | Antiviral (Arbovirus) (0.861); Lipid metabolism regulator (0.844); Antiviral (Picornavirus) (0.776) |
| 234 | Antiviral (Arbovirus) (0.947); Lipid metabolism regulator (0.884); Antiviral (Picornavirus) (0.782) |
| 235 | Antiviral (Arbovirus) (0.891); Lipid metabolism regulator (0.743); Antiviral (Picornavirus) (0.730) |
| 236 | Apoptosis agonist (0.886); Antineoplastic (0.763); Chemoprotective (0.700) |
| 237 | Antiviral (Arbovirus) (0.904); Lipid metabolism regulator (0.828); Antiviral (Picornavirus) (0.688) |
| 238 | Antiviral (Arbovirus) (0.886); Lipid metabolism regulator (0.874); Antiviral (Picornavirus) (0.744) |
| 239 | Lipid metabolism regulator (0.942); Anti-hypercholesterolemic (0.855); Hypolipemic (0.793) |
| 240 | Antiviral (Arbovirus) (0.853); Antiviral (Picornavirus) (0.760); Apoptosis agonist (0.731) |
| 241 | Antiviral (Arbovirus) (0.873); Lipid metabolism regulator (0.850); Antiviral (Picornavirus) (0.735) |
| 242 | Anti-ischemic, cerebral (0.910); Leukopoiesis stimulant (0.708); Antidiabetic (0.698) |
| 243 | Antiviral (Arbovirus) (0.944); Lipid metabolism regulator (0.882); Antiviral (Picornavirus) (0.766) |
| 244 | Anti-inflammatory (0.785); Antineoplastic (0.782); Preneoplastic conditions treatment (0.661) |
| 245 | Lipid metabolism regulator (0.846); Anti-hypercholesterolemic (0.827); Atherosclerosis treatment (0.538) |
| 246 | Lipid metabolism regulator (0.943); Hypolipemic (0.860); Anti-hypercholesterolemic (0.844) |
| 247 | Antineoplastic (0.844); Hypolipemic (0.782); Lipid metabolism regulator (0.699) |

* Only activities with Pa > 0.5 are shown.

The depsipeptides WS9326A, WS9326C, WS9326D, and WS9326E were detected in a culture of *Streptomyces* sp. 9078 [385]. Phenyl-containing (*E*)-3-(2-((*Z*)-pent-1-en-1-yl)-phenyl)-acrylic acid (**231**) was found in all depsipeptides, and other antibiotics with unsaturated fatty acids were obtained from Streptomyces hawaiiensis culture extracts [386]. Thus, antibiotic A 54556A contains (2*E*,4*E*)-hexa-2,4-dienoic acid (**232**), antibitotic A 54556B contains (2*E*,4*E*,6*E*)-octa-2,4,6-trienoic acid (**233**), and four same acyl depsipeptides contain (2*E*,4*E*)-octa-2,4-dienoic (**234**), (2*E*,4*E*)-7-hydroxyhepta-2,4-dienoic (**235**), (2*E*,4*E*)-6-oxohexa-2,4-dienoic (**236**), and (2*E*,4*E*)-7-hydroxyocta-2,4-dienoic acids (**237**), respectively.

Structurally and functionally similar lipodepsipeptide antibiotics, enduracidin and ramoplanin, have been found in cultivated cultures of *Streptomyces fungicidicus* B5477 [387–390]. (2*Z*,4*E*)-10-Methylundeca-2,4-dienoic acid (**238**) was detected in the structures of enduracidin A, C, D, F and (2*Z*,4*E*)-10-methyldodeca-2,4-dienoic acid (**239**) was found in enduracidin B, E, and G.

*Streptomyces macrosporeus* ATCC 21 strain produces janiemycin as the main peptide antibiotic, which serves as a bactericidal ointment against *Streptococcus pyogenes* C203 and *Diplococcus pneumoniae* [388–391]. Janiemycin and ramoplanin A1 contain (2*E*,4*E*)-octa-2,4-dienoic acid (**234**), ramoplanin contains A2-(2*Z*,4*E*)-7-methylocta-2,4-dienoic acid (**240**), and ramoplanin A3 contains (2*Z*,4*E*)-8-methylnona-2,4-dienoic acid (**241**) [390].

It is known that the endophytic fungus *Pestalotiopsis* sp. produces the cyclopeptolide antibiotic HUN-7293 pesthivin and anti-HIV agent pesthivin DM, which are used to treat chronic inflammatory diseases and inhibit VCAM-1 expression on activated endothelial cells [392–394]. All antibiotics contain the same unusual (*R*)-4-cyano-2-hydroxybutanoic acid (**242**).

The lipopeptides rotihibin A and B, known as plant growth inhibitors, were obtained from *Streptomyces graminofaciens* 3C02 filtrate [395], and both contain (*Z*)-dec-2-enoic acid (**243**). The predicted and calculated antiviral activity of unsaturated FA (**234**, **237** and **243**) is shown in Figure 51.

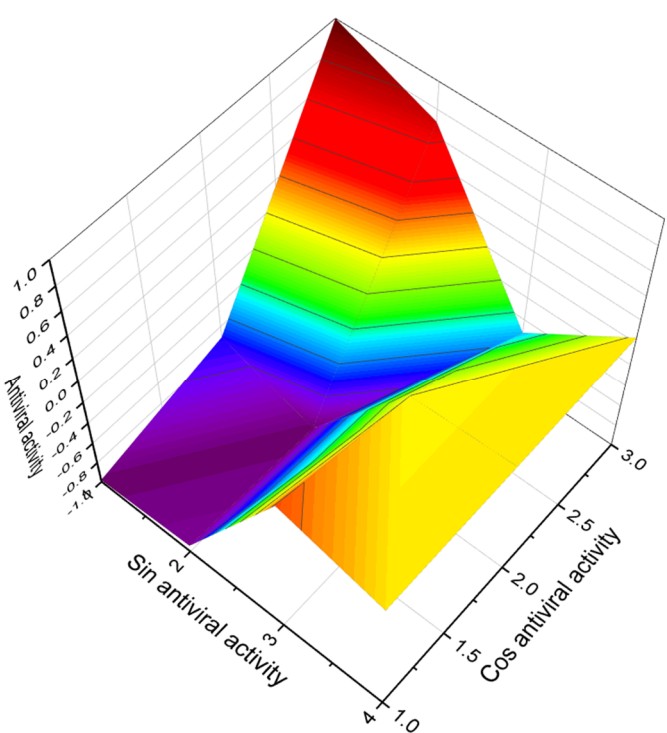

**Figure 51.** 3D graph showing the predicted and calculated antiviral activity of unsaturated FA (**234**, **237** and **243**) with over 90% confidence. FA (**234** and **237**) are found in the depsipeptide which is synthesized by *Streptomyces hawaiiensis*, and the acid (**243**) is present in the depsipeptide which is produced by *Streptomyces graminofaciens*.

A cultivated endophytic fungus from *Copris tripartitus* is a producer of coprisamides A and B, which have significant activity for the induction of quinone reductase [396]. Both cyclic peptides contain (Z)-3-(2-((1Z,3E,5E)-hepta-1,3,5-trien-1-yl)-phenyl)-acrylic acid (**244**). A lipopeptide antibiotic related to amphomycin and named laspartomycin has a side chain in the form of (*E*)-13-methyltetradec-2-enoic acid (**245**) [397].

The Floridian marine sediment-derived fungus *Microascus* sp. EGM-556 produces a hybrid biosynthetic cyclodepsipeptide [398], which was structurally identical to turnagaino-

lide A. It is known that turnagainolide A is produced in culture by a *Bacillus* sp. [399]. These cyclodepsipeptides, EGM-556 and turnagainolide A, contain (*E*)-3-hydroxy-5-phenylpent-4-enoic acid (**246**).

The actinomycete *Streptomyces pristinaespiralis* synthesizes a mixture of antibiotics called pristinamycin IA and IIA in a ratio of 30:70 [400,401]. Both antibiotics are known to inhibit protein synthesis elongation, exhibiting powerful bacteriostatic activity [402,403]. (4*R*,5*R*,*E*)-5-hydroxy-4,6-dimethylhept-2-enoic (**247**) FA was incorporated into both antibiotics IA and IIA.

Unusual guanidine-containing fatty acids (**248–251**) are produced by some fungus species. Monoamidocin, *N*-[(*S*)-5-guanidino-2-hydroxypentanoyl]-l-phenylalanine, is a dipeptide analogue has been isolated from *Streptomyces* sp. NR 0637. Monoamidocin inhibits the binding of fibrinogen to GP IIb/IIIa receptors [404]. This compound contains (*S*)-5-guanidino-2-hydroxypentanoic acid (**248**, for structure see Figure 52, and activity is shown in Table 21). The monoamidocin analogue was shown to have a 10-fold increase in activity and contains (*R*)-5-guanidino-2-hydroxypentanoic acid (**249**). The fusaricidins A, B, C and D, depsipeptide antibiotics, have been isolated as minor components from the culture broth of *Bacillus polymyxa* KT-8 which was obtained from the rhizosphere of garlic suffering from basal rot caused by *Fusarium oxysporum*. The fusaricidins B, C and D are active against fungi and Gram-positive bacteria, as well as fusaricidin A [405]. All the fusaricidins contain 15-guanidino-3-hydroxypentadecanoic acid (**250**).

**Figure 52.** Unusual FA derived from fungi and fungal endophyte lipopeptides.

**Table 21.** Predicted biological activity of FA from lipopeptides of fungi.

| No. | Predicted Biological Activity, Pa * |
|---|---|
| 248 | Anti-ischemic, cerebral (0.919); Mucositis treatment (0.913); Lipid metabolism regulator (0.885) Antithrombotic (0.745); Platelet antagonist (0.685); Antihypertensive (0.555) |
| 249 | Anti-ischemic, cerebral (0.919); Mucositis treatment (0.913); Lipid metabolism regulator (0.885) Antithrombotic (0.745); Platelet antagonist (0.685); Antihypertensive (0.555) |
| 250 | Lipid metabolism regulator (0.923); Mucositis treatment (0.910); Antithrombotic (0.805) Anti-ischemic, cerebral (0.801); Platelet antagonist (0.751); Fibrinolytic (0.684) |
| 251 | Mucositis treatment (0.960); Lipid metabolism regulator (0.869); Platelet antagonist (0.836) Antithrombotic (0.773); Anti-ischemic, cerebral (0.740); Fibrinolytic (0.717) |
| 252 | Antineoplastic (0.844); Antiviral (Arbovirus) (0.790); Antiviral (Picornavirus) (0.659) |
| 253 | Antihypertensive (0.962); Anti-Helicobacter pylori (0.839); Chemoprotective (0.718) Antineoplastic (myeloid leukemia) (0.707); Apoptosis agonist (0.674); Antineoplastic (0.656) |
| 254 | Anti-Helicobacter pylori (0.744); Antiviral (Arbovirus) (0.715); Antiviral (Picornavirus) (0.547) |
| 255 | Hypolipemic (0.795); Antineoplastic (0.791); Antiviral (Arbovirus) (0.742) Preneoplastic conditions treatment (0.740); Lipid metabolism regulator (0.739) |
| 256 | Lipid metabolism regulator (0.902); Anti-hypercholesterolemic (0.871) Hypolipemic (0.812); Atherosclerosis treatment (0.603) |
| 257 | Antiviral (Arbovirus) (0.890); Antiviral (Picornavirus) (0.817); Mucositis treatment (0.743) |
| 258 | Hypolipemic (0.845); Antihypertensive (0.622); Lipid metabolism regulator (0.621) Atherosclerosis treatment (0.590); Antiprotozoal (Coccidial) (0.569); Antibacterial (0.562) |
| 259 | Lipid metabolism regulator (0.874); Anti-hypercholesterolemic (0.785); Hypolipemic (0.707) |
| 260 | Lipid metabolism regulator (0.942); Anti-hypercholesterolemic (0.855); Hypolipemic (0.793) Atherosclerosis treatment (0.635); Multiple sclerosis treatment (0.521) |
| 261 | Antiviral (Arbovirus) (0.927); Lipid metabolism regulator (0.887); Antiviral (Picornavirus) (0.669) |
| YM-170320 | Antifungal (0.867); Antibacterial (0.744); Antineoplastic (0.711); Chemoprotective (0.594) |
| 262 | Acute neurologic disorders treatment (0.944); Lipid metabolism regulator (0.868) Antineoplastic (0.814); Antifungal (0.829); Antibacterial (0.698) |

* Only activities with Pa > 0.5 are shown.

Eulicin is a potent antibiotic against a broad range of Gram-positive and Gram-negative bacteria which was isolated from a *Streptomyces* sp. [406]. More recently, eulicin and its related analogues, as a muscarinic receptor antagonist, have also been isolated from a *Streptomyces* strain SCC 2268 [407]. Recently, it has been shown that eulicin inhibits human immunodeficiency virus infection and replication in a dose-dependent manner [408]. Both compounds contain 9-guanidinononanoic acid (**251**).

It is known that depsipeptides and streptogramins are potent drugs against numerous highly resistant pathogens and are used for human treatment, and one of them is virginiamycin [409,410]. This depsipeptide was obtained by fermentation of a culture of *Streptomyces* sp. G-89 [411], and contains (4*R*,5*S*,*E*)-5-hydroxy-4,6-dimethylhept-2-enoic acid (**252**).

Pterulamides I-VI are linear lipopeptides that were isolated from the fruiting bodies of a Malaysian fungus *Pterula* sp. Pterulamides I and IV are cytotoxic against the P388 cell line with IC$_{50}$ values of 0.55 and 0.95 μM/mL (0.79 and 1.33 μM), respectively [412]. Two sulfur-containing acids, (*E*)-3-(methylsulfinyl)-acrylic (**253**) and (*E*)-3-(methylthio)-acrylic (**254**) acids were present in pterulamides I–VI.

The arylomycins are lipopeptide antibiotics, and they were detected in the culture filtrate and mycelium extracts of *Streptomyces* sp. Tu6075b. The isolated antibiotics demonstrate antimicrobial activity against Staphylococcus aureus, Escherichia coli, and *Pseudomonas aeruginosa* [413]. Two monoenoic fatty acids, (*E*)-2,4-dimethylhept-2-enoic acid (**255**) and (*E*)-11-hydroxy-14-methylpentadec-2-enoic acid (**256**), were isolated from arylomycin D.

Analysis of a fermentation broth of Streptomyces RK-1051 showed that the solution contains two depsipeptides that inhibit the growth of Staphylococcus pyogenes [414]. Both depsipeptides contain (2*E*,4*E*,6*E*,8*E*,10*E*)-dodeca-2,4,6,8,10-pentaenedioic acid (**257**). This fatty acid also was found in enopeptin A, which was isolated from *Streptomyces griseus* and showed antimicrobial activity against several microorganisms, and enopeptin A exhibited anti-bacteriophage activity [415,416]. A cyclic depsilipopeptide, colisporifungin, was isolated from a fungal isolate of *Fusarium* sp. and contained 3,5,11-trihydroxy-2,6-dimethyl-dodecanoic acid (**258**) [417].

The actinomycete *Streptomyces fungicidicus* has yielded three enduracidin analogues. (2*Z*,4*E*)-10-methylundeca-2,4-dienoic acid (**259**) was found in enduracidin and monodeschloro-enduracidin A, and (2*Z*,4*E*)-10-methyldodeca-2,4-dienoic acid (**260**) was present in mono-deschloroenduracidin B [418]. Enamidonin, a cyclic lipopeptide antibiotic, has been isolated from a culture broth of *Streptomyces* sp. 91–75 [419]. (2*E*,4*E*,9*E*)-13-hydroxytetradeca-2,4,9-trienoic acid (**261**) was found in structure of enamidonin.

Strain *YL-03706F*, a mutant of *Candida tropicalis* pK233, has produced a lipopeptide antibiotic designated YM-170320 (for structure see Figure 52) [420,421]. (8*E*,10*E*)-3-Hydroxy-2,6,10,14-tetramethyl-7-oxoicosa-8,10-dienoic acid (**262**) was included into the lipopeptide structure. Comparative analysis of the biological activity of this linear lipopeptide YM-170320 and FA (**262**) included in its structure showed that this lipopeptide demonstrates antifungal and antibacterial activity, and the same activity is demonstrated by FA (see Table 21).

Analyzing the activity of FA (**248–262**), which was obtained using PASS, acid (**253**) showed antihypertensive properties with a confidence level of more than 96%. Figure 53 presents the 3D graph which shows the predicted and calculated activity of this FA.

The fermentation broth actinomycete *Streptomyces* sp. contains two cyclic lipopeptides K97-0239 A and B, both having (2*E*,4*E*)-13-hydroxytetradeca-2,4-dienoic acid (**263**, for structure see Figure 54, and biological activity see in Table 22) [422].

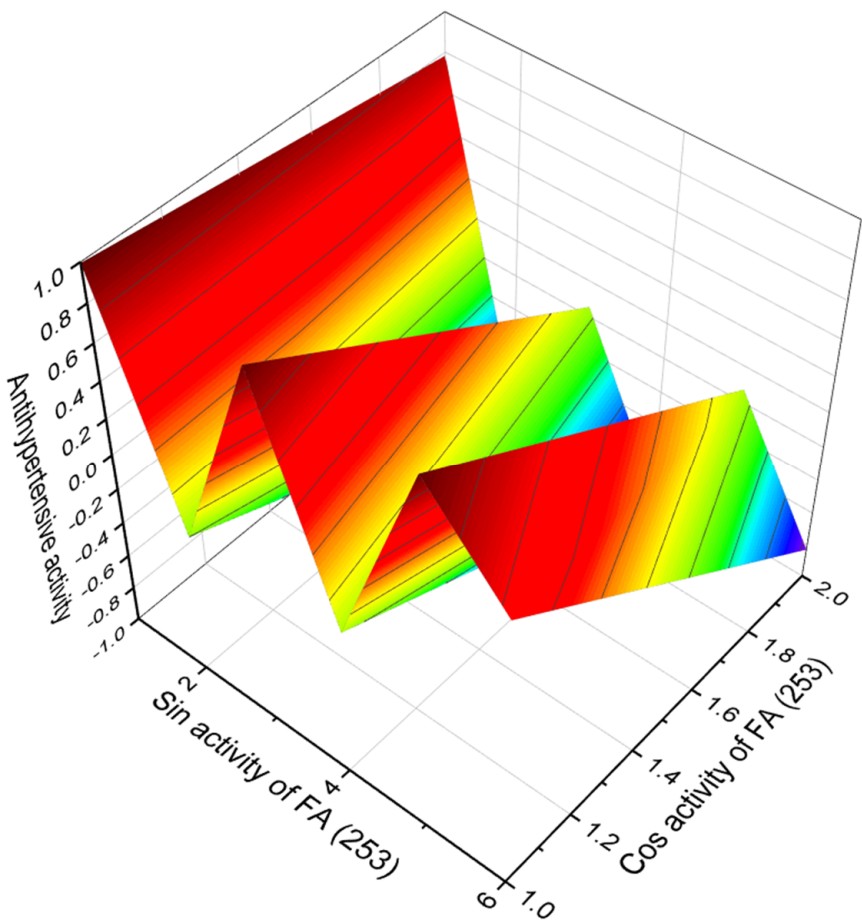

**Figure 53.** 3D graph shows the predicted and calculated antihypertensive activity of a rare sulfur-containing FA (**253**). The antihypertensive property of this acid appears to be related to the presence of sulfur monoxide. It is known that many antihypertensive drugs, such as indapamide, chlorthalidone, metalozone, xipamide or clopamide, contain sulfur dioxide molecules.

Ulleungamides A and B, cyclic depsipeptides, were obtained from cultures of *Streptomyces* sp. Ulleungamide A with (*S*)-2-isopropylsuccinic acid (**264**) displayed growth inhibitory activity against *Staphylococcus aureus* and *Salmonella typhimurium* without cytotoxicity [423].

More than 40 years ago, Eli Lilly published the isolation of a similar depsipeptide antibiotic A54556, a complex of eight depsipeptide factors A-H, which was produced by aerobic fermentation of *Streptomyces hawaiiensis* NRRL 15010 [424]. The depsipeptides had promising in vitro activity against enterococci and streptococci but only moderate in vitro potency against staphylococci. A complex antibiotic A54556 depsipeptide factor contains several acids: A and F a (2*E*,4*E*,6*E*)-octa-2,4,6-trienoic acid (**265**), B-(2*E*,4*E*)-6-hydroxyhexa-2,4-dienoic acid (**266**), C and D-(2*E*,4*E*)-hexa-2,4-dienoic acid (**267**), and H-(2*E*,4*E*)-octa-2,4-dienoic acid (**268**).

The peptide–polyketide glycoside totopotensamide A and its aglycone totopotensamide B were detected in the fermentation broth of *Streptomyces* sp. 1053U. This Actinomycete was isolated from the gastropod mollusc, *Lienardia totopotens,* collected in the Philippines (Mactan Is., Cebu) [425]. Similar to the glycolipid, (**269**) and (8*S*,9*R*,10*S*,11*R*,12*S*)-9,11,12-trihydroxy-4,6,8,10-tetramethyl-3-oxotri-decanoic (**270**) FA were found in totopotensamide A and B, respectively.

**Figure 54.** Rare and unusual FA derived from fungal lipopeptides.

**Table 22.** Predicted biological activity of FA derived from lipopeptides of fungi.

| No. | Predicted Biological Activity, Pa * |
|---|---|
| 263 | Antiviral (Arbovirus) (0.927); Lipid metabolism regulator (0.887); Antiviral (Picornavirus) (0.669) |
| 264 | Antitoxic (0.733); Leukopoiesis stimulant (0.730); Preneoplastic conditions treatment (0.721) |
| 265 | Antiviral (Arbovirus) (0.861); Lipid metabolism regulator (0.844); Antiviral (Picornavirus) (0.776) |
| 266 | Antiviral (Arbovirus) (0.860); Antiviral (Picornavirus) (0.751); Antidiabetic (0.713) |

**Table 22.** *Cont.*

| No. | Predicted Biological Activity, Pa * |
|---|---|
| 267 | Antiviral (Arbovirus) (0.861); Lipid metabolism regulator (0.844); Antiviral (Picornavirus) (0.776) |
| 268 | Antiviral (Arbovirus) (0.947); Lipid metabolism regulator (0.884); Antiviral (Picornavirus) (0.782) |
| 269 | Antineoplastic (0.902); DNA synthesis inhibitor (0.675); Angiogenesis inhibitor (0.668) |
| 270 | Antineoplastic (0.872); Apoptosis agonist (0.653); Antiarthritic (0.611) |
| 271 | Hypolipemic (0.808); Anti-hypercholesterolemic (0.695); Lipid metabolism regulator (0.691) |
| 272 | Lipid metabolism regulator (0.887); Atherosclerosis treatment (0.690); Hypolipemic (0.647) |
| 273 | Lipid metabolism regulator (0.886); Hypolipemic (0.788); Anti-hypercholesterolemic (0.755) |
| 274 | Lipid metabolism regulator (0.886); Hypolipemic (0.788); Anti-hypercholesterolemic (0.755) |
| 275 | Apoptosis agonist (0.886); Antineoplastic (0.763); Chemoprotective (0.700) |
| 276 | Apoptosis agonist (0.781); Antimutagenic (0.777); Preneoplastic conditions treatment (0.600) |
| 277 | Natural killer cell stimulant (0.795); Leukopoiesis stimulant (0.784); Antineoplastic (0.778) |
| 278 | Natural killer cell stimulant (0.795); Leukopoiesis stimulant (0.784); Antineoplastic (0.778) |
| 279 | Leukopoiesis stimulant (0.802); Natural killer cell stimulant (0.769); Immunosuppressant (0.686) |
| 280 | Lipid metabolism regulator (0.929); Hypolipemic (0.786); Lymphocytopoiesis inhibitor (0.726) |
| 281 | Antidiabetic (0.763); Antihypoxic (0.736); Natural killer cell stimulant (0.570) |
| 282 | Myasthenia Gravis treatment (0.962); Cell adhesion molecule inhibitor (0.863); Antidiabetic (0.757) |
| 283 | Angiogenesis stimulant (0.915); Lipid metabolism regulator (0.913); Myasthenia Gravis treatment (0.779) Cell adhesion molecule inhibitor (0.745); Antidiabetic (0.664) |
| 284 | Angiogenesis stimulant (0.914); Myasthenia Gravis treatment (0.844) Cell adhesion molecule inhibitor (0.788); Antidiabetic (0.700) |
| 285 | Angiogenesis stimulant (0.915); Lipid metabolism regulator (0.913); Myasthenia Gravis treatment (0.779) Cell adhesion molecule inhibitor (0.745); Antidiabetic (0.664) |
| 286 | Antihypertensive (0.824); DNA intercalator (0.707); Antidiabetic (0.692) |
| 287 | Leukopoiesis stimulant (0.763); Natural killer cell stimulant (0.721); Antidiabetic symptomatic (0.659) |
| 288 | Myasthenia Gravis treatment (0.962); Cell adhesion molecule inhibitor (0.863); Antidiabetic (0.757) |

* Only activities with Pa > 0.5 are shown.

(*E*)-6-Hydroxy-4-methylhex-2-enoic acid (**271**) was detected in several lipopeptides and cyclic peptides, called asperchrome A, B1, B2, B3, C, D1, D2, D3, and rerrirubin. These antibiotics are produced by *Aspergillus ochraceous*; fusarinine C, and N,N′N″-triacetylfusarinin C are produced by *Aspergillus fumigatus*, *A.nidulans* and *Fusarium cubense*; coprogen and dimerumic acid are produced by *Aspergillus terreus*; fusarinine C and asperchrome F1 is produced by fungus *Aureobasidium pullulans* and *Penicillium chrysogenum*; basidiochrome is produced by *Ceratobasidium cornigerum*; basidiochrome, ferrirhodin, and *Ceratobasidium globisporum* [426].

The unique cyclic peptide thioviridamide is an apoptosis inducer found in the fermentation broth of *Streptomyces olivoviridis* with 2-hydroxy-2-methyl-4-oxopentanoic acid (**272**) [427]. The fermentation broths of the fungal strains *Chalara* sp. no. 22210 and *Tolypocladium parasiticum* 16616 contain the antifungal lipopeptides FR227673 and FR190293, with 10,12-dimethyltetradecanoic (**273**), and 12,14-dimethylhexadecanoic (**274**) FA, respectively [428].

A tripeptide, pre-sclerotiotide F, was isolated from a marine sediment-derived fungus, *Aspergillus insulicola*, and has showed cytotoxicity against selected cancer cells *in vitro*. The effects of pre-sclerotiotide F and sclerotiotide F on LPS-induced NF-κB and iNOS expression were also reported [429]. (2*E*,4*E*)-6-Oxohexa-2,4-dienoic acid (**275**) was present in pre-sclerotiotide F and sclerotiotide F.

The marine mangrove endophytic unidentified fungus from the South China Sea produced xyloallenoide A with (*E*)-3-(4-(buta-2,3-dien-1-yloxy)-phenyl)-acrylic acid (**276**) [430].

The marine fungus *Hypoxylon oceanicum* LL-15G256 is a producer of several lipodepsipeptides with antifungal activity. Compound 15G256γ contains (2*S*,3*S*,4*R*)-3-hydroxy-2,4-dimethyl-dodecanoic (**277**), 15G256d-(2*R*,3*R*,4*R*)-3-hydroxy-4-(hydroxymethyl)-2-methyl-dodecanoic (**278**), and 15G256e-(2*S*,3*S*,4*R*)-3-hydroxy-2,4-dimethyldecanoic (**279**) acids, respectively [431,432], and butanone extracts of another marine-derived fungus *Beauveria feline* with cytotoxic and anti-tuberculosis activity contains destruxin E chlorohydrin and pseudodestruxin C with (2S,4R)-5-chloro-2,4-dihydroxypentanoic acid (**280**) [433].

The fungal endophyte *Metarhizium anisopliae* produces several destruxins A, B, and E (DA, DB and DE). Destruxin-A4 chlorohydrin and destruxin Ed1 contain 2-hydroxy-3-((*S*)-oxiran-2-yl)-propanoic acid (**281**) [434–436].

The antibiotic E-64 is a thiol protease inhibitor, and its producer is *Aspergillus japonicus* TPR-64, which contains (2*R*,3*R*)-oxirane-2,3-dicarboxylic acid (**282**) [437,438] and a lipopeptide called cystargamide has been identified in the fermentation broth of the actinomycete *Kitasatospora cystarginea* and contains 3-heptyloxirane-2-carboxylic acid (**283**) [439].

The acidic lipopeptides produced by *Streptomyces* sp. such as CDA1b, CDA2a, CDA2b, CDA3a, CDA3b, CDA4a and CDA4b contain 3-pentyloxirane-2-carboxylic acid (**284**), and CDA1 and CDA2 contain 3-propyloxirane-2-carboxylic acid (**285**) [440–442]. The antitumor antibiotic carzinophilin A was derived from *Streptomyces sahachiroi* and azinomycin B was found in *Streptomyces griseofuscus* S42227, and both lipopeptides contain a 2-hydroxy-2-((*S*)-2-methyloxiran-2-yl)-acetic acid (**286**) [443]. The *Streptomyces* sp. strain associated with fungus-growing termites is a producer of microtermolide A, which contains a (2*S*,3*R*,4*R*)-3-hydroxy-2-(2-hydroxyethyl)-4-methylheptanoic acid (**287**) [444]. Oxirane-2,3-dicarboxylic acid (**288**) is part of many antibiotics that produce different fungal species, and it found in: antibiotic TMC 52A, TMC 52B, TMC 52C, rexostatine, cathestatin A, B, C, antibiotic AM 4299B, antibiotics WF 14861A, 14865A, 14865B, and antibiotic 460B [445].

Very interesting data were obtained in the analysis of the biological activity of FA (**263–288**). Acids containing the epoxy group **282**, **283**, **284**, **285** and **288** have been shown to treat Myasthenia Gravis with a confidence level of 78 to 96%. This is a rare property that epoxy FA exhibit (**282**, **283** and **284**), and their 3D activity is shown in Figure 55.

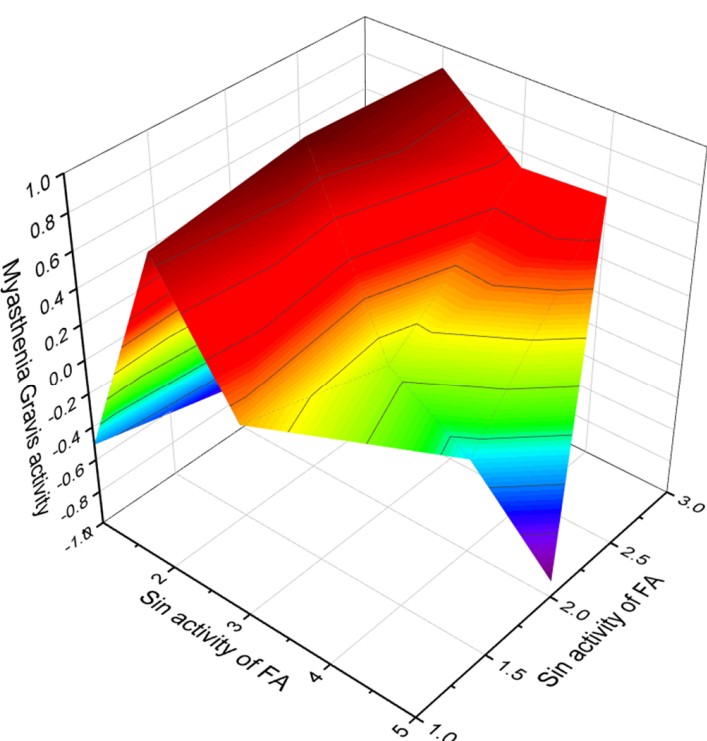

**Figure 55.** 3D graph showing the predicted and calculated Myasthenia Gravis activity of epoxy FA (**282**, **283** and **284**) at 78 to 96% confidence. *Aspergillus japonicus* TPR-64 synthesized acid (**282**), the actinomycete *Kitasatospora cystarginea* produces acid (**283**), and the fungus *Streptomyces* produces acid (**284**).

Pyrane-containing FA (**289–301**, for structures see Figure 56, and biological activity see in Table 23) have been found in the structures of many lipopeptides. Thus, two hexadepsipeptides were detected in a fermentation broth of *Streptomyces nobilis* JCM4274. Isolated hexadepsipeptides have shown an IC$_{50}$ of 30 nM against human lung cancer NCI-H358 cells [446], and both compounds contained (**289**) FA. A fermentation broth of *Streptomyces* sp. contained a cyclic hexadepsipeptide antibiotic GE3 A and a linear lipopeptide GE3 B, which had the same acid (**290**, see 3D graph in Figure 57) [447]. Polyoxypeptins A and B with FA (**291**) are potent apoptosis-inducing peptides and were detected in the culture broth of *Streptomyces* sp. [448].

The hexadepsipeptide antibiotics named aurantimycins A, B, and C with FA (**292**) were detected in the mycelium of *Streptomyces aurantiacus* JA4570 [449,450], and another actinomycete, *Streptomyces flavidovirens*, produced a cyclic hexadepsipeptide antibiotic, citropeptin, and contained pyrane-containing acid (**293**).

The hexadepsipeptide antibiotic, azinothricin with (**294**) FA was detected in extracts of the culture filtrate of *Streptomyces* sp. X-14950 [451], and a culture of the *Streptomyces karnatakensis* contained a cyclic hexadepsipeptide antibiotic designated A83586C with (**295**) FA [452].

Glyco-hexadepsipeptide-polyketide with FA (**296**) named mollemycin A is produced by a marine-derived *Streptomyces* sp. (CMB-M0244), which was isolated from a sediment collected off South Molle Island (Queensland) [453]. The cyclic hexadepsipeptide antibiotics which contained FA (**297**, 3D graph of activity sees in Figure 57) were obtained from a fermentation broth of *Streptomyces* species (PM0895172/MTCC 684) and showed antitumor activity [454].

**Figure 56.** Unique, rare, and unusual FA derived from fungal and bacterial lipopeptides.

**Table 23.** Predicted biological activity of FA of fungal lipopeptides.

| No. | Predicted Biological Activity, Pa * |
|-----|-------------------------------------|
| **289** | Antibiotic Glycopeptide-like (0.908); Antineoplastic (0.877); Apoptosis agonist (0.798) |
| **290** | Antineoplastic (0.942); Apoptosis agonist (0.897); Antibiotic Glycopeptide-like (0.785) |
| **291** | Antibiotic Glycopeptide-like (0.892); Antineoplastic (0.850); Antiprotozoal (Plasmodium) (0.834) |
| **292** | Antineoplastic (0.890); Antibiotic Glycopeptide-like (0.883); Apoptosis agonist (0.802) |

**Table 23.** *Cont.*

| No. | Predicted Biological Activity, Pa * |
|---|---|
| **293** | Antineoplastic (0.942); Apoptosis agonist (0.897); Antibiotic Glycopeptide-like (0.785) |
| **294** | Antineoplastic (0.913); Apoptosis agonist (0.882); Antibiotic Glycopeptide-like (0.771) |
| **295** | Antineoplastic (0.938); Apoptosis agonist (0.881); Antibiotic Glycopeptide-like (0.769) |
| **296** | Antibiotic Glycopeptide-like (0.813); Antineoplastic (0.800); Apoptosis agonist (0.684) |
| **297** | Antibiotic Glycopeptide-like (0.908); Antineoplastic (0.877); Apoptosis agonist (0.798) |
| **298** | Antibiotic Glycopeptide-like (0.879); Antineoplastic (0.857); Apoptosis agonist (0.803) |
| **299** | Antineoplastic (0.942); Apoptosis agonist (0.897); Antibiotic Glycopeptide-like (0.785) |
| **300** | Antibiotic Glycopeptide-like (0.906); Antineoplastic (0.865); Apoptosis agonist (0.763) |
| **301** | Antineoplastic (0.913); Antibiotic Glycopeptide-like (0.782); Apoptosis agonist (0.779) |
| **302** | Antidiabetic symptomatic (0.732); Leukopoiesis stimulant (0.671); Multiple sclerosis treatment (0.663) |
| **303** | Natural killer cell stimulant (0.739); Antidiabetic symptomatic (0.733); Leukopoiesis stimulant (0.687) |
| **304** | Antiviral (Arbovirus) (0.927); Lipid metabolism regulator (0.887); Antiviral (Picornavirus) (0.669) |
| **305** | Lipid metabolism regulator (0.881); Hypolipemic (0.851); Anti-hypercholesterolemic (0.728) |
| **306** | Lipid metabolism regulator (0.819); Natural killer cell stimulant (0.797); Hypolipemic (0.759) |
| **307** | Lipid metabolism regulator (0.919); Anti-hypercholesterolemic (0.803); Hypolipemic (0.763) |
| **308** | Anti-hypercholesterolemic (0.897); Lipid metabolism regulator (0.819); Antimutagenic (0.784) |
| **309** | Anti-inflammatory (0.842); Analgesic (0.766); Leukopoiesis stimulant (0.584) |
| **310** | Anti-inflammatory (0.814); Analgesic (0.746); Erythropoiesis stimulant (0.611) |

* Only activities with Pa > 0.5 are shown.

The cyclic hexadepsipeptide named pipalamycin with the rare FA (**298**) was isolated from a culture filtrate of *Streptomyces* sp. ML297-90F8 as an apoptosis-inducing agent [455].

Variapeptin and citropeptin were found to be hexadepsipeptide antibiotics produced by *Streptomyces variabilis* and *Streptomyces flavidovirens*, respectively. Both antibiotics were structurally related to azinothricin and A83586C, respectively. A culture of *Streptomyces variabilis* was also to produce a variapeptin. This antibiotic was active against Gram-positive bacteria and showed cytotoxic activity against mammalian cells [456,457], and acid (**299**) was present in variapeptin. Two antibacterial cyclic hexadepsipeptides named oleamycin A and B were detected in *Streptomyces* sp. [458] and contain pyrane-containing FA (**300**, see 3D graph in Figure 58)).

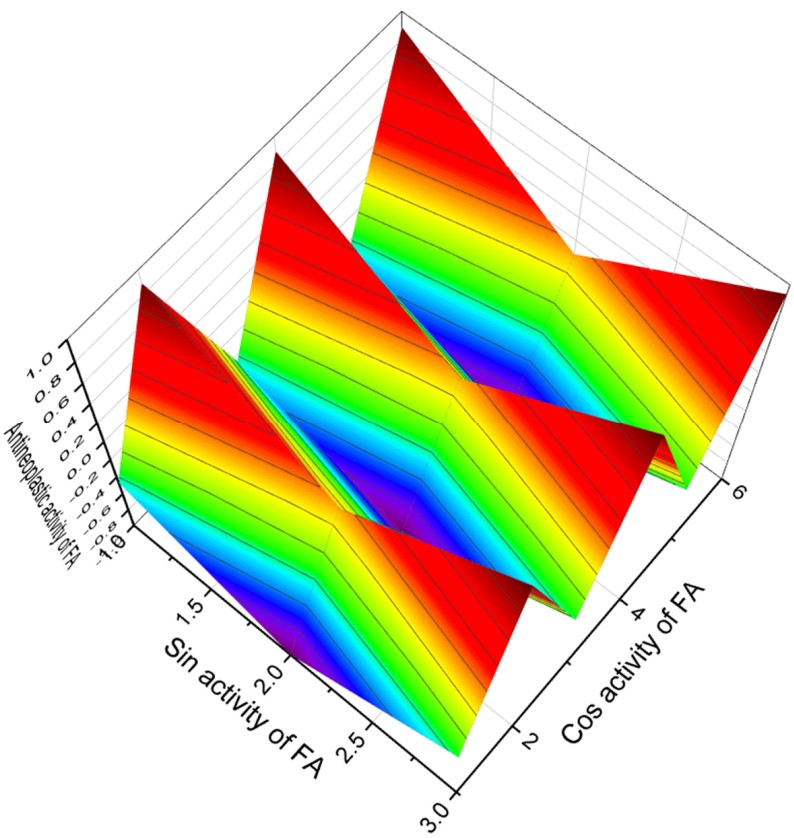

**Figure 57.** 3D graph shows the predicted and calculated antineoplastic activity of FA (**290**, **293** and **299**) with a confidence level of over 94%.

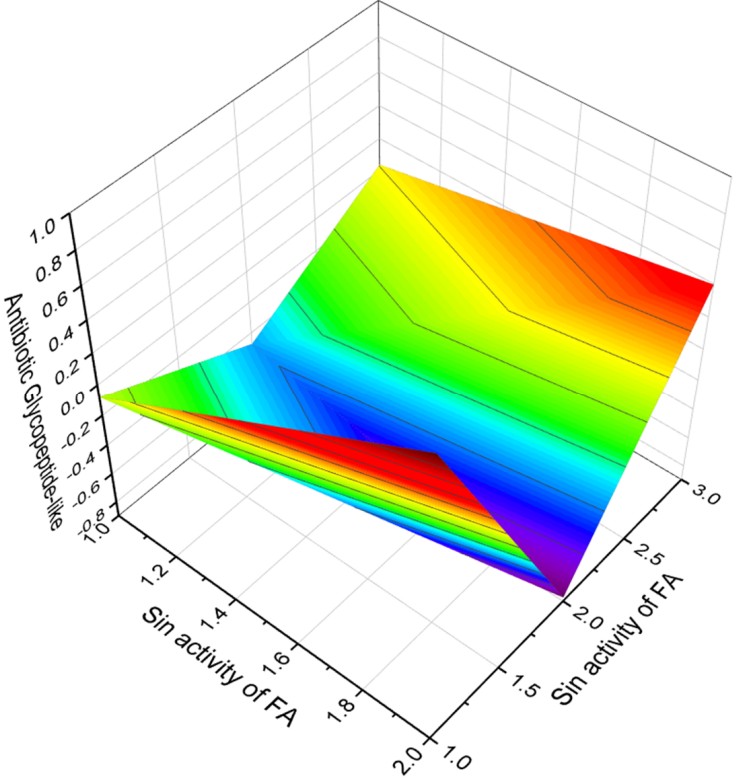

**Figure 58.** 3D graph shows the predicted and calculated antibiotic glycopeptide activity of FA (**297** and **300**) with a confidence level of over 90%.

Verucopeptin with FA (**301**) is an antitumor antibiotic and was found in the culture broth of *Actinomadura verrucosospora* Q886-2 [459,460]. Urauchimycins A and B are antimycin antibiotics and were isolated from a fermentation broth of a *Streptomyces* sp. Ni-80. Both antibiotics showed inhibitory activity against of *Candida albicans*, and contain different FA, 2-(1,2-dihydroxypropyl)-4-methylhexanoic (**302**) and 2-(1,2-dihydroxypropyl)-5-methylhexanoic acid (**303**) [461]. Two cyclic lipopeptides, K97-0239A and B, are produced by Actinomycete *Streptomyces* sp., and both compounds contain (2*E*,4*E*)-13-hydroxytetradeca-2,4-dienoic acid (**304**) [462].

A depsipeptide SCH 58149 containing 3-hydroxy-4-methyloctanoic acid (**305**) was found in the organic extract of the fermentation broth of a fungus of *Acremonium* sp. SCH 58149, which exhibited weak activity against cholesterol ester transfer protein (CETP) with an $IC_{50}$ of 50 mM [463]. Tachykinin (NK2) receptor inhibitors named SCH 378161, SCH 217048, SCH 378199, and SCH 378167 with 2-hydroxy-3-methylhexanoic (**306**) FA were detected in the fermentation broth of a taxonomically unidentified fungus [464].

Chlorinated polyketide peptides named peritoxins were produced only by a pathogenic fungus, Periconia circinata. For both compounds A and B, biologically inactive intermediates, N-3-(E-pentenyl)-glutaroyl-aspartate, circinatin with (*E*)-3-(pent-1-en-1-yl)-pentanedioic acid (**307**), and 7-chlorocircinatin with (2*S*,3*R*)-2-chloro-3-((*E*)-pent-1-en-1-yl)-pentanedioic acid (**308**) were detected only in the culture fluids of the Tox[(+)] strains, and peritoxin B contains FA (**309**); other toxins, peritoxin A, periconin A and B, contain the same FA (**310**) [465,466].

Tetrahydro-2H-pyran containing FA (**289–301**) isolated from fungal and bacterial lipopeptides are of considerable interest, since the lipopeptides themselves show antitumor activity, many of the acids incorporated into these lipopeptides also demonstrate antitumor activity, and some acids are both inhibitors of glycopeptide antibiotics. This is a rather rare function for FA. Figures 56 and 57 show 3D graphs of some biologically active FA.

The mycoparasitic fungus *Acremonium domschii* (NRRL 39465) was obtained from a basidioma of *Rigidoporus microsporus* found on a dead branch in a Hawaiian forest. The crude EtOAc extract of solid-substrate fermentation cultures of *A. domschii* showed significant anti-insectan and antifungal activities, and contained four depsipeptides, named domschisins A–D. Domschisin A exhibited significant antiinsectan activity against *Spodoptera frugiperda*. All isolated compounds contained (2*R*,3*S*,5*S*,6*S*,11*S*)-3,5,11-trihydroxy-2,6-dimethyldodecanoic acid (**311**, for structures see Figure 59, and biological activity see in Table 24) [467]. Depsipeptide β-D-glucosyl-hydroxydestruxin B is produced by the fungus *Alternaria alternata* f. sp. *mali*, belongs to the phytotoxins and contains FA (**312**, 3D graph of activity is shown in Figure 60) [468].

Linear lipopeptides named curmenins contain an α-substituted β-methoxyacrylate, and fatty acids, (2*E*,4*Z*)-2,11-dimethyldodeca-2,4-dienoic (**313**), and (2*E*,4*Z*)-2,10-dimethyl-undeca-2,4-dienoic (**314**) FA, and have been isolated by several higher fungi. Both peptides were inhibitors of the mitochondrial respiratory energy metabolism [469]. Liposidomycines, complex molecules, and two derivatives such as liposomycin A contain (*S*,7*Z*,10*Z*)-3-hydroxyhexadeca-7,10-dienoic acid (**315**, for structure see Figure 61) and liposomycin K contains (*S*,9*Z*,12*Z*)-3-hydroxyoctadeca-9,12-dienoic acid (**316**) [470]. Liposidomycins A, B and C strongly inhibited peptidoglycan synthetase prepared from *Escherichia coli*, and these lipopeptides are synthesized by the fungus *Streptomyces griseosporeus*.

Several lipopeptide antibiotics, friulimicin A–D and lipopeptides A1437 A, A1437 B, A1437 E, A1437 G, were detected in extracts of *Actinoplanes friuliensis*. These compounds showed activity against Gram-positive bacteria, such as methicillin-resistant *Staphylococcus epidermidis* and *Staphylococcus aureus* strains [469]. Friulimicin A and A1437 A contain (*Z*)-11-methyldodec-3-enoic (**317**), friulimicin B and A1437 B-(*Z*)-12-methyltridec-3-enoic (**318**), friulimicin C and A1437 E-(*Z*)-10-methyldodec-3-enoic (**319**), and friulimicin D and A1437 G-(*Z*)-12-methyltetradec-3-enoic (**320**) FA [468]. FA (**317–320**) also found in amphomycin-type lipopeptide antibiotics include: amphomycin (glumamycin) [184,471–484].

Antibiotic F contains (*E*)-10-methyldodec-3-enoic acid (**321**), antibiotics G and H-(*E*)-12-methyl-tetradec-3-enoic acid (**322**) [470].

Glycinocins A–D, types of cyclolipopeptides, were isolated from the fermentation broth of an unidentified Actinomycete species [485]. Glycinocins A and D contain FA (**245**), glycinocin B contains (*E*)-14-methylpentadec-2-enoic (**323**), and glycinocin C contains (*E*)-12-methyltridec-2-enoic acid (**324**). A rare (*R*)-2-hydroxypent-4-enoic acid (**325**) was detected in the toxic cyclodepsipeptides named roseotoxin B and destroxin A, which are produced by the fungus *Trichothecium roseum* [486,487].

**Figure 59.** Unusual and rare FA derived from fungal and bacterial lipopeptides.

The marine and endophytic fungus *Calcarisporium* sp. strain KF525 produced calcaripeptides A, B, and C from the German Wadden Sea [488]. The calcaripeptides A and B contain (6*R*,9*S*,*E*)-9-hydroxy-4,6-dimethyl-3-oxodec-4-enoic acid (**326**), and calcaripeptide C contains (2*S*,4*R*,7*S*)-7-hydroxy-2,4-dimethyl-3-oxooctanoic acid (**327**). The cyclodepsipeptide trichomide A with (2*R*,4*R*)-2,5-dihydroxy-4-methylpentanoic acid (**328**) was isolated from the fermentation products of the fungus *Trichothecium roseum* [489].

**Table 24.** Predicted biological activity of FA derived from fungal peptides.

| No. | Predicted Biological Activity, Pa * |
|---|---|
| 311 | Acute neurologic disorders treatment (0.793); Natural killer cell stimulant (0.711) |
| 312 | Anti-infective (0.945); Antitoxic (0.908); Natural killer cell stimulant (0.900) |
| 313 | Lipid metabolism regulator (0.923); Apoptosis agonist (0.849); Antineoplastic (0.803)Acute neurologic disorders treatment (0.799); Preneoplastic conditions treatment (0.649) |
| 314 | Lipid metabolism regulator (0.923); Apoptosis agonist (0.849); Antineoplastic (0.803)Acute neurologic disorders treatment (0.799); Preneoplastic conditions treatment (0.649) |
| 315 | Lipid metabolism regulator (0.960); Hypolipemic (0.898); Anti-hypercholesterolemic (0.886) |
| 316 | Lipid metabolism regulator (0.960); Hypolipemic (0.898); Anti-hypercholesterolemic (0.886) |
| 317 | Lipid metabolism regulator (0.848); Anti-hypercholesterolemic (0.770); Hypolipemic (0.760) |
| 318 | Lipid metabolism regulator (0.848); Anti-hypercholesterolemic (0.770); Hypolipemic (0.760) |
| 319 | Lipid metabolism regulator (0.930); Anti-hypercholesterolemic (0.842); Hypolipemic (0.830) |
| 320 | Lipid metabolism regulator (0.930); Anti-hypercholesterolemic (0.842); Hypolipemic (0.830) |
| 321 | Lipid metabolism regulator (0.930); Anti-hypercholesterolemic (0.842); Hypolipemic (0.830) |
| 322 | Lipid metabolism regulator (0.930); Anti-hypercholesterolemic (0.842); Hypolipemic (0.830) |
| 323 | Lipid metabolism regulator (0.846); Anti-hypercholesterolemic (0.827) |
| 324 | Lipid metabolism regulator (0.846); Anti-hypercholesterolemic (0.827) |
| 325 | Anti-ischemic, cerebral (0.907); Cell adhesion molecule inhibitor (0.876); Antidiabetic (0.702) |
| 326 | Antineoplastic (0.857); Apoptosis agonist (0.746); Preneoplastic conditions treatment (0.517) |
| 327 | Anti-ischemic, cerebral (0.835); Acute neurologic disorders treatment (0.783); Hypolipemic (0.749) |
| 328 | Anti-ischemic, cerebral (0.845); Leukopoiesis stimulant (0.783); Antitoxic (0.675) |
| 329 | Hypolipemic (0.879); Lipid metabolism regulator (0.825); Anti-hypercholesterolemic (0.769) |
| 330 | Acute neurologic disorders treatment (0.947); Antineoplastic (0.816); Apoptosis agonist (0.771) |
| 331 | Preneoplastic conditions treatment (0.770); Acute neurologic disorders treatment (0.757) |
| 332 | Lipid metabolism regulator (0.937); Acute neurologic disorders treatment (0.832) |
| 333 | Lipid metabolism regulator (0.937); Hypolipemic (0.866)Acute neurologic disorders treatment (0.832); Atherosclerosis treatment (0.653) |

* Only activities with Pa > 0.5 are shown.

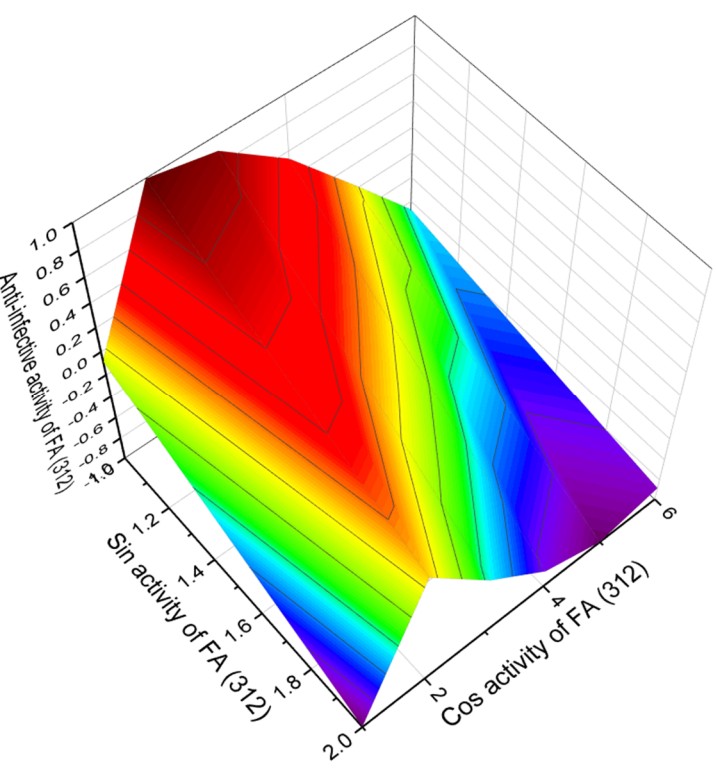

**Figure 60.** 3D graph shows the predicted and calculated anti-infective activity of glycosidic FA (**312**) with a confidence level of over 94%. This acid produced by fungus *Alternaria alternata* f. sp. *mali*.

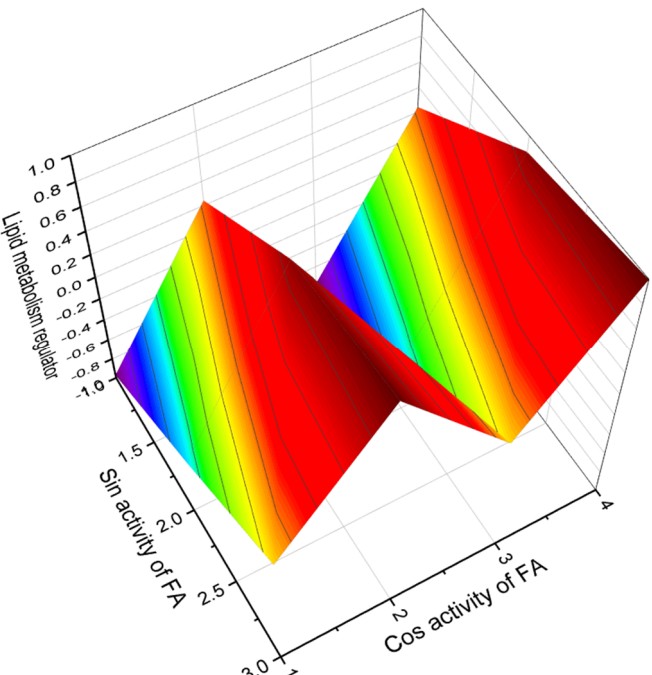

**Figure 61.** 3D Graph shows the predicted and calculated activity of FA (**315** and **316**) as lipid metabolism regulators with a confidence level of over 96%. Both unsaturated acids have been incorporated into liposidomycines A and K which are synthesized by the fungus *Streptomyces griseosporeus*.

The marine-derived fungus *Penicillium purpurogenum* G59 from the unidentified sponge produced antitumor lipopeptides, penicimutanin A, and penicimutanin B, and these drugs contain (*E*)-4,6-dimethyldodec-2-enoic acid (**329**) [490].

The lipopeptide topostatin with (8*E*,10*E*)-3-hydroxy-2,6,10,13-tetramethyl-7-oxoicosa-8,10-dienoic acid (**330**) is an inhibitor of topoisomerases and was isolated from the culture filtrate of *Thermomonospora alba* strain No. 1520 [491]. The cyclic lipopeptides pneumocandin A and pneumocandin B produced by the fungus *Glarea lozoyensis* contain (10*S*,12*R*)-10,12-dimethyltetradecanoic acid (**331**). [492–494]. The liquid culture broth of *Pseudomonas* sp. MF381-IODS yielded two antimicrobial peptides named pseudotrienic acid A and B, and both compounds contain (3*E*,5*E*)-7-hydroxy-4-methylhexadeca-3,5-dienoic (**332**) and (3*E*,5*E*)-7-hydroxy-4-methyltetradeca-3,5-dienoic (**333**) FA [495].

Cyclodepsipeptide derivatives named emericellamides A and B were produced by the marine-derived fungus *Emericella* sp. strain CNL-878 [496]. Emericellamides A, C, D, E, and F were also found in *Aspergillus nidulans.* Emericellamide A contains (2*R*,3*R*,4*S*)-3-hydroxy-2,4-dimethyldecanoic acid (**334**, for structures see Figure 62, and activity is shown in Table 25), and emericellamide B contains (2*R*,3*R*,4*S*,6*S*)-3-hydroxy-2,4,6-trimethyldodecanoic acid (**335**).

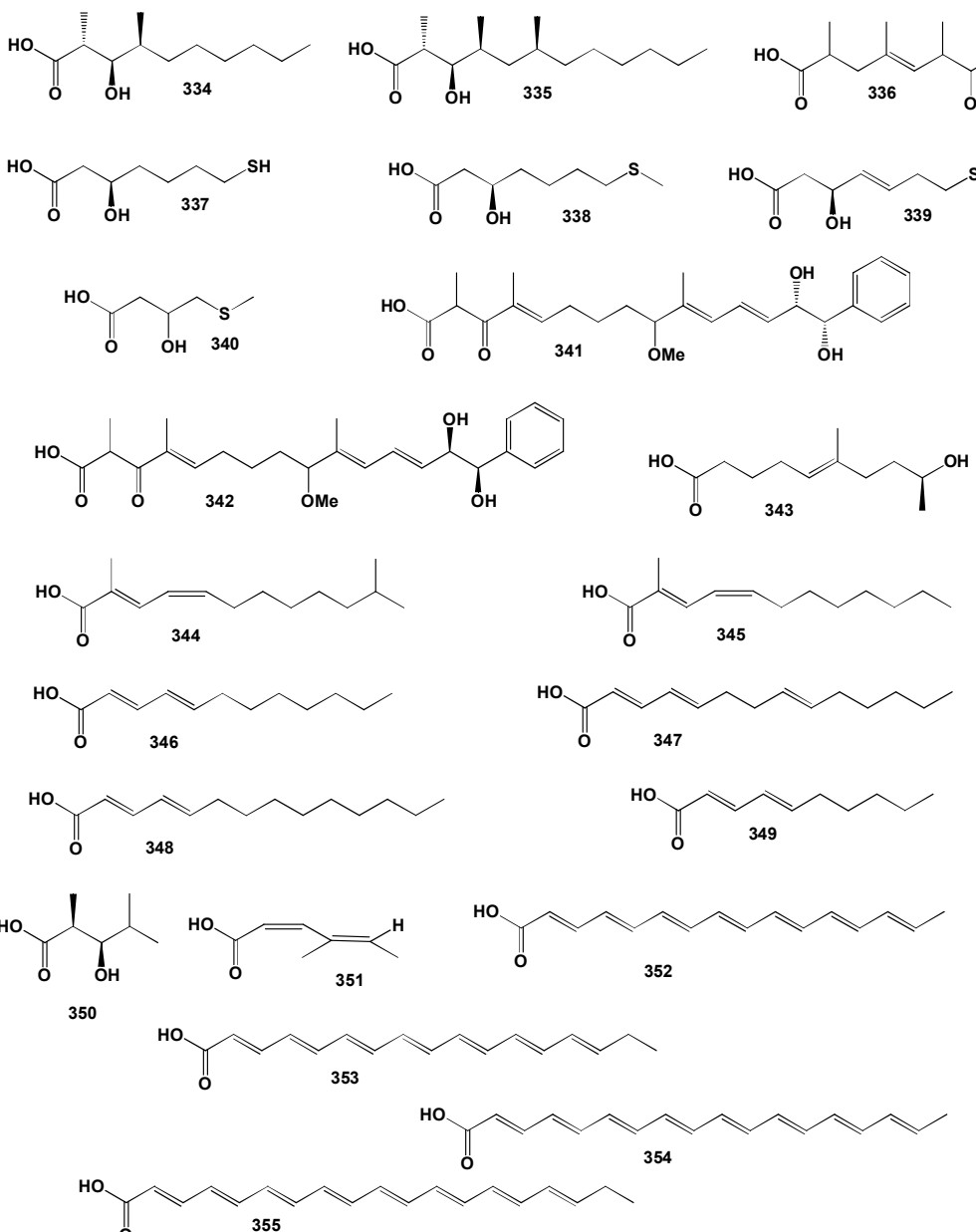

**Figure 62.** Unusual FA derived from fungal and bacterial lipopeptides.

**Table 25.** Predicted biological activity of FA from fungal peptides.

| No. | Predicted Biological Activity, Pa * |
|---|---|
| 334 | Natural killer cell stimulant (0.795); Leukopoiesis stimulant (0.784); Antineurotic (0.700) |
| 335 | Hypolipemic (0.828); Acute neurologic disorders treatment (0.774); Leukopoiesis stimulant (0.736) |
| 336 | Hypolipemic (0.880); Antineoplastic (0.858); Lipid metabolism regulator (0.732) |
| 337 | Lipid metabolism regulator (0.859); Acute neurologic disorders treatment (0.811) Hypolipemic (0.800); Mucositis treatment (0.756); Antidiabetic symptomatic (0.696) |
| 338 | Lipid metabolism regulator (0.891); Hypolipemic (0.861); Anti-hypercholesterolemic (0.784) |
| 339 | Lipid metabolism regulator (0.888); Acute neurologic disorders treatment (0.761) |
| 340 | Hypolipemic (0.816); Lipid metabolism regulator (0.793); Mucositis treatment (0.779) |
| 341 | Antifungal (0.836); Antibacterial (0.653); Antiparasitic (0.614) |
| 342 | Antifungal (0.836); Antibacterial (0.653); Antiparasitic (0.614) |
| 343 | Lipid metabolism regulator (0.952); Antineoplastic (liver cancer) (0.909) Anti-hypercholesterolemic (0.815); Antineoplastic (0.777); Atherosclerosis treatment (0.649) |
| 344 | Lipid metabolism regulator (0.923); Apoptosis agonist (0.849); Antineoplastic (0.803) Hypolipemic (0.712); Atherosclerosis treatment (0.651); Preneoplastic conditions treatment (0.649) |
| 345 | Lipid metabolism regulator (0.942); Antiviral (Arbovirus) (0.903); Hypolipemic (0.741) Acute neurologic disorders treatment (0.728); Antiviral (Picornavirus) (0.676) |
| 346 | Antiviral (Arbovirus) (0.952); Lipid metabolism regulator (0.903); Antiviral (Picornavirus) (0.790) |
| 347 | Antiviral (Arbovirus) (0.952); Lipid metabolism regulator (0.903); Antiviral (Picornavirus) (0.790) |
| 348 | Antiviral (Arbovirus) (0.952); Lipid metabolism regulator (0.903); Antiviral (Picornavirus) (0.790) |
| 349 | Antiviral (Arbovirus) (0.952); Lipid metabolism regulator (0.903); Antiviral (Picornavirus) (0.790) |
| 350 | Platelet antagonist (0.800); Anticoagulant (0.702); Fibrinolytic (0.700) |
| 351 | Apoptosis agonist (0.879); Antineoplastic (0.878); Preneoplastic conditions treatment (0.618) |
| 352 | Antiviral (Arbovirus) (0.861); Lipid metabolism regulator (0.844); Antiviral (Picornavirus) (0.776) |
| 353 | Antiviral (Arbovirus) (0.917); Antiviral (Picornavirus) (0.781); Antimutagenic (0.699) |
| 354 | Antiviral (Arbovirus) (0.861); Lipid metabolism regulator (0.844); Antiviral (Picornavirus) (0.776) |
| 355 | Antiviral (Arbovirus) (0.917); Antiviral (Picornavirus) (0.781); Antimutagenic (0.699) |

* Only activities with Pa > 0.5 are shown.

The depsipeptides named chondramides A–D were produced by several myxobacteria from the genus Chondromyces, and isolated compounds contain (*E*)-7-hydroxy-2,4,6-trimethyloct-4-enoic acid (**336**) [497]. The fungus *Schizosaccharomyces pombe* produced the antitumor antibiotic depsipeptide, FK228, and this drug contained an unusual (*R*)-3-hydroxy-7-mercaptoheptanoic acid (**337**) and (*R*)-3-hydroxy-7-(methylthio)-heptanoic acid (**338**) [498].

(*S,E*)-3-hydroxy-7-mercaptohept-4-enoic acid (**339**) was present in an anticancer depsipeptide named romidepsin, which was isolated from a culture of a Gram-negative, facultative anaerobic, coccobacillus known as *Chromobacterium violaceum* [499].

The macrocyclic depsipeptide named ngercheumicin D with 3-hydroxy-4-(methylthio)-butanoic acid (**340**) produced by *Photobacterium* strains was active against the non-pathogenic *Pseudovibrio denitrificans* [500,501]. Aetherobacter yielded the cyclic peptides aetheramides A and B. Aetheramides showed cytostatic activity against human colon carcinoma (HCT-116) cells with IC$_{50}$ values of 0.11 μM. Both peptides contained (4*E*,10*E*,12*E*,14*S*,15*S*)-14,15-dihydroxy-9-methoxy-2,4,10-trimethyl-3-oxo-15-phenyl-pentadeca-4,10,12-trienoic (**341**) acid and (4*E*,10*E*,12*E*,14*R*,15*R*)-14,15-dihydroxy-9-methoxy-2,4,10-trimethyl-3-oxo-15-phenyl-pentadeca-4,10,12-trienoic acid (**342**), respectively [502].

The depsipeptides known as miuraenamides A–D are produced by a slightly halophilic myxobacterial strain, SMH-27-4 [503]. All miuraenamides contain (*S,E*)-9-hydroxy-6-methyldec-5-enoic acid (**343**). Antifungal metabolites, cyrmenin A with (2*E*,4*Z*)-2,11-dimethyldodeca-2,4-dienoic acid (**344**) and cyrmenin B with (2*E*,4*Z*)-2-methyldodeca-2,4-dienoic acid (**345**), have been isolated from *Cystobacter armeniaca* and *Archangium gephyra*, respectively [504,505].

Antitumor agents BU-2867T A, B, and C with (2*E*,4*E*)-dodeca-2,4-dienoic acid (**346**, 3D graph sees in Figure 63), (2*E*,4*E*,8*E*)-tetradeca-2,4,8-trienoic acid (**347**), and (2*E*,4*E*)-tetradeca-2,4-dienoic acid (**348**) were produced by *Polyangium brachysporum* sp. nov [506,507]. Peptide antibiotics designated herein as *BU-2867T* F with (2*E*,4*E*)-deca-2,4-dienoic (**349**) and G with (**346**) FA were produced by fermentation of the *Polyangium brachysporum* strain K481-B101 [507].

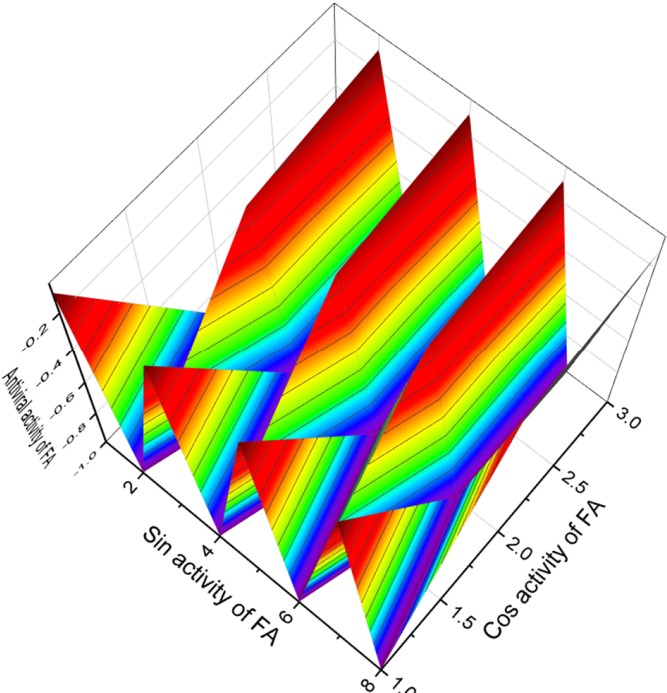

**Figure 63.** 3D graph shows the predicted and calculated antiviral activity of FA (**346**, **347**, **348** and **349**) with a confidence level of over 95%. All dienoic acids were produced by proteobacteria *Polyangium brachysporum*.

A rare class of bicyclic depsipeptide antibiotics, desmethylsalinamide C and salinamide A, were derived from the marine *Streptomyces* sp. CNB-091 [508] (specimens of *Streptomyces* found in various ecosystems, see in Figure 64). Desmethylsalinamide C contains (2*S*,3*R*)-3-hydroxy-2,4-dimethylpentanoic acid (**350**) and salinamide A contains (2*Z*,4*E*)-4-methylhexa-2,4-dienoic acid (**351**).

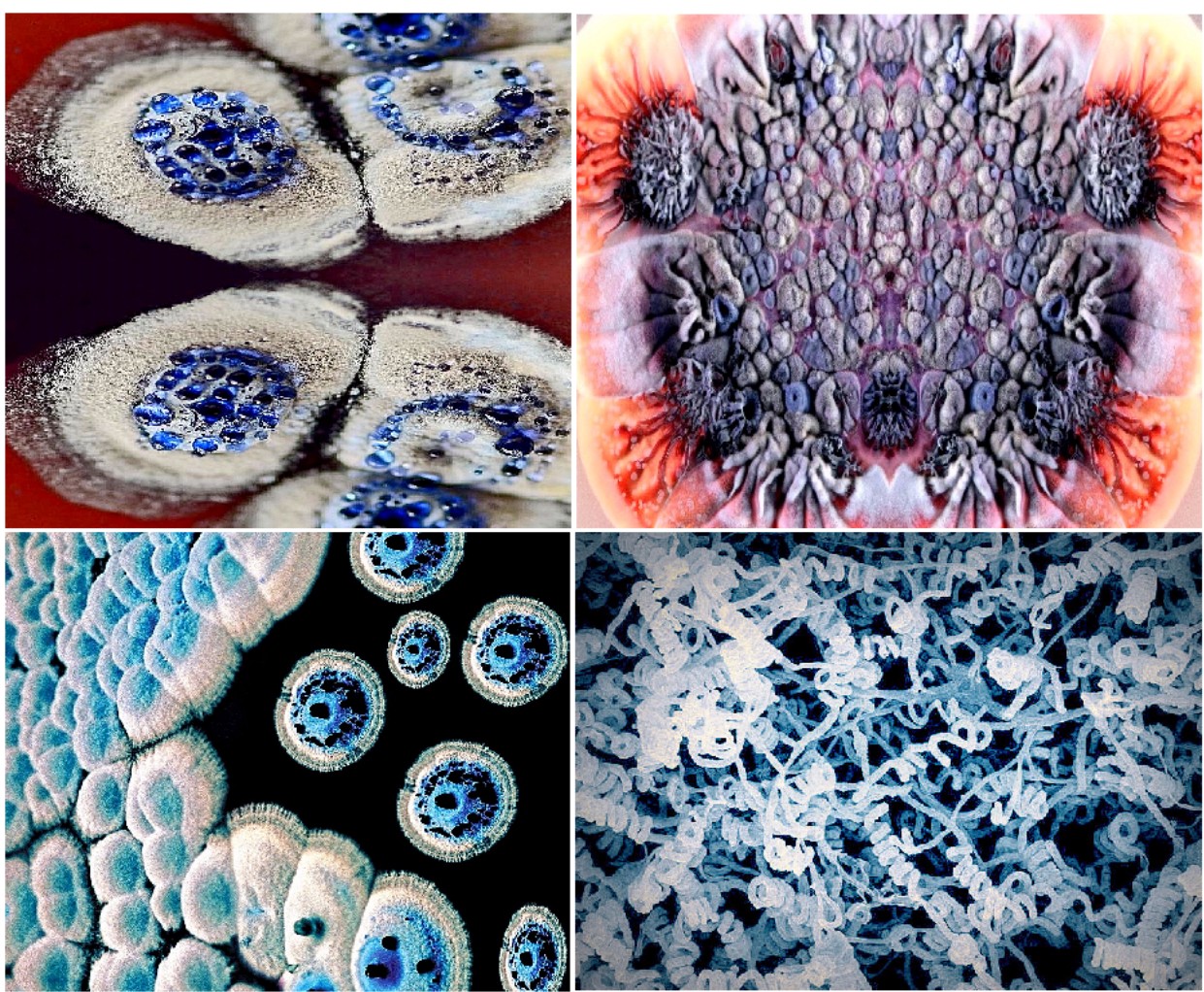

**Figure 64.** Specimens of *Streptomyces* fungus of various strains inhabiting the various ecosystems. Many fungi belonging to this genus synthesizes a lot of biologically active metabolites, including linear and cyclic peptides and their FA. Pictures adapted by the author.

Myxobacterial species such as *Myxococcus xanthus* and *Stigmatella aurantiaca* produce cyclic depsipeptides, myxochromides with an unsaturated polyketide side chain [509]. Thus, myxochromide S contains (**352**) acid, myxochromides A2 and S2 contain FA (**353**), A3 and S3—(**354**), and A4—(**355**) FA.

In this group, we included lipopeptides that are produced by Actinomycetes, fungal endophytes and fungi. A very interesting group, since these microorganisms themselves synthesize many biologically active substances. Many FAs are similar in structure and diversity to bacterial FA. Apparently, many of these microorganisms are symbionts in more complex biological structures.

## 7. Conclusions

The present review is devoted to an interesting topic of studying the biological activity of FAs that are part of linear and cyclic peptides produced by organisms living in both marine and freshwater habitats. The most extensively studied lipopeptides are marine and, to a lesser extent, freshwater invertebrates.

The study of FAs in various types of complex molecules such as neutral lipids, glyco- and phospholipids isolated from various organisms is of great interest to biochemists and molecular biologists due to the high biological activity of FAs. In recent years, more and more attention has been paid to lipopeptides by scientists since these complex molecules have pronounced and specific biological activities. Given such a great interest in this group of natural complex molecules, we tried to combine known and published data on the activity of lipopeptides and their constituent fragments. This review presents the biological activity of both individual lipopeptides, and more than 350 FAs incorporated into these molecules.

Based on the presented data on the biological activity of lipopeptides and individual fragments such as FA, we can draw a preliminary conclusion that the study of the activity of such complex molecules as lipopeptides is already a fait accompli, however, in the future, an important area of research is to determine the biological activity of individual fragments, in particular FA, as well as amino-containing fatty acids.

Of undoubted interest are activities that demonstrate FAs incorporated into lipopeptides. So, several fatty acids showed pronounced antibacterial, antineurotic, antimicrobial, antitoxic, antifungal, or antitumor activity. In addition, FAs were found that exhibited rare beneficial properties, such as antiparasitic, antidiabetic, anthelmintic, anti-inflammatory, and anti-psoriasis effects. Separate fatty acids were stimulants of leukopoiesis, natural killer cells, and had anti-infective action.

We must be aware that the activities of FA incorporated into lipopeptides presented in this review are computer simulations. In the pool of this program (QSAR) there are about 1,000,000 natural and synthesized complex molecules that are associated with more than 10,000 experimentally obtained biological activities. The prospects for this predictive activity (QSAR) in medicine and pharmacology are developing very rapidly, and the probability of predicting the activity of simple and complex molecules currently reaches about 90% in many cases. Currently, about 20 million organic and inorganic compounds have been synthesized, but their biological activity has not yet been determined, and using the QSAR method, several new useful active molecules can be identified to combat numerous human diseases.

For readers and researchers who are interested in PASS, they can go to the website of this program [510]. In addition, those who wish can use this program to determine any organic molecule, both of natural origin and synthetic type, may find it useful. The site also describes all the details of the PASS.

**Funding:** This work did not receive any specific grant from funding agencies in the public, commercial, or not-for-profit sectors.

**Institutional Review Board Statement:** Not applicable.

**Informed Consent Statement:** Not applicable.

**Data Availability Statement:** Not applicable.

**Acknowledgments:** The author is grateful to *Tatyana A. Gloriozova* (Institute of Biomedical Chemistry, Moscow, No.122030100170-5) for prompt help in determining the biological activity of lipopeptides and FA presented in this article.

**Conflicts of Interest:** The author declares that he has no known competing financial interest or personal relationships that could affect the work described in this article.

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
