# Peer review of "Hydrobiological Aspects of Fatty Acids: Unique, Rare, and Unusual Fatty Acids Incorporated into Linear and Cyclic Lipopeptides and Their Biological Activity"

_2673-9917, doi:10.3390/hydrobiology1030024_

Round 1

Reviewer 1 Report

The review "Hydrobiological Aspects of Fatty Acids. Unique, Rare, and Unusual Fatty Acids Incorporated into Linear and Cyclic Lipopeptides and their Pharmacological Profile" is interesting and well written nut certain points required more attention by the author

Among these points:

1- In the title I would rather recommend the use of biological instead of pharmacological to be more general

2- The photos in the figures should be clearly mentioned if it is the author own photos or they are adopted from other sources and in all cases the magnification power should be added for each 

3- The structures in each figure should be grouped with similar skeleton and use R to reduce the total number of the structures 

4- The conclusion part is the weakest part of the review, the author should summarize all what was written in the main review giving the main take home message, shedding lights on the future perspectives that might guide the audience about what is still missed to be done 

5- I would recommend if the author could comment even briefly about each section to give more comprehensive review rather than just copy, past and paraphrase.

6- The review should be checked by an English native speaker to remove some syntax 

Author Response

The author is grateful to referee 1 for a detailed review of the review. I give detailed answers to all questions.

 All fixes are marked in green.

  1. We have replaced pharmacological with biological. I agree that this is a more general name and corresponds to the article.

  1. An explanation is given for all photos that they are taken from sites where their use for non-commercial purposes is allowed. Highlighted in the text.

  1. Structures are grouped for each species of invertebrates and algae. As shown in the text.

4 and 5. The final part is completely rewritten, according to the recommendations. Each section is commented.

  1. The review was submitted to the Department of English Philology to improve the quality of the article.

Reviewer 2 Report

This is an excellent review of the biological activities of about 350 different fatty acids that are produced in both marine freshwater systems as part of lipopeptides.  It represents a substantial contribution to the field.  It is not clear, however, if the biological activities of these fatty acids are different when these fatty acids are in either the: 1) free fatty acid or 2) lipopeptide forms.  The author should make this clear in the revised MS.   The MS is quite well written, but it does need some minor revisions.

Line 64 - Should be re-written to state: ...with about 40% of them being lipopeptides.

Line 164 - Should be revised to state:...of all natural products...

Line 1634 - Should be revised to state:...living in both marine and freshwater habitats.

Tables 2 through 22 - It is not clear what the numbers given in parentheses indicate.  Please provide this information in the revised tables.

Author Response

Reviewer 2

The author is grateful to referee 2 for a detailed review of the review. I give detailed answers to all questions.

  All fixes are marked in green.

Figure 1 shows the structure of the lipopeptide, cryptophycin-1, and its constituent fragments. Table 1 shows their activities. According to these data, Cryptophycin-1 and acid have anticancer activity.

Fixed on lines 64, 164, and 1634. Explanations are given for the tables.

The conclusion has been completely rewritten.

Round 2

Reviewer 1 Report

The work has been improved few typos and syntax should be checked